# Elevated temperature fatally disrupts nuclear divisions in the early *Drosophila* embryo

Girish Kale [1,2] ✉, Pratika Agarwal[1], J. Jaime Diaz-Larrosa[1] & Steffen Lemke [1,2] ✉

Temperature variations challenge animal survival, with elevated temperatures presenting distinct vulnerabilities throughout the animal life cycle. Embryonic development is especially vulnerable, though molecular mechanisms remain unclear. We address this in *Drosophila melanogaster*, an insect model with extensively characterized embryonic development. Here we show that pre-gastrulation development - syncytial blastoderm formation and cellularization - is particularly vulnerable to elevated temperature. Embryos exposed to elevated temperature during this period exhibit gastrulation defects and increased lethality. We observe mitotic failures causing loss of cortical nuclei during cellularization, preceded by a local nuclear crowding and increased division asynchrony; both features cooperatively amplify mitotic failures, leading to blastoderm holes. Mitotic failures trigger DNA damage response due to weakened cytoskeletal interaction. Genetic rescue experiments support the hypothesis that cortical F-actin and astral microtubule interaction is disrupted at elevated temperatures, causing mitotic failures. We propose that expression levels of corresponding genes could predict how increasing temperature variations affect insect populations.

It has been suggested that the current rise in average global temperatures and the amplitude of temperature fluctuations can affect species fitness and result in a loss of biodiversity[1]. It is unclear whether effects on species fitness are direct consequences of the temperature rise, particularly since many species are likely experiencing a rather gradual departure from optimal conditions of temperature, which have previously evolved in a species-specific manner. Previous studies suggest that, under some circumstances, even such gradual departures could affect animal survival. For instance, studies have shown that a gradual departure from normal temperatures, e.g., experienced by fevers during pregnancy produce birth defects in humans[2–4], indicating that temperature conditions tolerable by an adult may be more detrimental to the embryo. This raises the possibility that specific aspects of embryo development are more vulnerable to temperature increase. Consistent with this notion, previous studies in model organisms have shown that temperature is a key factor that determines the fidelity of embryonic development; rat and zebrafish embryos show an increase in developmental defects at elevated temperatures[5–8]. Yet the mechanisms underlying vulnerability in embryonic development at elevated temperatures remain unclear. Here we explore whether exposure to elevated temperature is detrimental to embryonic development in the insect model *Drosophila melanogaster* (henceforth, *Drosophila*), and examine the consequences in terms of population-level parameters such as embryo survival.

In nature, animals rarely experience highly steady temperatures, but they do experience a range of temperatures to which they are generally well adapted. Increasing temperatures accelerate diverse physiological processes, including development, growth, and regeneration, making the timing of events temperature-dependent[9]. Developmental processes can be viewed as networks of biochemical reactions occurring simultaneously or in rapid succession. The

[1]Centre for Organismal Studies, University of Heidelberg, Heidelberg, Germany. [2]Institute of Biology, University of Hohenheim, Stuttgart, Germany.
✉e-mail: girish.kale@uni-hohenheim.de; steffen.lemke@uni-hohenheim.de

Arrhenius equation, long used to describe the temperature dependence of biochemical reactions, has more recently been applied to capture the effects of temperature on the rate of physiological processes, particularly during embryonic development[10,11]. However, in contrast to chemical reactions in vitro, where temperatures often exceed physiological ranges, complex biological processes in vivo—such as embryonic development—follow the Arrhenius equation, provided that temperatures remain within the relatively narrow range typically experienced by the organism. With a transition from 28 to 30 °C, *Drosophila* no longer exhibit an acceleration of embryonic development[10–12], suggesting that 29 °C marks a threshold at which elevated temperatures challenge development and exceeds the physiological range over which the Arrhenius equation accurately describes developmental rates, while lower temperatures remain effectively normal. To test this hypothesis, we analyzed the fidelity of development at an elevated temperature of 29 °C, compared with controls at the normal culture temperature of 25 °C.

Previous studies have found that *Drosophila* embryo development beyond the first 4–6 h is robust to even acute exposure to high temperatures[13], suggesting that potential embryonic vulnerabilities are likely to occur and be observed during the first few hours of development. *Drosophila* embryo development starts with fertilization and egg deposition. Following fertilization, the embryo undergoes nine rounds of syncytial nuclear divisions within the yolk. Following these divisions, most nuclei are anchored to the periphery of the embryo, where four more rounds of nuclear division take place during a stage called syncytial blastoderm[14]. These divisions are facilitated by pseudo-cleavage furrows, i.e., temporary membrane invaginations that resemble a half-cell and slowly form around the nuclei during interphase-metaphase. They are subsequently reabsorbed into the embryo cortex rapidly during anaphase-telophase[15]. Following a total of 13 nuclear divisions, the resulting ~6000 nuclei simultaneously segregate into individual cells and produce the blastoderm epithelium in a process called cellularization[16]. This occurs at about 3 hrs from fertilization. It marks the beginning of gastrulation, which involves changes in cell shape[17], rearrangements of cell contacts[18], and further rounds of cell divisions in isolated mitotic domains[19]. All of these events facilitate early morphogenesis.

In our experiments, we found that exposing embryos to 29 °C during the first 3 h of embryonic development leads to a significant increase in population-level embryonic lethality during the gastrulation stage. Examinations at the cell and tissue level revealed mitotic failures, which lead to the loss of nuclei during the syncytial blastoderm and especially the cellularization stage. Additionally, embryos developing at 29 °C showed nuclear crowding in the central region of the embryo and greater asynchrony of nuclear division cycles between the central regions and at the poles. We show that nuclear crowding and nuclear cycle asynchrony combine to increase the local propensity of mitotic failures. Local clusters of mitotic failures lead to regions devoid of nuclei/cells ("holes"), which disrupts the continuity of the blastoderm as primary epithelial tissue. The molecular dissection of mitotic failures in the blastoderm revealed that they trigger a DNA damage response, which mitotically arrests the defective daughter nuclei, resulting in their expulsion from the embryo cortex. We show that cytoskeletal elements, F-actin and microtubules, show incongruent dynamics at 29 °C, suggesting a weakening of their interaction during mitosis, implicating mitotic failures at the core of developmental vulnerabilities at elevated temperature. In an attempt to enhance the resilience of nuclear divisions at 29 °C and rescue temperature-induced developmental defects, we overexpressed specific genes that regulate cell polarity, microtubule-F-actin interactions, and mitotic spindle orientation. Our results suggest a genetically rescuable vulnerability in microtubule-F-actin interactions during nuclear divisions at elevated temperatures. Notably, the genes we identified as capable of providing a genetic rescue to temperature-induced developmental defects could be recovered in a genome-wide analyses of data on genetic variation in natural populations along temperature clines. In summary, our work identifies cell biological defects caused by elevated temperatures and links them to population-level lethality and fitness. This offers insights into the broader implications of vulnerability to elevated temperatures during a defined time span in the life history of flies and provides an initial handle to quantitatively and mechanistically predict embryo fitness at elevated temperature.

## Results

### Early exposure to elevated temperature reduces embryo survival due to gastrulation defects

To explore how elevated temperature affects the early development of *Drosophila* embryos, we exposed young embryos to 29 °C and asked how many of them survived and how they developed. To investigate alterations in the rate of embryos hatching into larvae, we first differentiated the effects of temperature exposure by timing: from the moment of deposition until just before gastrulation begins (~3 h), and during a similar time starting from gastrulation onset (refer to Fig. 1A for experimental temperature regimen). In both scenarios, we observed a notable increase in embryonic lethality (Fig. 1B), pinpointing a critical vulnerability especially during the pre-gastrulation phase of development. This finding was unexpected, as prior research suggested that the genetic mechanisms governing embryonic patterning during this phase were resilient to temperature fluctuation[20,21], indicating a more systemic effect, possibly rooted in cell biology.

To address when and how embryos died after early exposure to 29 °C, we maintained embryos until late cellularization stage at 29 °C and then recorded their development using confocal microscopy time-lapse imaging at 25 °C. Embryonic development in these recordings was analyzed qualitatively as well as quantitatively. Qualitatively, many embryos displayed moderate anomalies during gastrulation, including twisting along the embryo's anterior-posterior axis (Fig. 1C, D), exemplifying transient defects in the bilateral symmetry of embryonic development. Notably, these twisted embryos seemed to recover from their defects and continued to develop. By contrast, a minority of embryos exhibited severe gastrulation defects, culminating in developmental cessation and lethality (Fig. 1E). Quantitatively, surviving embryos after exposure to 29 °C did not differ statistically significantly from those after control exposure to 25 °C, neither in the rate of germband extension (GBE, Fig. 1F) nor in the timing of other key developmental milestones (Fig. 1G). Overall, this suggests that if embryos surpass the gastrulation stage, they proceed without detectable developmental delays.

Taken together, our findings underline increased embryonic lethality during gastrulation due to prior exposure to 29 °C, highlighting the detrimental effects of elevated temperature. Previous work has demonstrated that genetic patterning is robust to temperature perturbations[20,21], suggesting that the observed defects are unlikely to be directly caused by a failure of embryonic patterning per se. To explain the observed fatal defects during gastrulation stage, we instead hypothesized that nuclei, who ultimately interpret embryonic patterning information into cell and tissue movements, might have exhibited fatal defects in their dynamic behavior during blastoderm formation. To address this possibility, we investigated nuclear dynamics during pre-gastrulation development.

### A clustered loss of nuclei at elevated temperature likely causes gastrulation defects and embryo lethality

Pre-gastrulation development in flies and other insects involves nuclei dividing deep within the yolk before their arrival at embryo's cortex, where they must be securely anchored while undergoing further rounds of division. Ultimately, the tight and even cortical distribution of those syncytial nuclei is crucial for the generation of blastoderm

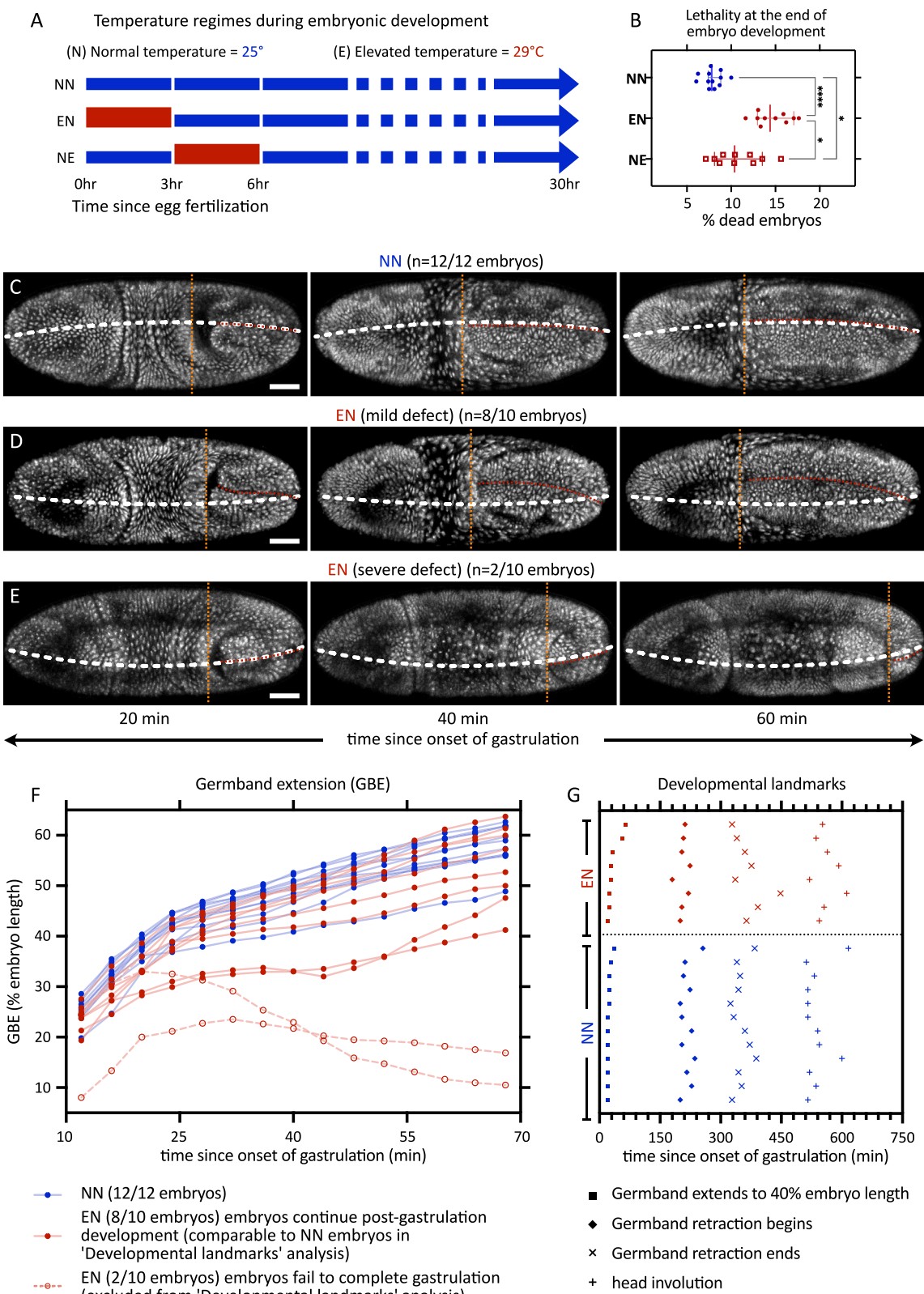

Germband extension (GBE) legend:

- NN (12/12 embryos)
- EN (8/10 embryos) embryos continue post-gastrulation development (comparable to NN embryos in 'Developmental landmarks' analysis)
- EN (2/10 embryos) embryos fail to complete gastrulation (excluded from 'Developmental landmarks' analysis)

Developmental landmarks legend:

- ■ Germband extends to 40% embryo length
- ◆ Germband retraction begins
- × Germband retraction ends
- + head involution

epithelium and its subsequent morphological changes during gastrulation.

To address whether defects during gastrulation could stem from defective nuclear distribution rather than defective blastoderm patterning, we focused our analyses on the distribution of cortical nuclei during cellularization, i.e., just before gastrulation, using fixed embryos stained for DNA to visualize nuclei in the embryo volume (see "Methods"). In embryos developing at 29 °C, we observed a subcortical cluster of nuclei in a small fraction of embryos, corresponding to a hole in the cortical layer of nuclei (Fig. 2A). To better visualize and differentiate cortical and subcortical nuclei, we segmented the embryo volume and color-coded the image voxels based on their distance from the embryo surface (Fig. 2B). We noticed a few subcortical nuclei in embryos developing at normal temperature (Fig. 2C, F, I, J), while a

**Fig. 1 | Embryonic lethality and developmental defects in embryos exposed to elevated temperature. A, B** To assess developmental robustness, we exposed embryos to various temperature regimes and estimated the embryo lethality. **A** Schematic showing the temperature regimes in the embryo survival experiment. Here we have three conditions depending on exposure to normal or elevated temperature regimen in the 3-h time windows as shown, normal-normal [NN], elevated-normal [EN], and normal-elevated [NE]. In all cases, the first time-window also includes 1 h of embryo collection. Embryonic development beyond 6 h continued at normal temperature. **B** The scatter plots show the distribution of the percentage of embryos that don't hatch into larvae. $n = 12$ (NN), 10 (EN), 10 (NE) samples; each sample has ≥26 embryos. **C–E** Gastrulation defects after exposure to elevated temperature. Montages show embryos, expressing Histone-RFP, from early gastrulation till late gastrulation, in embryos at 25 °C. Till late cellularization stage, these embryos were exposed to either 25 °C, i.e., NN temperature regimen

(C), or 29 °C, i.e., EN temperature regimen, showing mild (**D**), or severe (**E**) defect. White dashed lines, dorsal midline drawn based on the egg shape; red dashed lines, the embryonic midline in the dorsal posterior ectoderm; orange dashed lines, anterior end of the posterior mid-gut (extent of GBE). **F, G** Quantification of developmental defects after exposure to elevated temperature. **F** Plot show extent of GBE: hollow markers, embryos with gastrulation defects, which are excluded from further analysis of developmental landmarks. **G** The analysis of Developmental landmarks shows the time at which various events can be seen in the embryos. Time points from individual embryos are plotted at the same $y$-value. Scale bars, 50 µm. **B** show scatters with median, with whiskers showing 95% CI. Non-parametric Kruskal–Wallis test with Dunn's uncorrected test for multiple comparisons: *, $p < 0.05$; **, $p < 0.01$; ***, $p < 0.001$; ****, $p < 0.0001$, non-significant comparisons are not shown. Source data and exact p-values for plots are available in the Source Data file.

large fraction of embryos developing at 29 °C showed an increase in subcortical nuclei, albeit not in large clusters (Fig. 2D, G, K, L). Notably, however, in a small fraction of embryos developing at 29 °C, we observed a notable void in the cortical layer of nuclei (henceforth, blastoderm hole), coinciding with a subcortical cluster of at least 10 nuclei (Fig. 2E, H, M, N). Within the coordinate system of the embryo, the distribution of neither isolated subcortical nuclei nor blastoderm holes was left-right symmetric. Additionally, even if blastoderm holes were predominantly observed in the central region of the embryo, their location varied across embryos (Fig. 2O, P). These results further support the idea that loss of nuclei and blastoderm holes are unlikely to be directly caused by defective patterning: embryonic patterning is left-right symmetrical and accordingly, observed defects in patterning would likewise show left-right symmetry.

To investigate the underlying cause for the appearance of subcortical nuclei, we followed nuclear dynamics using time-lapse recordings: Histone-RFP was used to visualize chromatin, and Jupiter-GFP to visualize microtubules for keeping track of nuclear division cycles. Starting with the first appearance of nuclei at the embryo cortex (occurring during the interphase of the 10th nuclear cycle), we observed nuclei through their cortical divisions during syncytial blastoderm (through mitoses 10–13), then through cellularization (interphase 14), and until the onset of gastrulation (beyond interphase 14). To determine whether the nuclear loss at 29 °C reflects a threshold response associated with a transition to non-Arrhenius behavior, we examined temperature dependencies additionally at 22, 25, and 28 °C, temperatures within the normal physiological range.

The cortical arrival of nuclei during interphase 10, from the yolk to the embryo cortex, was comparable between embryos developing at 22, 25, 28, and 29 °C (Fig. 3A–E, top panels), indicating that the pre-blastoderm stage spreading of nuclei along the AP axis[22] was comparable between these temperatures. During the syncytial blastoderm stage and cellularization, however, embryos that developed at 29 °C much more frequently "lost" nuclei from their cortical layer than those developing at 22, 25, and 28 °C, i.e., nuclei that were initially positioned within the cortical layer of the egg, appeared to get expelled, descended into the yolk, and did not become part of the embryonic blastoderm (Fig. 3, also see below), with variability in the extent of the nuclear loss, producing blastoderm holes in a fraction of embryos (Fig. S1).

Taken together, in pre-gastrulation embryonic development, we narrowed down the developmental vulnerability at 29 °C to the syncytial blastoderm stage and cellularization, where we see nuclear loss from the embryo cortex. Notably, the fraction of embryos showing blastoderm holes (Figs. 2A and S1) was similar to the fraction of embryos with gastrulation defects (Fig. 1F) and was consistent with embryo lethality at 29 °C (Fig. 1B). The embryo-to-embryo variability, both in the distribution and intensity of nuclear loss, further supports the argument that these defects were not dictated by defective embryonic patterning, and were likely caused by defects in non-

deterministic self-organised processes. While consistent with the idea that the presence of blastoderm holes was causing the observed increase in embryonic lethality, the temperature-dependent mechanism leading to blastoderm holes was still unclear. To further address the mechanistic link between elevated temperature and embryo lethality, we decided to quantify the loss of nuclei and possible patterns that could be predictive of blastoderm holes.

## At elevated temperature, embryos lose nuclei from the cortex after defective divisions

Closer analyses of lost cortical nuclei indicated that these nuclei were lost either individually or as pairs of two adjacent nuclei. To identify their relative contribution to the early developmental vulnerability at elevated temperatures, we decided to quantify these two types of nuclear loss from the embryo cortex.

Our analyses first focused on the isolated loss of nuclei. An isolated nucleus could be lost by a division perpendicular to the embryo surface (out-of-plane division), where the nucleus facing the inside of the embryo may not get properly anchored to the cortex and is consequently lost into the yolk. Alternatively, an isolated nucleus may be lost after an in-plane division, e.g., if steric hindrance and insufficient space would not allow for simultaneous anchoring of two adjacent nuclei to the embryo cortex. To account for these possibilities, we visually tracked every individually lost nucleus back in time to its last division.

Out-of-plane divisions could be detected in embryos at all of our experimental temperatures. These out-of-plane divisions could be easily identified, as they formed a characteristic mitotic spindle that was oriented perpendicular to the embryo surface (Fig. 4A), and after the division only one of the daughter nuclei remained attached to the cortex. However, out-of-plane divisions occurred only during the mitoses of the 10th nuclear cycle, thus couldn't explain the loss of nuclei from the 11th nuclear cycle onwards. Loss of nuclei following an in-plane division could also be detected. These losses occurred at a low frequency throughout the syncytial blastoderm stage and cellularization, at all tested temperatures (Fig. 4B). When we quantified the isolated loss of nuclei, we noted that (i) the overall frequency of out-of-plane divisions was very low (< 5%) (Fig. 4C), (ii) loss of individual nuclei after an in-plane division was rather infrequent, and, even if slightly more frequent during interphase 14 (cellularization), (iii) the loss of individual nuclei did not differ significantly within the range of experimental temperatures (Fig. 4D). The overall extremely low frequency of isolated nuclear loss−even in embryos developing at 29 °C− strongly suggests that isolated nuclear loss was not the main driver of temperature-dependent nuclear loss in the fly blastoderm.

Next, we focused on the pairwise losses of nuclei. As above, we tracked lost pairs of nuclei to their last division cycle. This tracking revealed that every pairwise loss of nuclei was, in fact, a loss of sister nuclei. This observation suggested that specific characteristics of cortical nuclear divisions might determine the future fate of the

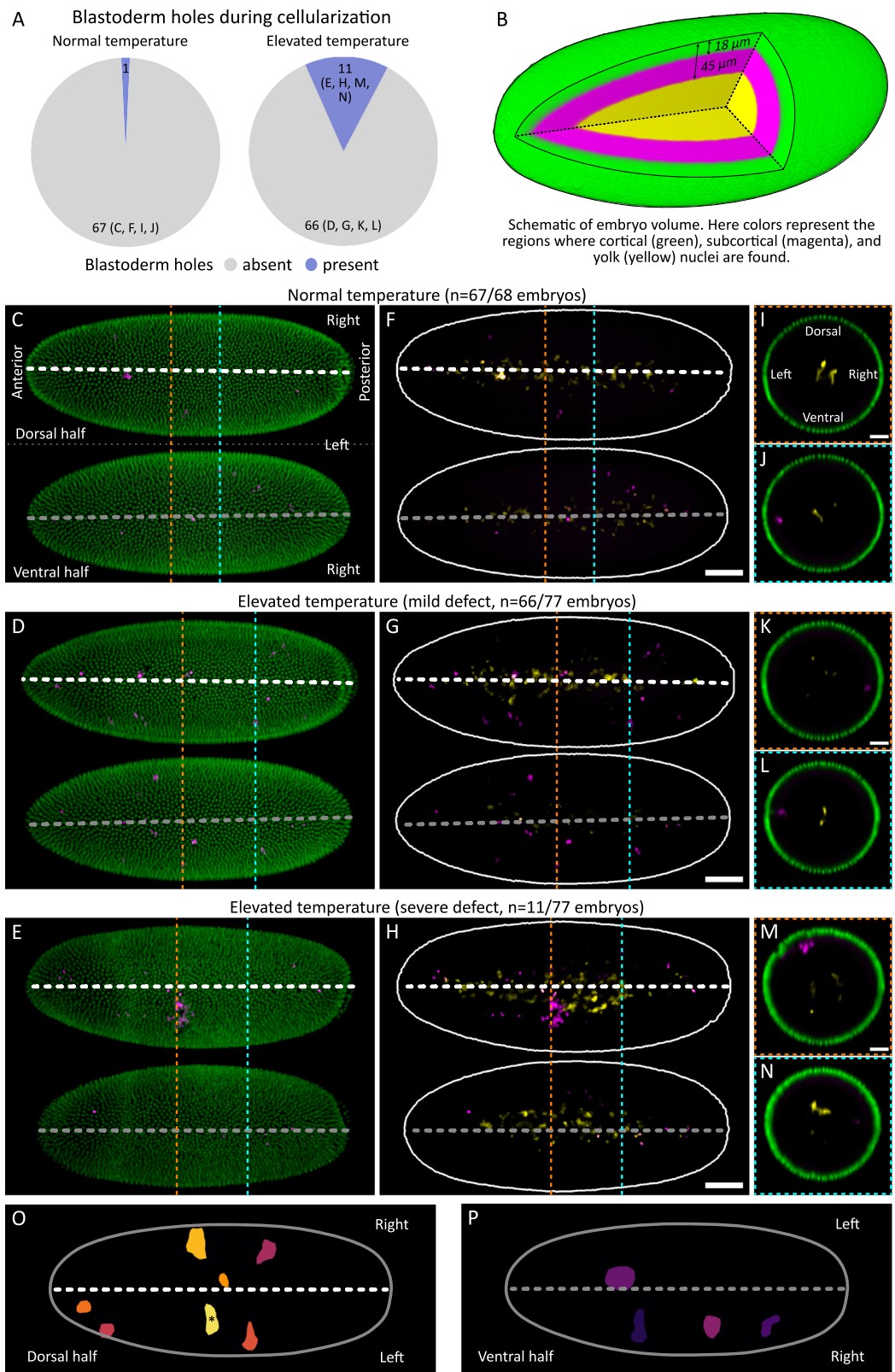

**A** Blastoderm holes during cellularization

Normal temperature

Elevated temperature

11
(E, H, M, N)

67 (C, F, I, J)

66 (D, G, K, L)

Blastoderm holes ⬤ absent ⬤ present

**B**

18 μm
45 μm

Schematic of embryo volume. Here colors represent the regions where cortical (green), subcortical (magenta), and yolk (yellow) nuclei are found.

Normal temperature (n=67/68 embryos)

**C** Right / Anterior / Posterior / Dorsal half / Left / Ventral half / Right

**F**

**I** Dorsal / Left / Right / Ventral

**J**

Elevated temperature (mild defect, n=66/77 embryos)

**D** **G** **K** **L**

Elevated temperature (severe defect, n=11/77 embryos)

**E** **H** **M** **N**

**O** Right / Dorsal half / Left

**P** Left / Ventral half / Right

newborn nuclei and trigger their release from the egg cortex, possibly in response to failed checkpoints. Consistent with this idea, we observed chromosome segregation defects and a delay in telophase, preceding the pairwise loss of nuclei (Fig. 4E), at all of our experimental temperatures. Because chromosome segregation defects can lead to aneuploidy, our observations suggested that the subsequent expulsion of daughter nuclei from the egg cortex was due to prior defective

mitosis. To address whether the observed loss of nuclei at 29 °C could be attributed to an overall increase in defective mitoses, we asked if their frequency was changing depending on the temperature (see "Methods"). For this, we focused on interphase 14, which is when we observed the most nuclear losses. While the frequency of mitotic failures was overall low at 22–28 °C, it increased significantly at 29 °C (Fig. 4F, G). Of note, defective mitoses, both at normal or at elevated

**Fig. 2 | Nuclear loss lacks left-right symmetry in embryos developing at elevated temperature. A** Pie charts show occurrence of blastoderm holes during cellularization in fixed embryos at normal temperature (*n* = 68 embryos) or elevated temperature (*n* = 77 embryos): blastoderm hole, a cluster of >10 subcortical nuclei with a corresponding discontinuity in the cortical layer. **B** Schematic shows various layers of the embryo volume: green, cortical nuclei, <18 μm from embryo surface; magenta, sub-cortical nuclei, >18 μm and <45 μm from embryo surface; yellow, yolk nuclei (vitellophages), >45 μm from embryo surface. **C–H** Distribution of nuclei in embryos during mid-cellularization (interphase 14), in fixed embryos that were developing at normal temperature (**C**, **F**) or elevated temperature that shows either a mild defect (**D**, **G**) or a severe defect (**E**, **H**). Top/bottom panels, maximum intensity projections in dorsal/ventral halves of the embryos; white/grey dashed lines, midlines drawn based on the egg shape. The voxels are color-coded as in **B**. Embryo orientation is consistently presented in all panels and is indicated in **C**.

**C–E** Cortical and subcortical nuclei are shown. Embryos at normal temperature (**C**) show rarely any gaps, while those at elevated temperature occasionally show small gaps (**D**) or blastoderm holes (**E**), indicating discontinuities in the distribution of cortical nuclei, corresponding also with the locations of subcortical nuclei. **F–H** The corresponding subcortical nuclei are shown separately for improved visibility, along with yolk nuclei. **I–N** Cross-sectional views long the orange and cyan dashed lines in **C–H**. Embryo orientation is consistently presented in all panels and is indicated in **I**. The voxels are color-coded as in **B**. **O**, **P** Blastoderm holes from 11 embryos at elevated temperature were mapped onto a model embryo surface (one blastoderm hole per embryo). Images show projections in dorsal (**O**) and ventral (**P**) halves of the model embryo. White/grey dashed lines, midlines drawn based on the egg shape; *, blastoderm hole from embryo in **E/H**. Scale bars: **F–H**, 50 μm; **I**, **K**, **M**, 25 μm.

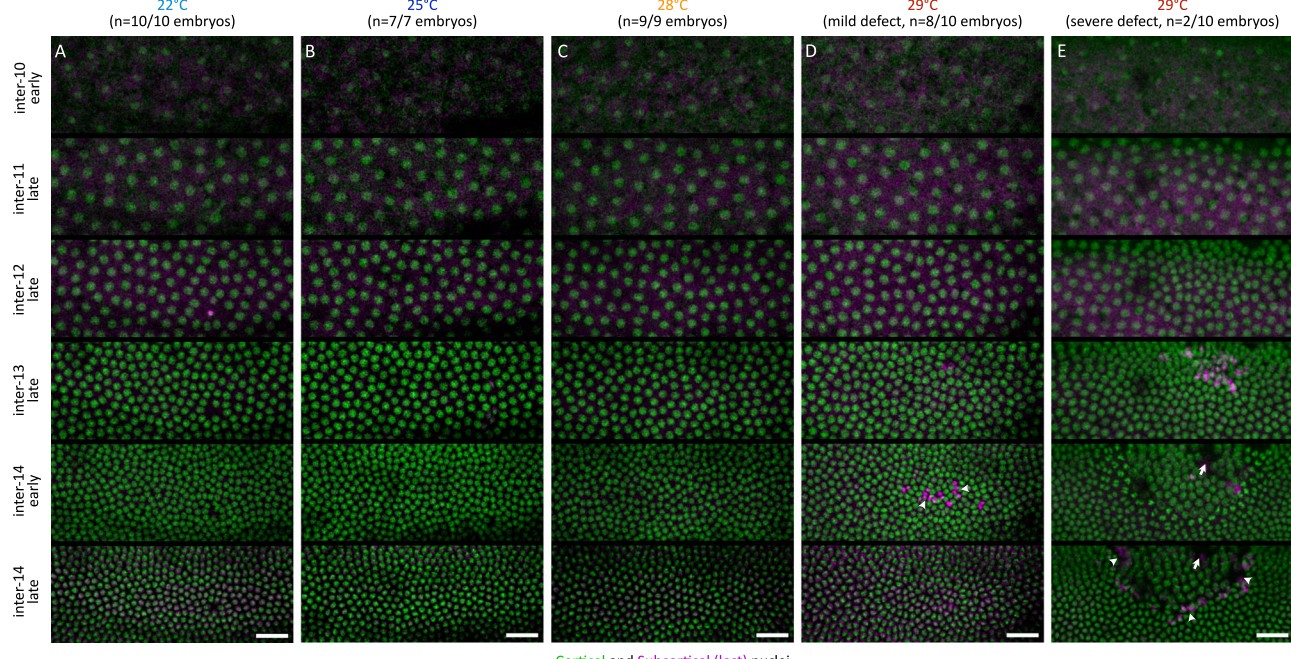

Cortical and Subcortical (lost) nuclei

**Fig. 3 | Cortical nuclei are expelled in embryos at elevated temperature. A–E** Montages showing live embryos, expressing Histone-RFP, from early interphase 10 till late interphase 14, developing either at 22 °C (**A**), 25 °C (**B**), 28 °C (**C**), or at 29 °C, showing mild (**D**), or severe (**E**) defect. The nuclei were labelled as a function of their distance from the embryo surface. Green, cortical nuclei; magenta, sub-cortical nuclei that are lost (see also schematic in Fig. 4B). At elevated temperature, a lot more nuclei are expelled, and as a result, we can see gaps between cortical nuclei, coinciding with subcortical nuclei: arrowheads, small gaps; arrows, blastoderm hole. Scale bars, 25 μm.

temperature, were not associated with a stereotypical time course of nuclear expulsion. Defective nuclei could be lost in the interphase immediately following defective divisions, or later (Fig. S2), suggesting that nuclear expulsion after defective mitosis was a passive process and not triggered by molecular signaling at failed checkpoints.

Taken together, our findings identify nuclear divisions in the blastoderm as a thermolabile cellular process that shows a disproportionate increase in failures with the transition from 28 to 29 °C, resulting in the loss of sister nuclei from the egg cortex, when born in defective mitoses. However, which characteristics of nuclear divisions render them thermolabile was unclear.

### At elevated temperature, nuclei displace drastically, especially during divisions

For embryos at 25 °C, we noted that the nuclei were rather dynamic in their localization, showing local movements throughout the syncytial blastoderm stage. For embryos at 29 °C, this dynamicity increased, and we could see nuclei being displaced rather drastically. We hypothesized that such displacement poses a challenge to

nuclear anchorage at elevated temperatures, especially during mitosis, causing mitotic defects and the paired loss of nuclei. To that end, we tracked nuclear groups starting from individual mother nuclei at end of interphase 10 to the 8 great-granddaughter nuclei at end of interphase 13. At each time point, we computed the center of the nuclear group, using the positions of the nuclei in that group, yielding one track per group (Fig. 5A–C, also see "Methods"). Our tracking showed that the centers of nuclear groups had two components to their movement, a global drift and a local displacement.

First, the nuclear groups exhibited a global consistent drift-like movement, which represented a concerted global movement of all nuclei. We captured this component using a linear fit to the tracks of the center of the nuclear groups over time (Fig. 5B dotted lines). The slope of the linear fit indicated that the groups drifted towards the posterior end at all of our experimental temperatures (Fig. 5D, E). This component of nuclear movements was reminiscent of the previously reported dynamic reordering of nuclei in the early syncytial blastoderm[23]. Notably, however, the extent of drift in nuclear groups

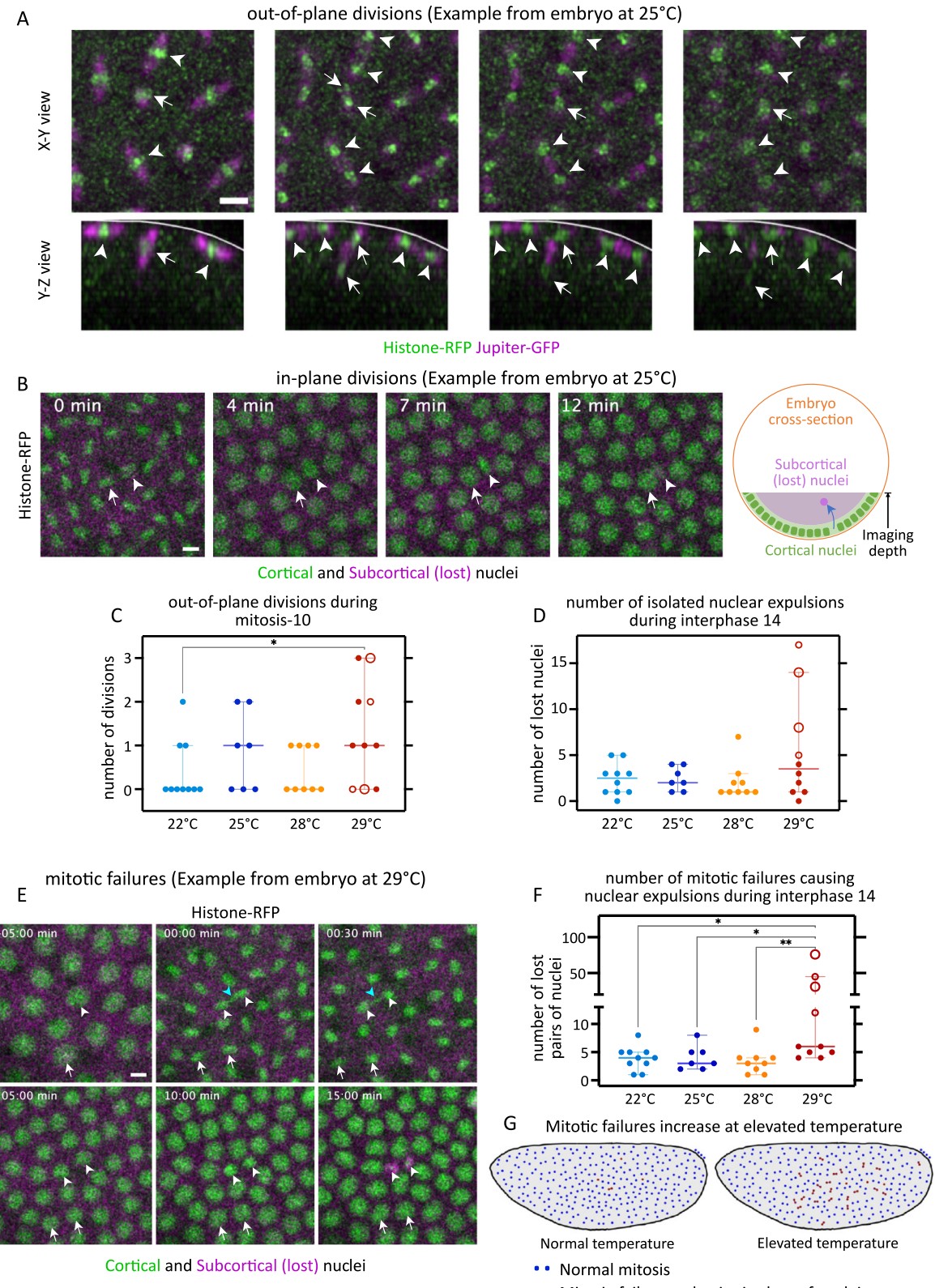

did not correlate with the occurrence of blastoderm holes (Fig. 5D, large hollow markers).

Second, independent of this drift, we observed local deviations in the position of the nuclear groups over time. We decided to measure these local deviations relative to the global drift. We reasoned these local deviations could be a proxy for mechanical stresses experienced by the nuclear division machinery, and quantified these deviations during interphase-metaphase and anaphase-telophase. The deviations relative to the constant drift were more drastic along the anterior-posterior (AP) axis (Fig. 5F, H) than those along the dorsal-ventral (DV) axis (Fig. 5G, I), and deviations during the anaphase-telophase period (Fig. 5H) were more drastic than those during the interphase-metaphase period (Fig. 5F). In the context of length scales typical for nuclear divisions, this meant that nuclear group deviations in embryos

**Fig. 4 | Nuclear loss as isolated or paired expulsions at elevated temperature.** **A**–**D** Following isolated nuclear expulsions back in time, we found them to stem from both out-of-plane and in-plane divisions. **A** Montages exemplify out-of-plane division during mitosis 10 at 25 °C. Histone-RFP marks nuclei (green), Jupiter-GFP marks microtubules (magenta). Arrowheads, in-plane divisions; arrows, out-of-plane division; white lines, embryo surface. Each subsequent frame is 1 min apart. **B** Montage exemplifies in-plane divisions at 25 °C, from telophase 12 to interphase 13. Histone-RFP marks nuclei: cortical nuclei, green; subcortical nuclei, magenta; see schematic. The arrowheads/arrows track the daughter nucleus that will be expelled/cortically retained. Inset, time since telophase 12. **C**, **D** Scatter plots show number of out-of-plane divisions during mitosis 10 (**C**) and number of isolated nuclear expulsions (D) during interphase 14. **E**–**G** To assess the pairwise loss of nuclei, we counted mitotic failures. **E** Montage exemplifies mitotic failure at 29 °C,

from late interphase 12 to late interphase 13. Histone-RFP marks nuclei: cortical nuclei, green; subcortical (lost) nuclei, magenta; see the schematic in **B**. White arrows, normal nuclear division; white arrowheads, mitotic failure; cyan arrowheads, lagging chromosomes. Inset, time since telophase 12. **F** Scatter plot shows number of mitotic failures during interphase 14. **G** The schematic illustrates the phenomenon of mitotic failures at elevated temperature. Scale bars: **A**, 10 μm; **B** and **E**, 5 μm. **C**, **D**, and **F** show scatters with median, with whiskers showing 95% CI. Large hollow markers, embryos with blastoderm holes; small hollow markers, embryos with small gaps in the blastoderm. $n = 10$ (22 °C), 7 (25 °C), 9 (28 °C), and 10 (29 °C) embryos. Non-parametric Kruskal–Wallis test with Dunn's uncorrected test for multiple comparisons: *, $p < 0.05$; **, $p < 0.01$; non-significant comparisons are not shown. Source data and exact $p$-values for plots are available in the Source Data file.

developing at 25 °C and during anaphase-telophase were comparable to that of the length of the spindle itself, while for embryos at 29 °C, nuclear group deviations could be up to fourfold higher. This increase in nuclear group deviations suggested increased mechanical stresses, especially during the critical anaphase-telophase period, indicating an extensive propensity for displacement relative to embryo surface at a time when the dividing nuclei need to be robustly anchored to the embryo cortex. Consistent with this interpretation, embryos with blastoderm holes showed the highest nuclear group deviations, especially during anaphase-telophase (Fig. 5H, large hollow markers).

To further test the link between mitotic failures and the short-range local nuclear displacements, i.e., nuclear group deviations, we followed the division of each of the 8 great-granddaughter nuclei in every tracked nuclear group at 29 °C from mitosis 13 to interphase 14, expecting to see mitotic failures in nuclear groups with greater displacements. Here, however, we failed to see a correlation between extent of nuclear group deviations and a subsequent mitotic failure (Fig. 5J). In that, the nuclear groups with one or more subsequent mitotic failures (Fig. 5J, hollow markers) do not show particularly higher deviations compared to their normally dividing counterparts. In fact, when we count the number of nuclear groups with subsequent mitotic failure defects, we see that 15 of these groups were below their respective embryo medians, while only 7 are above, arguing against a correlation between increased nuclear displacements and subsequent mitotic failures during interphase 14.

In addition to the analysis of nuclear movements parallel to the embryo cortex, we wondered about the movements orthogonal to cortex, i.e., the future apical-basal (AB) axis in the blastoderm cells[24]. In embryos developing at 25 °C, we observed that nuclei maintained a constant distance from the cortex (Fig. S3A). However, in embryos developing at 29 °C, we could see two waves of nuclear displacement towards the cortex, which originate from the anterior and the posterior, and meet at the central region of the embryo (Fig. S3B). These waves of nuclear displacement were tightly correlated with waves of nuclear divisions, with peaks of the waves corresponding to anaphase-telophase transition. This suggests that as nuclei divide, the spindle moves upwards, presumably to gain more space. This need for space indicated additional challenges for the spindles during nuclear division, explaining, at least in parts, the mitotic failure at 29 °C.

Taken together, our quantifications and observations captured an embryo-wide increase in nuclear displacement locally, along the AP- and AB-axis. In a fraction of embryos, the mitotic failures occurred in great numbers and in close vicinity, leading to holes in the blastoderm epithelium at 29 °C (Fig. 4F, large hollow markers), and these embryos also exhibited greater displacement of nuclei during anaphase-telophase (Fig. 5H, large hollow markers). Notably, however, our quantifications and observations of an increased nuclear displacement could not reveal any correlation with nuclear loss at the level of individual nuclear groups (Fig. 5J). This result argues against the idea of increased dynamicity being the cause of nuclear division defects at

29 °C. Thus, although the formation of blastoderm holes (Fig. S1) could be directly attributed to clustered mitotic failures, it remained unclear why mitotic failures were stochastically distributed in most embryos, yet locally clustered in the central region of a small fraction of embryos, producing blastoderm holes.

## Nuclei are crowded and divide with a time shift in embryos at elevated temperature

To better understand the differences between stochastic mitotic failures and locally clustered failures, we focused on cases with severe nuclear losses in embryos at 29 °C. The most severe loss of nuclei in embryos at 29 °C was often observed during cellularization (interphase 14) in the central domain of the embryo (Figs. 2, 3, and S1). Thus, we wondered if the central regions of the embryo were somehow special, and if there were characteristics in nuclear distribution or nuclear division, already during nuclear cycle 13, that rendered the central regions to be distinct than the anterior/posterior poles.

The central region of the embryo had more tightly packed nuclei, indicating a defect in self-organized nuclear reordering mechanisms[23]. To quantify this nuclear crowding, we focused on the end of interphase 13 (see "Methods"), as that's the time we observed maximum nuclear crowding, before the most intense nuclear loss. To quantify the space available for cortical blastoderm nuclei, we segmented the nuclei and used Voronoi tessellation to define a region around each nucleus, which we call pseudo-cells. The areas of pseudo-cells were rather homogeneous in embryos developing at 22 °C (Fig. 6A, E, I). Already at 25 °C (Fig. 6B, F, J) and 28 °C (Fig. 6C, G, K), the available space for cortical nuclei was no longer uniform across the embryo, and it showed signs of local crowding in the central regions of the embryo. At 29 °C, the space available for nuclei along the anterior-posterior axis varied substantially and revealed strong nuclear crowding in the central region of the embryo (Fig. 6D, H, L). We quantified the extent of crowding (see "Methods") and found a progressive increase in nuclear crowding (Fig. 6M), with peak crowding at 29 °C. To specifically test whether the central region of the embryo was distinct from the poles in inter-nuclear spacing, we quantitatively described elevated nuclear crowding at 29 °C in the central region of the embryo (Fig. 6N).

The central regions of the embryos at 29 °C coincided with the "collision" of two waves of nuclear divisions starting from the poles (see above, Fig. S3). This observations was in stark contrast with embryos at 25 °C, where nuclear divisions are meta-synchronous[25], with anterior-posterior waves (Fig. 6O). We could further observe that in contrast to the meta-synchrony at 25 °C, phases of nuclear division in the central region at 29 °C were lagging behind those at the anterior/posterior pole of the embryos (Fig. 6P). Accordingly, the same phase of the nuclear division cycle could be observed at different time points in different regions of the embryo (Fig. S4A, B), indicating defects in the self-organised coordination of nuclear

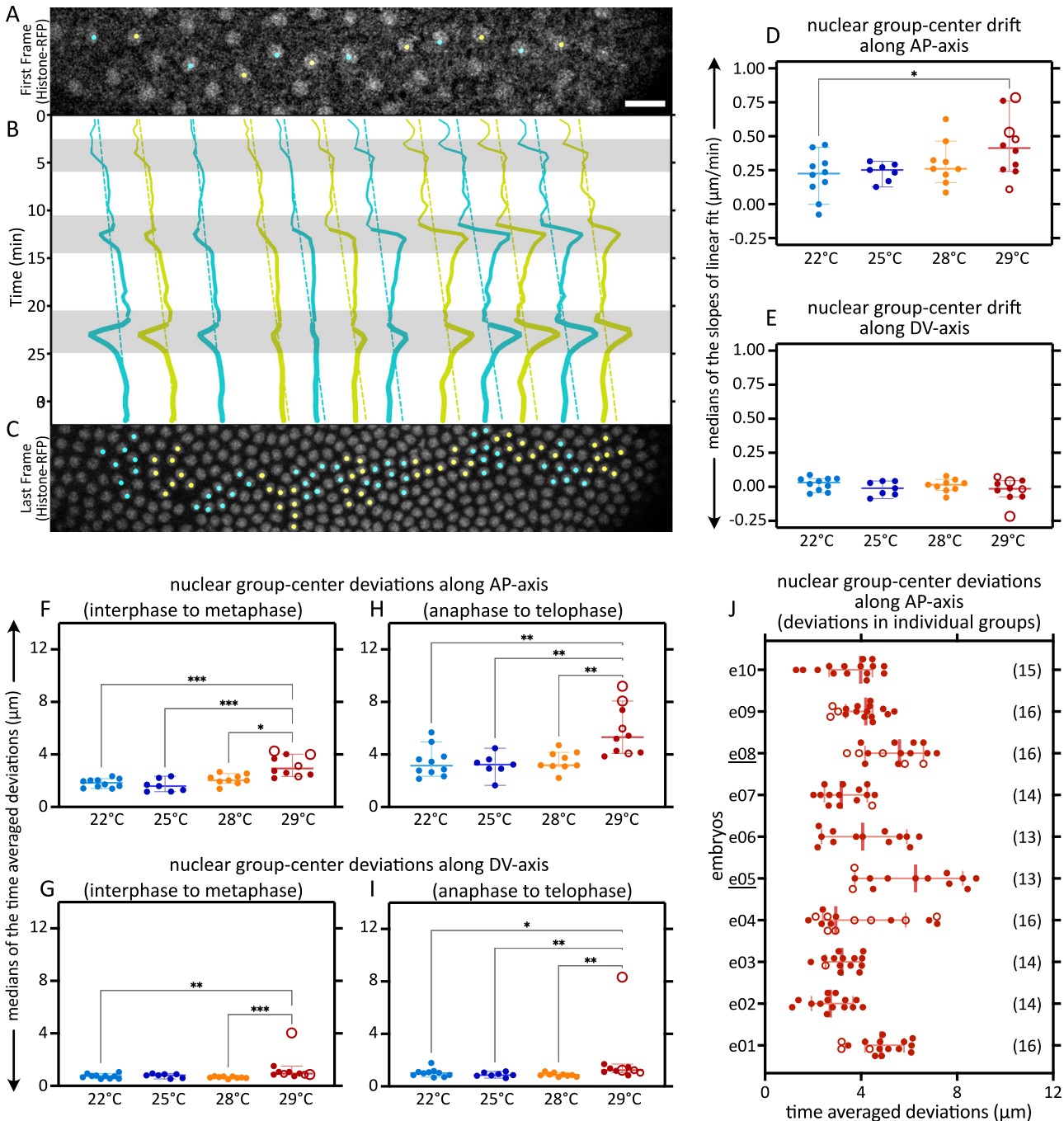

**Fig. 5 | Nuclear movement parallel to the embryo surface is amplified at elevated temperature. A–C** An example of nuclear movement tracking. Snapshots from embryo at 29 °C, with Histone-RFP marking nuclei, showing the mother nuclei at the end of interphase 10 (**A**) and the corresponding 8 great-granddaughter nuclei, each, at the end of interphase 13 (**C**), connected via the group-center trajectories (**B**). Cyan and yellow, neighbouring nuclear groups; thickening solid lines, trajectories of group-centers with increasing number of nuclei due to divisions; dotted lines, linear fits to group-center trajectories; shaded region, periods of greatest movement (see "Methods"). **D–I** To quantify the drift in the nuclear group centers, we compute the slope of the linear fits (dotted lines in **B**). Plots show the distribution of embryo medians, for the drift along the AP axis (**D**) and DV axis (**E**). To quantify the movements in nuclear group centers during interphase-metaphase vs. anaphase-telophase (shaded vs. unshaded region in **B**), we compute the average of the deviations from the linear fits. Plots show the distribution of embryo medians, for the fluctuations along the AP axis (**F, H**) and DV axis (**G, I**), during interphase-metaphase (**F, G**) and anaphase-telophase (**H, I**). Large hollow markers, embryos with blastoderm holes; small hollow markers, embryos with small gaps in the blastoderm. $n = 10$ (22 °C), 7 (25 °C), 9 (28 °C), and 10 (29 °C) embryos. **J** To test whether nuclear groups with highest movement also correspond to mitotic failures, we plot the deviations along AP-axis, now showing all of the tracked nuclear groups at 29 °C. Hollow markers, nuclear groups with at least one subsequent mitotic failure; underlined, embryo with blastoderm holes; insets, number of tracked nuclear groups. Scale bar, 25 μm. **D–J** show scatters with median, with whiskers showing 95% CI. Non-parametric Kruskal–Wallis test with Dunn's uncorrected test for multiple comparisons: *, $p < 0.05$; **, $p < 0.01$; ***, $p < 0.001$; non-significant comparisons are not shown, statistical tests not performed on data in **J**. Source data and exact $p$-values for plots are available in the Source Data file.

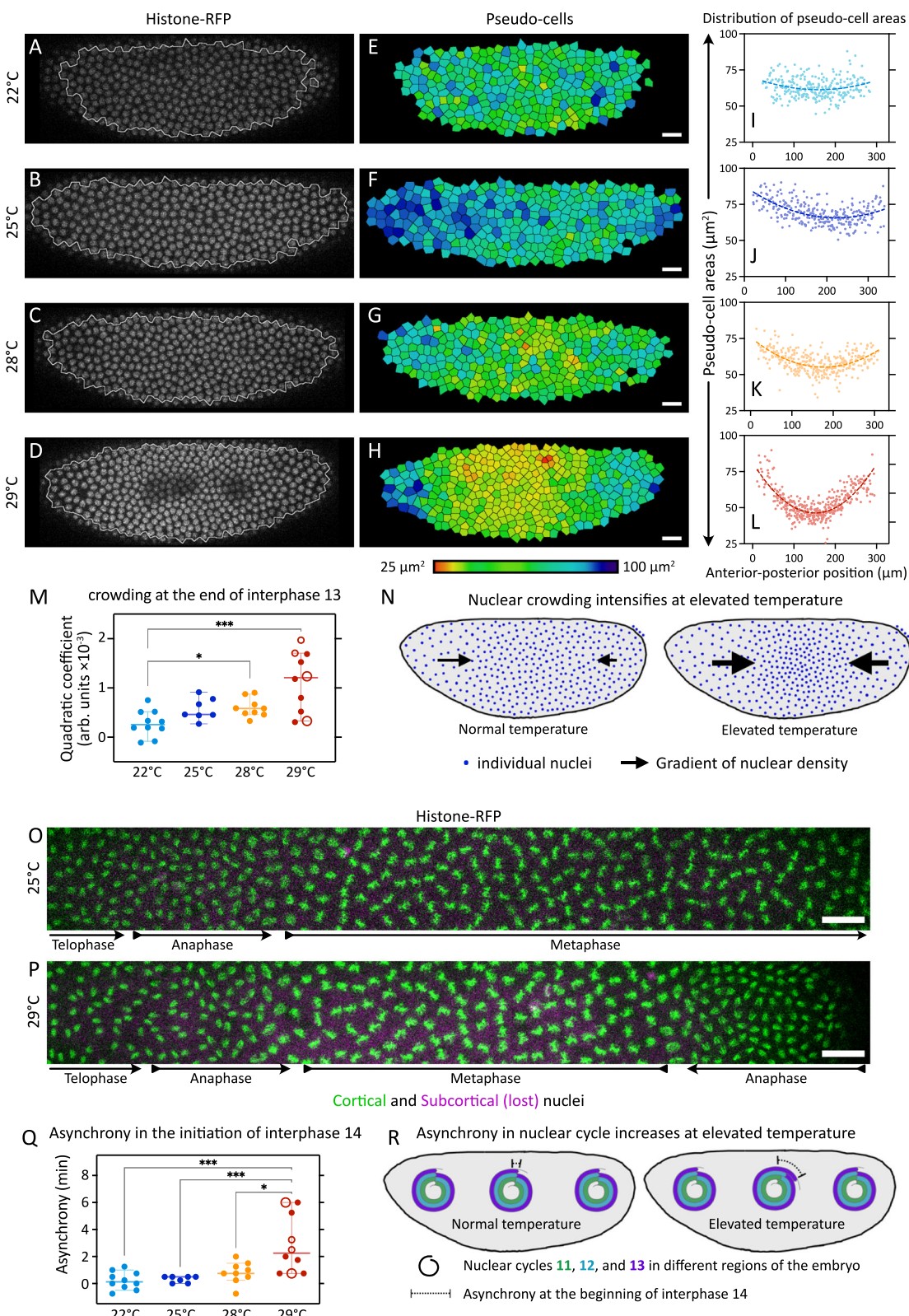

**M** crowding at the end of interphase 13

**N** Nuclear crowding intensifies at elevated temperature

Normal temperature — Elevated temperature

• individual nuclei    → Gradient of nuclear density

Histone-RFP

**O** 25°C — Telophase → Anaphase → Metaphase

**P** 29°C — Telophase → Anaphase → Metaphase → Anaphase

Cortical and Subcortical (lost) nuclei

**Q** Asynchrony in the initiation of interphase 14

**R** Asynchrony in nuclear cycle increases at elevated temperature

Normal temperature — Elevated temperature

◯ Nuclear cycles **11**, **12**, and **13** in different regions of the embryo

├┄┄┄┤ Asynchrony at the beginning of interphase 14

division cycles across the syncytium[25]. Thus, we decided to quantify this asynchrony of nuclear division cycles at temperatures 22, 25, 28, and 29 °C, as nuclei exited nuclear cycle 13 and entered cycle 14 (see "Methods"). Our quantifications show that the asynchrony significantly increases in embryos developing at 29 °C as compared to other temperatures (Fig. 6Q). This indicated that the central region

of the embryo is distinct from the poles in nuclear division dynamics, with the phases of the nuclear cycle lagging (Fig. 6R).

Taken together, in the central region of the *Drosophila* syncytial blastoderm embryo, nuclear division cycles lag behind those at the anterior/posterior poles, and nuclear density is greater than that at the anterior/posterior poles. Given that blastoderm holes also appear

**Fig. 6 | Nuclear crowding and Nuclear cycle asynchrony increase at elevated temperature. A–N** To assess the crowding of nuclei, we segmented the nuclei and generated pseudo-cells. **A–D** Snapshots from embryos at the end of interphase 13 at indicated temperatures: Histone-RFP marks the nuclei; outline, region of the image considered for quantifications. **E–H** The pseudo-cells for the corresponding embryos in **A–D**. Color indicates pseudo-cell area in $\mu m^2$, LUT at the bottom. **I–L** Plots show distribution of pseudo-cell areas as a function of AP position embryos in **A–D**. Dotted line, quadratic fit. **M** Scatter plots show distribution of the coefficient of the quadratic term, with a larger quadratic coefficient indicating greater crowding. **N** The schematic illustrates the crowding effect at elevated temperature: size of the black arrows, gradation in nuclear density. **O–R** To assess the asynchrony in nuclear cycles, we identified the phases of the nuclear cycle along the AP axis. **O, P** Image shows the entire anterior-posterior extent of the field of view, from embryos at 25 °C (**O**) and 29 °C (**P**). Histone-RFP marks the nuclei: green, cortical nuclei; magenta, subcortical (lost) nuclei (see also schematic in Fig. 4B). Arrows show the direction in which the wave of the nuclear division cycle will progress. At 25 °C, we see a single wave from the anterior, while at 29 °C we see an anterior- and a posterior-wave. **Q** Scatter plots show asynchrony in initiation of interphase 14. **R** The schematic illustrates the progressive increase in asynchrony at elevated temperature. Scale bars, 20 μm. **M** and **Q** shows scatters with median, with whiskers showing 95% CI. Large hollow markers, embryos with blastoderm holes; small hollow markers, embryos with small gaps in the blastoderm. $n = 10$ (22 °C), 7 (25 °C), 9 (28 °C), and 10 (29 °C) embryos. Non-parametric Kruskal–Wallis test with Dunn's uncorrected test for multiple comparisons: *, $p < 0.05$; **, $p < 0.01$; ***, $p < 0.001$; non-significant comparisons are not shown. Source data and exact $p$-values for plots are available in the Source Data file.

in the central regions of the embryo, we tested whether the clustering of mitotic failures (leading to blastoderm holes) at 29 °C correlated better with the regions of highest nuclear crowding, or the most delayed nuclear cycles, or both.

## Region of embryo with greater nuclear crowding and with delayed nuclear cycles are associated with greater frequency of mitotic failures

To test whether nuclear crowding and/or nuclear cycle asynchrony resulted in a predisposition towards a local increase in mitotic failure, we first defined the regions of the embryo with high nuclear crowding (Fig. 7A) at the end of interphase 13, or regions with substantially delayed nuclear division cycles (Fig. 7B) at the end of mitosis 13. In parallel, we marked the locations of mitotic failures occurring during cellularization (interphase 14), which could be overlaid on any pre-defined region to calculate the fraction of mitotic failures in that region. Then, to measure the fraction of mitotic failures associated with either nuclear crowding, or nuclear cycle asynchrony, or both, we constructed the regions of the embryo with either high nuclear crowding, i.e., crowded regions (Fig. 7C); or substantial delay in nuclear division cycles, i.e., delayed regions (Fig. 7D); or a combination of crowded and delayed regions, i.e., combined region (Fig. 7E) (see "Methods").

In crowded regions, the fraction of mitotic failures increased progressively with higher temperatures, with statistically significant differences between 22 and 29 °C (Fig. 7F). In delayed regions, the fraction of mitotic failures showed an abrupt increase from 28 to 29 °C, albeit with only marginal statistical significance (Fig. 7G). In the combined regions, the fraction of mitotic failures was greater than those in the crowded or delayed regions alone (Fig. 7H). Additionally, the fraction of mitotic failures in all of the tested temperatures could not be explained by the area fractions themselves (Fig. 7I–K). This suggests that the combination of nuclear crowding and nuclear cycle asynchrony contributes to a temperature-dependent predisposition toward mitotic failure in the syncytial blastoderm, with nuclear crowding exerting the stronger influence.

One plausible mechanism is that local nuclear crowding limits the space for proper mitotic spindle formation, thereby increasing the likelihood of mitotic errors. In addition, crowding may locally elevate the Nuclear-to-Cytoplasmic (N/C) ratio, a parameter that reflects the amount of cytoplasm available to each nucleus. The N/C ratio is known to regulate the pace of nuclear division cycles, such that higher N/C ratios are associated with slower nuclear division cycle[25–30], likely due to limited cytoplasmic resources required for the cell cycle. Nuclear crowding would therefore slow nuclear cycle indirectly through its effect on N/C ratio. Consistent with this interpretation, the reduced area of the "combined regions" at 29 °C (Fig. 7K) indicates substantial spatial overlap between crowded regions and delayed regions, particularly in embryos exhibiting blastoderm holes. Thus, at elevated temperature, these combined regions likely reflect both the direct mechanical consequences of crowding and its indirect effects mediated through altered nuclear division timing.

Taken together, the spatial overlap between nuclear crowding and nuclear cycle asynchrony helps explain why clustered mitotic failures arise preferentially in the central region of the embryo, leading to blastoderm holes. However, while this framework clarifies where and how mitotic failures are spatially organized, it does not explain why elevated temperature enhances their frequency. This raises the question of how elevated temperature perturbs mitosis at the molecular level.

## Mitotic failures trigger DNA damage response

To distinguish whether the observed loss of nuclei were primarily caused by temperature-induced DNA damage leading to mitotic failure, or instead by a temperature-induced failure of mitosis that subsequently resulted in damaged DNA, we employed a previously described DNA damage reporter system[31]. Briefly, this reporter relies on the nuclear localization of Stem-Loop Binding Protein (SLBP), which normally shuttles between nucleus and cytoplasm during interphase, exporting the mRNAs for key early zygotic genes[31]. Following DNA damage, Chk2 phosphorylates SLPB at Serine 118, thus restricts nuclear localization of SLBP, targeting the protein to degradation. Thus, exclusion of SLBP from the interphase nucleus can act as a live marker for DNA damage in that nucleus (Fig. 8A), provided that the nuclear envelope remains intact.

To determine the cause for the cortical loss of nuclei, we used the absence of SLBP nuclear accumulation as an indicator for DNA damage and followed such nuclei over a nuclear division cycle. All nuclei displaying evidence of DNA damage were mitotically arrested and were subsequently lost into the interior of the embryo (Fig. 8B). DNA damage therefore consistently coincided with mitotic failures and, just like the mitotic failures, affected sister nuclei.

To quantify the kinetics of SLBP nuclear localization over the different phases of the nuclear division cycle, we measured SLBP-GFP nuclear intensity from late interphase 13 through early interphase 14 (see "Methods"). SLBP showed high nuclear localization at the end of interphase 13, was then excluded from the nuclei during mitosis, and progressively re-accumulated in nuclei at the onset of interphase 14 (Fig. 8C, F). This behavior was comparable between 25 and 29 °C during normal nuclear divisions. In contrast, following mitotic failure, we observed a progressive reduction in SLBP nuclear localization within the first ~4 min of interphase 14, followed by a more pronounced loss over the next ca. 10 min (Fig. 8F), after which the nuclei were typically expelled from the forming blastoderm. The absence of differences in SLBP localization during the preceding interphase, together with indistinguishable behavior during mitosis, indicates that activation of the DNA damage response occurs specifically during interphase 14 and only after the mitotic failure.

Mitotic failure can trigger a variety of signaling responses, raising the possibility that the observed loss of SLBP nuclear localization

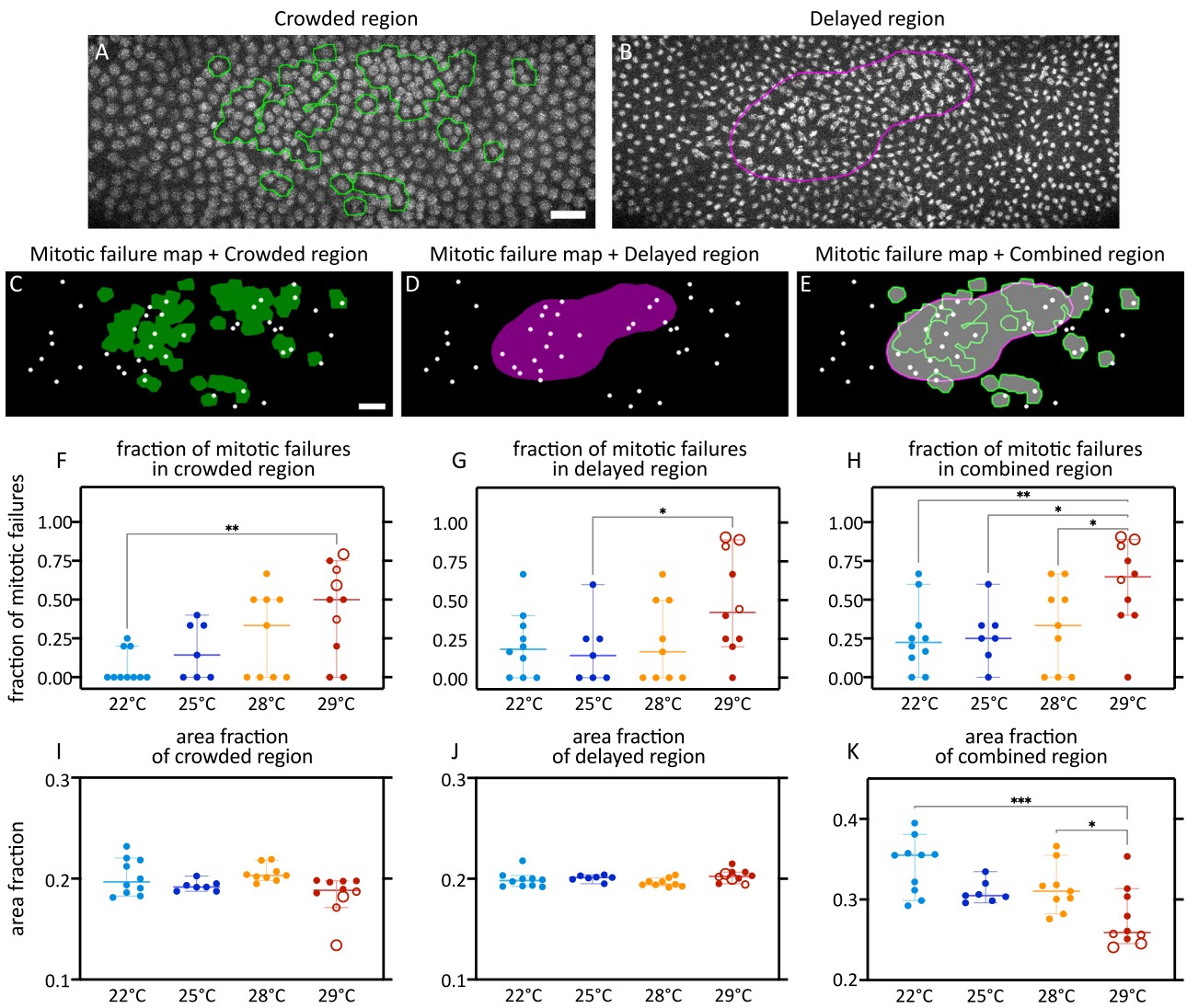

**Fig. 7 | Nuclear crowding and nuclear cycle asynchrony combinatorially enhance mitotic failures. A**, **B** Snapshots from embryo at 29 °C, at the end of interphase 13 (**A**) and 14 min afterwards (**B**). Histone-RFP marks nuclei; green outline, region of the image with nuclear crowding; magenta outline, region containing the most delayed nuclei, i.e., those that are most lagging in nuclear division cycles. **C**–**E** The locations of lost pairs of nuclei during interphase 14 (mitotic failures during mitosis 13) are marked as white dots. Then, various regions are defined in the embryo to illustrate the fraction of mitotic failures occurring in that region. These regions are composed based on, the crowded region (**C**), the delayed region (**D**), and the combined region (**E**). **F**–**K** Scatter plots show distribution of the fraction of mitotic failures occurring in the crowded (**F**), delayed (**G**), and combined (**H**)

regions of the embryos, and the distribution of the area fractions in crowded (**I**), delayed (**J**), and combined (**K**) regions of the embryos. In the case of the area fraction of the combined regions (**K**), the increase in the overlap of crowded and delayed regions reduces the overall size of the combined regions. Scale bars, 20 μm. **F**–**K** Show scatters with median, with whiskers showing 95% CI. Large hollow markers, embryos with blastoderm holes; small hollow markers, embryos with small gaps in the blastoderm. $n = 10$ (22 °C), 7 (25 °C), 9 (28 °C), and 10 (29 °C) embryos. Non-parametric Kruskal–Wallis test with Dunn's uncorrected test for multiple comparisons: *, $p < 0.05$; **, $p < 0.01$; ***, $p < 0.001$; non-significant comparisons are not shown, significance not tested for **I** and **J** (see "Methods"). Source data and exact $p$-values for plots are available in the Source Data file.

reflects a response other than the DNA damage signaling. To address this, we followed the nuclear localization of SLBP.S118A-GFP, a point mutant that cannot be phosphorylated at serine 118 and is therefore insensitive to Chk2-mediated DNA damage signaling[31]. Quantification of SLBP.S118A nuclear localization revealed behavior comparable to wildtype SLBP in normally dividing nuclei (Fig. 8D, G). In nuclei with DNA damage, however, SLBP.S118A was retained in the nucleus during the early phase of interphase 14, remaining detectable for approximately 4–5 min, before being progressively lost from the nucleus at later time points (Fig. 8G). These observations suggest that the loss of nuclear SLBP following mitotic failure comprises at least two components; an early, DNA damage-dependent component

and a later component that is independent of SLBP phosphorylation at serine 118.

DNA damage can induce nuclear envelope destabilization, resulting in the release of nuclear proteins, including SLBP. To contextualize the similarities and differences in the kinetics of SLBP and SLBP.S118A, we analyzed the behavior of a GFP reporter bearing a nuclear localization signal (NLS-GFP). In normally dividing nuclei, NLS-GFP showed nuclear localization dynamics similar to those of both SLBP and SLBP.S118A (Fig. 8E, H). In contrast, following mitotic failure, the kinetics of NLS-GFP localization closely resembled those of SLBP.S118A, but not wildtype SLBP (Fig. 8F–H). This indicates that the altered nuclear localization observed specifically during the early phase of interphase 14 reflects

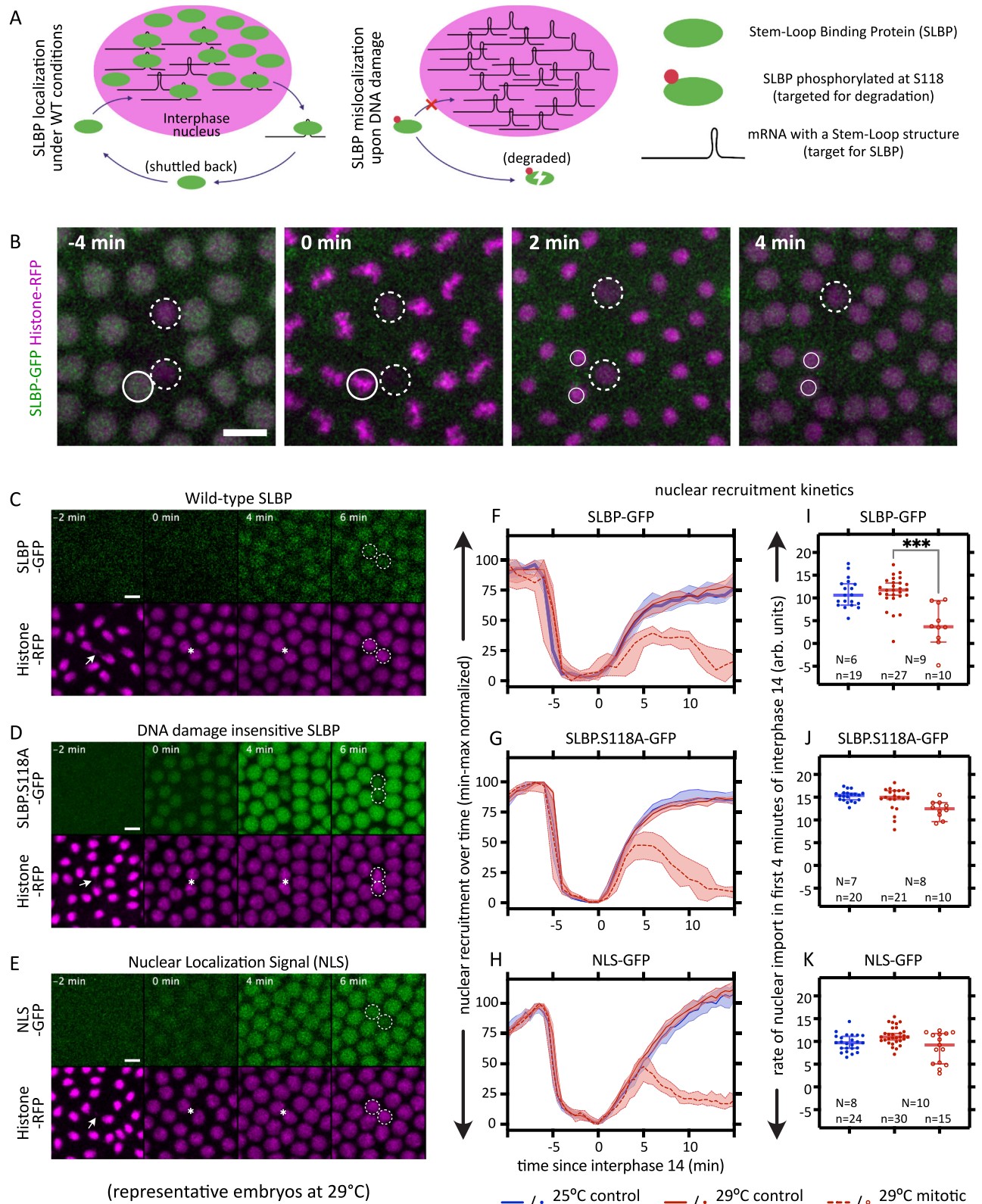

(representative embryos at 29 °C)

activation of the DNA damage response, which is captured by SLBP-GFP but absent in both SLBP.S118A-GFP and NLS-GFP.

To quantify this DNA damage-dependent effect, we compared the rates of nuclear accumulation during the initial phase of interphase 14. SLBP nuclear import was significantly slower in nuclei undergoing division defects at 29 °C compared to normally dividing nuclei at the same temperature (Fig. 8I). In contrast, nuclear import rates of both

SLBP.S118A (Fig. 8J) and NLS (Fig. 8K) were unaffected by mitotic failure during this early interphase period.

Taken together, these results indicate that DNA damage arises as a consequence of mitotic failure rather than acting as its primary cause. Moreover, the DNA damage response is rapidly engaged, becoming detectable within minutes after entry into interphase 14. This rapid onset highlights mitosis as a critical point of vulnerability and underscores the

**Fig. 8 | Mitotic failures lead to DNA damage response. A** Schematic shows how the mislocalization of the Stem-Loop Binding Protein (SLBP) can be used as a reporter for DNA damage. During interphase, SLBP normally localizes to the nuclei, while it is excluded in nuclei with DNA damage. **B** Nuclei with DNA damage are mitotically arrested. The nuclear localization of SLBP (green) allows us to distinguish nuclei with DNA damage (dashed circles) vs. normal ones (circles). Following these nuclei over a mitosis shows that nuclei with DNA damage are mitotically arrested, while the normal nuclei divide, giving rise to two daughter nuclei (smaller circles). One of the nuclei with DNA damage is lost to embryo interior by the last timepoint. Inset: time since mitosis. $n = 137/137$ nuclei with DNA damage that did not divide in nuclear cycles 11 to 13 while their neighbors did; pooled from $N = 6$ (25 °C) and 9 (29 °C) embryos. **C–K** Mitotic failure leads to DNA damage response. **C–E** Montages show representative embryos developing at 29 °C from telophase 13 till early interphase 14, in embryos expressing SLBP-GFP (**C**), SLBP.S118A-GFP (**D**), and NLS-GFP (**E**). Arrows, chromosome segregation defect; asterisks, pair of defective daughter nuclei; dashed circles, reduction in GFP-signal in the nuclei with mitotic failure, still localized cortically; inset, time since beginning of interphase 14. **F–K** Quantifications showing the kinetics of nuclear localization of SLBP-GFP (**F**), SLBP.S118A-GFP (**G**), and NLS-GFP (**H**) in control divisions in embryos at 25 °C, control division in embryos at 29 °C, and mitotic failures in embryos at 29 °C. **I–K** Slope of a straight-line fit between timepoints 0–4 min in **F–H**. Insets, the number of nuclear divisions ($n$), pooled from (N) embryos. Scale bars; **B**, 10 μm; **C–E**, 5 μm. **F–H** plots median values over time, with shaded regions showing 95% CI. **I–K** show scatters with median, with whiskers showing 95% CI. Non-parametric Kruskal–Wallis test with Dunn's uncorrected test for multiple comparisons: ***, $p < 0.001$; non-significant comparisons are not shown. Source data and exact p-values for plots are available in the Source Data file.

---

importance of understanding the mechanisms underlying mitotic failure in order to improve embryonic robustness at elevated temperatures.

## Cytoskeletal dynamics during mitosis at elevated temperature

Our results so far link the vulnerability of mitosis at elevated temperature—and its central role in the formation of blastoderm holes—with embryo lethality at elevated temperature. Given the essential roles of the cytoskeletal elements, F-actin and microtubules, in orchestrating mitosis, we next asked whether elevated temperature alters cytoskeletal dynamics during syncytial nuclear divisions. Specifically, we quantified changes in cytoskeletal organization during mitosis at 29 °C as compared to 25 °C, and examined how these dynamics differed between normally completing divisions and mitotic failures.

During syncytial divisions, cortical F-actin undergoes extensive remodeling across the nuclear division cycle, most prominently supporting the formation of the transient pseudo-cleavage furrows during mitosis[15,24,32]. These furrows progressively deepen perpendicular to the embryo surface while elongating parallel to it, encapsulating the microtubule spindle throughout division. The spindle axis aligns with the long axis of the pseudo-cleavage furrow (Fig. 9A), and proper spindle elongation is essential for faithful chromosome segregation. Defects in pseudo-cleavage furrow formation and/or mitotic spindle organization are therefore expected to compromise mitotic fidelity.

Based on these considerations, we visualized F-actin-based pseudo-cleavage furrows and microtubule-based mitotic spindles simultaneously in the same embryos, and quantitatively compared their morphology between 25 and 29 °C, as well as between normal divisions and mitotic failures.

During mitosis, F-actin caps are elongated and expand to fill the cortical space, thereby outlining the pseudo-cleavage furrows. Using these outlines, we measured the area of the F-actin caps and estimated aspect ratio by fitting an ellipse, which also allowed us to determine the orientation of the long axis (Fig. 9A). In parallel, to characterize microtubule spindle geometry, we measured spindle length by connecting the spindle poles and extracted the orientation of the spindle axis (Fig. 9A). Qualitatively, both F-actin caps and microtubule spindles appeared intact and comparable in overall intensity and structure across conditions, with the notable exception of their relative alignment (Fig. 9B–E). This prompted a quantitative comparison of F-actin cap and spindle geometry.

We found that F-actin caps were significantly larger at 29 °C than at 25 °C, both in normally completing divisions and in divisions that subsequently failed (Fig. 9F), while their aspect ratios remained comparable across conditions (Fig. 9G). In contrast, spindle length was selectively reduced in divisions that went on to fail, irrespective of temperature (Fig. 9H). Consistent with this observation, spindles associated with mitotic failures at 29 °C showed little to no net elongation during mitosis (Fig. 9I).

Together with the increased F-actin cap area, the reduction in spindle length indicated that the distance between spindle poles and the cortical F-actin network is increased at elevated temperature, especially in case of mitotic failures. Under these conditions, spindles appeared less constrained by the cortex and were frequently observed to rotate. To quantify this behavior, we measured changes in spindle orientation between consecutive time points throughout metaphase and summed these angular displacements. Cumulative spindle rotation was significantly higher at 29 °C than at 25 °C, both in control divisions and in mitotic failures (Fig. 9J). In addition, spindle axes were often misaligned with the long axis of the corresponding F-actin caps. Quantification of the angular mismatch between spindle and cap orientations at metaphase confirmed that this misalignment was increased at 29 °C in both control divisions and mitotic failures (Fig. 9K).

Taken together, these analyses indicate that mitotic spindles are less securely anchored to cortical F-actin at elevated temperature. We propose that this weakened coupling increases the susceptibility of spindles to rotational instability, thereby elevating the probability of mitotic failure. In combination with the spatial bias of mitotic failures toward central regions of the embryo and the tight coupling between mitotic failure and subsequent DNA damage, these findings identify mitotic failure as the primary driver of developmental vulnerability at elevated temperature.

## Testing robustness of cell polarity establishment during syncytial divisions at elevated temperature

Our findings thus far identify mitosis in the early blastoderm as a temperature-labile process and link mitotic failure to embryo lethality at elevated temperature. This raised the question of how a deeply conserved and generally robust process such as mitosis could become destabilized over a narrow temperature range. We therefore asked whether increasing the abundance of key components of the nuclear division machinery could compensate for temperature-induced fragility and restore developmental robustness at 29 °C.

A distinctive feature of syncytial divisions in the *Drosophila* blastoderm is their reliance on transient pseudo-cleavage furrows, which are repeatedly assembled and disassembled at each nuclear cycle. These furrows form through localized membrane insertion from late interphase to metaphase, support the alignment of mitotic spindles parallel to the embryo surface during pro- and metaphase, and regress rapidly during anaphase and telophase. As a consequence, each nuclear cycle requires the repeated reestablishment of cortical polarity cues alongside extensive cytoskeletal reorganization[24]. This dependence on transient cortical polarity represents a major deviation from canonical cell division in fully cellularized tissues and may constitute an evolutionarily young and potentially labile feature of insect embryogenesis. Moreover, polarity signaling is closely intertwined with orientation of mitotic spindles[33], and cytoskeletal organization in

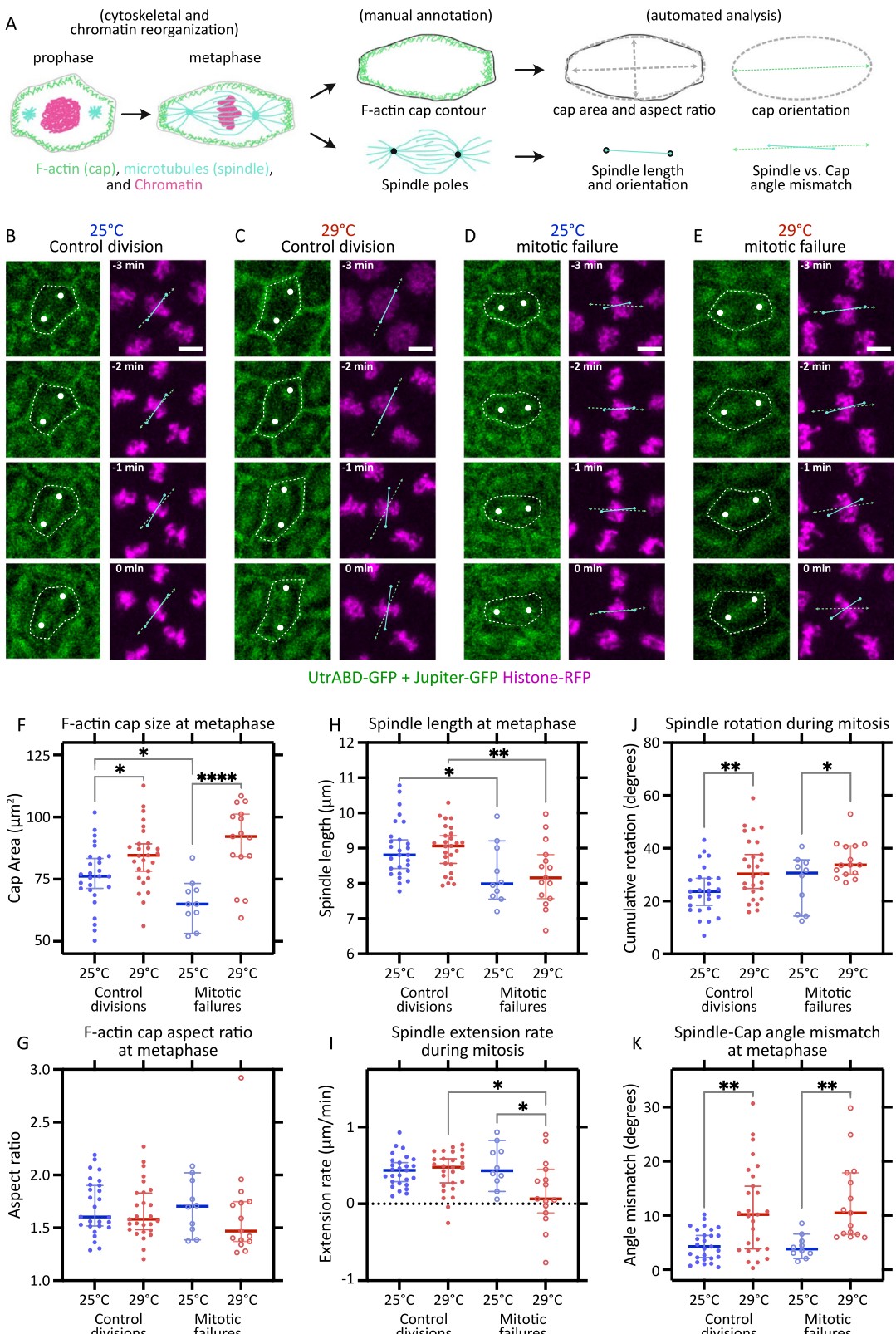

UtrABD-GFP + Jupiter-GFP Histone-RFP

epithelia[34]. We therefore tested whether temperature sensitivity during syncytial mitoses reflects insufficient robustness of polarity establishment: we aimed to strengthen polarity signaling through overexpression of two key polarity regulators active in the early blastoderm, Bazooka (baz; *Par-3* homolog) and Discs-large (dlg)[35]. Specifically, we asked whether either overexpressions could rescue embryo lethality at 29 °C, and if so, whether this rescue was accompanied by a

reduction in mitotic failures in crowded and combined regions of the blastoderm.

In embryos at 25 °C, we did not observe a significant change in embryo lethality with either *baz* or *dlg* overexpression as compared to control embryos with wildtype levels of baz and dlg (Fig. S5A). At 29 °C, lethality increased in control embryos (Fig. S5A), similar to results in previous experiments (Fig. 1B). By contrast, embryos with

**Fig. 9 | Weakened interaction between F-actin and microtubules precedes mitotic failures. A** Schematic depicts dramatic reorganisation of F-actin, microtubules, and chromatin as the nuclei transition from prophase to late metaphase (left). The images are then manually annotated (center) to facilitate automated quantitative analysis of F-actin caps and microtubule spindles (right). **B–E** F-actin cap and microtubule spindle dynamics during mitosis. Montages from representative control nuclear divisions (**B, C**) and mitotic failures (**D, E**) from embryos developing at 25 °C (**B, D**) and 29 °C (**C, E**), during mitosis 13. Left panels show montage of UtrABD- and Jupiter-GFP signal: dashed contour, manually outlined F-actin cap; white dots, manually marked spindle poles. Right panels show montage of corresponding Histone-RFP signal: dashed green line, long axis of F-actin cap;

solid cyan line, spindle position; insets, time till end of metaphase. **F–K** Quantifications showing F-actin cap size (**F**) and aspect ratio (**G**) at metaphase, microtubule spindle length at metaphase (**H**), spindle extension rate (**I**) and cumulative rotation (**J**) during mitosis, and the angle mismatch between spindle axis and cap long axis (**K**) at metaphase. $n = 27$ (25 °C control division), 27 (29 °C control division), 10 (25 °C mitotic failure), 15 (29 °C mitotic failure) nuclei, pooled from $N = 9$ (25 °C) and 9 (29 °C) embryos. Scale bars, 5 µm. **F–K** show scatters with median, with whiskers showing 95% CI. Non-parametric Kruskal–Wallis test with Dunn's uncorrected test for multiple comparisons: *, $p < 0.05$; **, $p < 0.01$; ***, $p < 0.001$; ****, $p < 0.0001$; non-significant comparisons are not shown. Source data and exact $p$-values for plots are available in the Source Data file.

either *baz* or *dlg* overexpression performed notably better at 29 °C, with lethality reducing to about two-thirds (*baz* overexpression) and half (*dlg* overexpression) of the controls (Fig. S5A). We note however, that the rescue of lethality at 29 °C in *baz* overexpressing embryos was only marginally significant, with a lot of variability between samples. Additionally, *dlg* overexpressing embryos at 25 °C showed higher variability in embryo lethality than their control counterparts. Consistent with these results, *baz* overexpression did not rescue the increase in the fraction of mitotic failures, either in the crowded regions or in the combined regions at 29 °C (Fig. S5B, C), and while the large blastoderm holes were absent in these embryos, *baz* overexpression occasionally produced small gaps in blastoderm (Fig. S5B, C). Then, *dlg* overexpression rescued the increase in the fraction of mitotic failures only in the crowded region at 29 °C (Fig. S5B, C), and while the large blastoderm holes were absent in these embryos, *dlg* overexpression increased the fraction of mitotic failures at normal temperature in both, the crowded region and in the combined region, occasionally producing small gaps in blastoderm (Fig. S5B, C).

Taken together, these results indicated that polarity establishment during syncytial divisions is comparatively robust to elevated temperature and that deviation from canonical polarity mechanisms are not, by themselves, sufficient to explain temperature-induced mitotic failure. Instead, the partial rescue effects suggest that the primary temperature-sensitive vulnerability lies downstream of polarity establishment, at the level of cytoskeletal regulation required to anchor and stabilize the mitotic spindle. This led us to focus on molecular interactions that directly link cortical F-actin to astral microtubules of the spindle during mitosis.

## Identifying vulnerabilities in cytoskeletal interactions during mitosis at elevated temperature

Since strengthening polarity establishment was not sufficient to robustly suppress mitotic failures at elevated temperature, we next turned to molecular interactions acting downstream of polarity cues that directly execute spindle positioning and chromosome segregation. Proper assembly and anchoring of the mitotic spindle at the cortex requires a dynamic interaction between microtubules and F-actin, which together anchor chromatin at the cortex throughout nuclear division[24]. Faithful segregation of sister chromatids, therefore, depends on tightly regulated interactions between the actin cortex and the astral microtubules of the mitotic spindle[36,37].

In *Drosophila*, these critical actin-microtubule interactions have been associated with a tripartite protein complex consisting of α-Catenin (αCat), which associates with cortical F-actin; β-Catenin (βCat, or *armadillo* in *Drosophila*), which binds to αCat; and Adenomatous polyposis coli 2 (Apc2), which links to βCat and binds to the astral microtubules of the mitotic spindle[38]. Within this complex, the interaction of αCat and βCat is direct and appears constitutive, whereas the interaction of βCat and Apc2 is mediated by a specific kinase, the *Drosophila* homolog of the *Glycogen Synthase Kinase 3*β, Shaggy (sgg)[39].

We reasoned that, if this tripartite protein complex was thermolabile, then our high temperature conditions should, at least in parts, phenocopy the loss of function of one of the components of the complex (αCat, βCat, or Apc2) or sgg. The loss of function of this protein complex has been associated with an "empty actin caps" phenotype, where cortical dome-like structures of F-actin are no longer associated with nuclei[38]. Thus, to test for the possibility that this tripartite protein complex is thermolabile during early embryonic development in *Drosophila*, we analyzed embryos that were fixed and stained to visualize nuclei and cortical F-actin (see "Methods"), both at 25 °C (Fig. S6A) and 29 °C (Fig. S6B). At control temperature of 25 °C, we could observe a tight association between nuclei and F-actin during interphases in the syncytial blastoderm, where cortical F-actin surrounds each nucleus, forming the actin cap. Notably, it was possible to identify gaps between neighboring actin-caps in cases of recently expelled nuclei (Fig. S6C), or in cases where the spacing between nuclei is large enough (Fig. S6D). Similar to embryos at 25 °C, such gaps could also be observed between neighboring actin-caps in cases of recently expelled nuclei at 29 °C (Fig. S6E). In contrast to controls at 25 °C, however, embryos developing at 29 °C additionally exhibited the empty actin caps phenotype (Fig. S6F), reminiscent of embryos mutant for *Apc2* or *βCat* (see Fig. 3d–f in McCartney et al.[38]). Thus, we hypothesized that stabilizing the activity of the complex, through elevated levels of αCat, βCat, Apc2, or sgg, could increase the fidelity of cortical mitoses at 29 °C, possibly through a tighter control of actin and microtubule coupling. Such tighter coupling, in return, might decrease the fraction of mitotic failures in crowded and combined regions and ultimately attenuate embryonic lethality.

To test the effects of elevated levels of either αCat or sgg on embryo lethality at control and elevated temperatures, we overexpressed either gene and quantified embryo lethality. At 25 °C, we did not observe a significant difference in embryo lethality between control embryos and those with *αCat* or *sgg* overexpression (Fig. 10A). At 29 °C, lethality was overall increased in controls, similar to results in similar experiments described above (Figs. 1B and S5A). By contrast, embryos with either *αCat* or *sgg* overexpression survived significantly better at 29 °C, with lethality reducing to about half (*αCat* overexpression) or a third (*sgg* overexpression) relative to the controls (Fig. 10A). Consistent with these results, blastoderm holes were absent in these embryos, and the fraction of mitotic failures in crowded as well as in the combined regions was reduced compared to control embryos at 29 °C (Fig. 10B, C). These results suggest that overexpression of either *αCat* or *sgg* strengthen the interaction between microtubule and F-actin, which in turn attenuates temperature-dependent destabilizations of the mitotic spindle. We therefore propose that the scaffolding activity of αCat, βCat, and Apc2, mediated by sgg, constitutes a thermosensitive controller of nuclear division fidelity in the syncytial *Drosophila* embryo.

To test whether thermosensitivity of nuclear division fidelity was dependent specifically on the scaffolding activity of αCat, βCat, and Apc2 rather than a common capacity of any regulator of mitotic

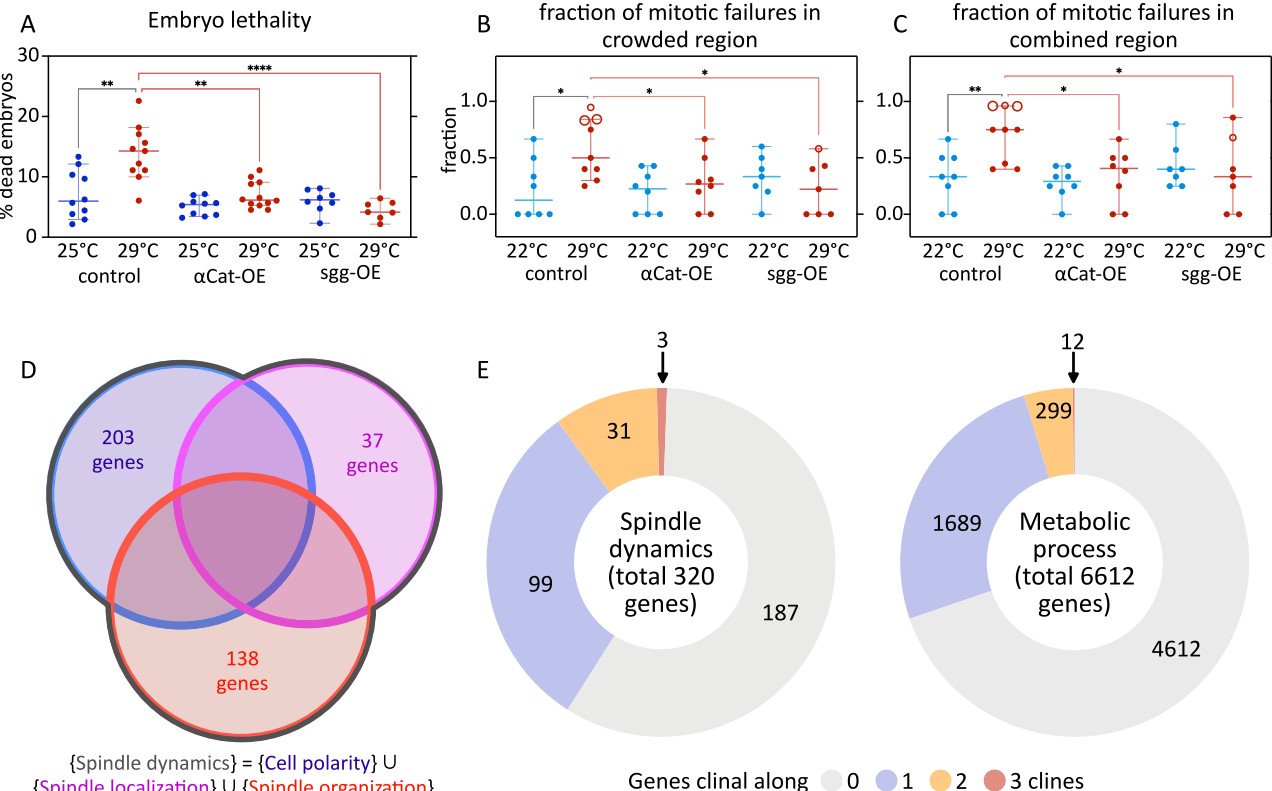

**Fig. 10 | Rescue of embryonic lethality at elevated temperature through overexpressions, and clinal genes as "natural" candidates. A–C** Testing the rescue of embryonic lethality and the fraction of mitotic failures with over-expression of *αCat* or *sgg*. **A** Scatter plot shows distribution of percent lethality in embryos from various genotypes, at normal (25 °C) vs elevated temperature (29 °C), calculated based on the fraction of embryos that don't hatch into larvae. *n* = 10 (25 °C control), 11 (29 °C control), 10 (25 °C αCat-OE), 12 (29 °C αCat-OE), 8 (25 °C sgg-OE), 7 (29 °C sgg-OE) samples; each sample has ≥18 embryos. **B, C** Scatter plots show distribution of the fraction of mitotic failures in crowded regions (**B**), and in combined regions (**C**), in the embryos at normal (22 °C) vs elevated temperature (29 °C). *n* = 8 (22 °C control), 9 (29 °C control), 8 (22 °C αCat-OE), 8 (29 °C αCat-OE), 7 (22 °C sgg-OE), 7 (29 °C sgg-OE) embryos. **D, E** Analysis of "natural" candidate genes that vary along thermoclines. **D** Venn diagram illustrates how we constructed a non-redundant list of genes associated with "spindle dynamics"

through the union of genes associated with GO terms, cell polarity (203 genes), spindle localization (37 genes), and spindle organization (138 genes). **E** Pie charts show clinal distribution scores for various genes associated with "spindle dynamics" as constructed in **D** (320 genes), or with GO term "metabolic process" (6612 genes). -1% of genes associated with "spindle dynamics", while -0.2% of the genes associated with "metabolic process", are clinal in all three continental datasets. See also, Supplementary Data 1 and 2. **A–C** shows scatters with median, with whiskers showing 95% CI. In **B** and **C**, large hollow markers, embryos with blastoderm holes; small hollow markers, embryos with small gaps in the blastoderm. Non-parametric Kruskal–Wallis test with Dunn's uncorrected test for multiple comparisons: *, *p* < 0.05; **, *p* < 0.01; ***, *p* < 0.001; ****, *p* < 0.0001; non-significant comparisons are not shown. Source data and exact *p*-values for plots are available in the Source Data file.

spindle anchorage, we asked whether the elevated levels of other regulators could similarly increase robustness against elevated temperatures. To address this question, we focused on regulators known to dictate the orientation of the mitotic spindle during cell division[40]. Both the activity of "Mushroom body defect" (mud; *NuMA* homolog) and that of Inscuteable (insc) has been previously described to regulate the anchorage and orientation of the mitotic spindle during cell division. A similar activity might be present for nuclear divisions during the syncytial blastoderm stage, raising the possibility that an overexpression of *mud* or *insc* could rescue embryo lethality, presumably by reducing mitotic failures. However, neither *mud* nor *insc* overexpression ameliorated the high embryo lethality at 29 °C (Fig. S7), supporting the notion that elevated temperature specifically challenges the core machinery of the mitotic spindle rather than being a non-specific failure that could be rescued by any mitosis-associated regulator.

### Thermosensitive regulators of spindle dynamics exhibit natural variation along thermoclines

While temperatures above 25 °C are generally considered non-standard rearing conditions for *Drosophila* lab strains, naturally

occurring populations likely do experience temperatures above this presumed optimum temperature. Since our analyses showed that elevated temperatures during the first hours of fly development affect both embryonic cell biology and organismal fitness, it may be tempting to predict that natural populations could be adapting to temperature changes through standing genetic variation and shifting allele frequencies in the same or similar genes that we identified as thermolabile regulators of nuclear divisions. To address this possibility, we analyzed the distribution of standing genetic variations associated with the regulators of spindle dynamics (Fig. 10D) and asked whether genetic variation would be distributed randomly with respect to the continent-scale gradients of annual mean temperature, i.e., thermoclines (see "Methods"). While most standing genetic variation associated with putative "spindle dynamics" regulators was indeed distributed without a clear thermoclinal pattern and did not allow for rejection of our null hypothesis (Fig. 10E and Supplementary Data 1 and 2), *sgg* was a clear and notable outlier in this analysis and exhibited a thermoclinal distribution of natural variations independently in populations in the North America, Europe, and Australia. Additionally, the fraction of genes associated with "spindle dynamics" that showed thermoclinal pattern was greater than the

fraction of genes associated with the GO-term "metabolic process" that showed thermoclinal pattern (Fig. 10E and Supplementary Data 1 and 2). These findings strongly suggest that the thermosensitivity that *sgg* exhibits in lab conditions constitutes a biological property, which can be exposed and selected for also in natural populations and during the course of evolution.

## Discussion

Is early fly development susceptible to elevated temperature? Our work demonstrates that processes of syncytial blastoderm formation and cellularization in *Drosophila* embryonic development are vulnerable to elevated temperature. Embryos exposed to 29 °C during these stages exhibit defects in maintaining the bilateral symmetry of developmental processes, and in extreme cases, enter a lethal trajectory during gastrulation (Fig. 1). We argue that these extreme cases of developmental failure likely coincide with structural defects in the blastoderm epithelium, which manifest as regions devoid of nuclei/cells and discontinuities in this tissue (Figs. 2, 3, and S1). The result is holes in the blastoderm, which arise from a greater incidence of mitotic failures in a small region of the embryo during syncytial nuclear divisions (Fig. 4). Our analysis also establishes a link between mitotic failures and the subsequent loss of nuclei, via DNA damage response (Fig. 8). We show that mitotic failures trigger DNA damage response in the defective daughter nuclei, which are then mitotically arrested, lose the integrity of their nuclear envelop, and are "kicked-out" of the blastoderm. We note that the DNA damage response is active only in nuclei with DNA damage, and not in its normal neighbors. It remains to be determined whether these lost nuclei deplete transcription factors required for local cell fate specification. This will, in turn, reveal whether blastoderm holes cause gastrulation defects purely through mechanical discontinuity, or whether loss of patterning information also contributes.

Our analysis shows that clustering of mitotic failures occurs in embryonic regions that lag behind in the nuclear division cycle and exhibit nuclear crowding (Fig. 6). While we treat these factors as independent for most of their analysis, a combination of these factors —captured in "combined regions"—predispose particular regions of the embryo to increased mitotic failures at 29 °C (Fig. 7). Thus, our findings are consistent with previous studies, which suggest that they are mechanistically coupled through the N/C ratio[25–30]. These findings suggest a mechanism in which the dependence of nuclear cycle duration on nuclear density could amplify the likelihood for mitotic failure at elevated temperature; whether a similar interplay affects embryonic development under normal temperature conditions remains unclear.

Alternatively—or additionally—the source of vulnerability of early blastoderm formation could stem from the non-deterministic distribution of nuclei, which relies on self-organization and is driven by the activity of cytoskeletal elements such as F-actin and microtubules[23]. A related form of self-organization may be seen in the progression of nuclear division cycles, which move across the surface of the embryo as non-deterministic waves[25]. Both processes appear to be challenged at elevated temperatures, suggesting that developmental mechanisms relying on biological self-organization may be particularly vulnerable at elevated temperature. Given the widespread importance of self-organization in biology, it may be valuable to explore more generally whether developmental processes that depend heavily on self-organizing components constitute common points of vulnerability.

What is the potential impact of elevated temperature on essential protein-protein interactions? We show that mitosis itself is vulnerable at elevated temperatures due to weakened interactions between microtubules and F-actin (Fig. 9). The tripartite complex, involving αCat, βCat, and Apc2 (mediated by the protein kinase sgg), has been implicated in mediating this interaction[38], at least partially, and our experiments overexpressing *αCat* or *sgg* rescue the developmental defects and embryo lethality at elevated temperature (Fig. 10A–C). These results support the idea that compromised coupling between microtubules and F-actin represents a key molecular vulnerability at elevated temperature.

A mild temperature increase, such as the one we studied here, is unlikely to denature proteins, but may instead weaken protein-protein interactions, such as the ones between components of the tripartite scaffold, involving αCat, βCat, and Apc2 (mediated by the protein kinase sgg). Such weakening would effectively deplete functional protein scaffolds, creating a loss-of-function-like scenario, as the equilibrium shifts towards the dissociated state of the scaffold. Nuclear crowding can exacerbate this effect: in crowded regions, individual nuclei experience a relative reduction in the local concentration of the scaffold owing to an increase in the N/C ratio. Thus, the combination of elevated temperature and local crowding provides a plausible mechanism for regional predisposition to mitotic failure and the formation of blastoderm holes. This interpretation is consistent with the observation that overexpression of either *αCat* or *sgg* rescues these defects (Fig. 10A–C), as increasing even a single component should be sufficient to raise overall scaffold abundance and counteract both temperature-dependent dissociation and crowding effects.

Independent support for this mechanistic framework comes from previous work in a different context. The nuclear crowding defect we report here is reminiscent of a previous study, focusing on embryos with increased maternal cyclin B, though at normal tempeatures[41]. While the impact of elevated temperature on cyclin B activity remains to be characterized, this study found that genetic perturbations enhancing microtubule-F-actin interactions suppress the phenotype. Even though the upstream perturbation differs from heat stress, the convergence on the same interaction interface supports the view that microtubule-F-actin coupling represents a common point of vulnerability.

Given that protein-protein interactions are omnipresent in biology, it is unlikely that microtubule-F-actin coupling is the only process directly modulated by temperature, and other aspects of basic cell biology may similarly be vulnerable. Because of their reliance on formation of protein scaffolds, it may be particularly rewarding to assess potential temperature sensitivity in processes such as cellular signaling[42,43], mitochondrial metabolism[44,45], cell polarity pathways[46,47], or the regulation of membrane trafficking[48], the cell cycle[49], and cytoskeletal dynamics[50,51], all of which are conserved across animal kingdom.

Are the effects of elevated temperature distinct from those of a heat shock? Previous studies in *Drosophila* have shown that temperatures of 33 °C or higher trigger a classic heat-shock response, characterized by de novo transcription of heat-shock protein genes[52]. This response has been observed across embryos[53], larvae, pupae[54,55], and adult flies[56], and overexpression of heat-shock proteins such as hsp23 can increase embryo survival under acute heat stress[57]. Heat shock proteins, accordingly, are often considered to act as capacitors, buffering cells against environmental and genetic variation[58,59].

However, the extreme temperatures that provoke a heat-shock response are only transiently encountered in natural environments[55]. Consequently, these conditions may not impose strong selective pressure to improve tolerance to sustained temperature increases associated with global warming. Supporting this, our analyses of three continental datasets did not reveal consistent allelic variation in canonical heat-shock genes. We can't exclude, however, that the lack of allelic variation stems from a limited standing genetic variation, given their critical role in survival, precluding the evolution of thermoclinal distributions. In either case, it is unlikely that heat-shock proteins would constitute a mechanism to improve developmental vulnerability in face of global warming.

In contrast, we observed allelic variation in *sgg* along thermoclines, suggesting it has been shaped by natural selection. Seasonal

shifts in *sgg* allele frequencies provide additional evidence, whereas heat-shock genes show no such pattern[60]. These observations suggest that while heat-shock proteins protect against acute temperature fluctuations, they may not confer selective advantage under elevated average temperatures. Instead, regulators of cytoskeletal interactions, such as *sgg*, could be targets of natural selection and serve as mechanisms to mitigate the effects of gradual temperature increases.

What are the implications of early embryonic vulnerability to elevated temperatures for insect survival and evolution? Early developmental processes from fertilization to blastoderm formation are highly conserved across insects[61,62], suggesting that the vulnerabilities we report in *Drosophila* embryos could apply to many other species. Weakening of cytoskeletal interactions under elevated temperature leads to asymmetric nuclear distributions, potentially disrupting the interpretation of positional information for subsequent morphogenesis, increasing the risk of embryo lethality. Such cellular vulnerabilities provide a mechanistic link between temperature stress and developmental failure, one that may be broadly conserved across diverse insect taxa.

Flies and other insects are essential for ecosystem stability[63], and rising temperatures pose a tangible threat to their survival. Reports of contemporary insect biodiversity loss[64,65] resonate with patterns of extinction and diversification during past climatic fluctuations[66,67]. Although multiple abiotic factors contribute to these historical trends[68–70], the current rapid rise in global temperature highlights heat stress as a unique relevant driver. Limited data make it difficult to test whether past warming events directly drove innovation in blastoderm architecture, but it is intriguing to speculate that early developmental processes may serve as both vulnerabilities and substrates for evolutionary change.

By identifying the cellular and molecular "breaking points" of heat sensitivity during early development, we can better predict which species and life stages are most at risk, guide conservation strategies, and even explore genetic pathways that enhance developmental robustness. Understanding these mechanisms offers a potential bridge from cell-biology to ecosystem-level predictions in a warming world.

## Methods
### Fly lines and genetics
*OregonR* line was used as WT *Drosophila*, and *w1118* line was used for outcrosses.

*Histone-RFP* expresses RFP tagged version of His2Av. Insertion on 3rd chromosome. Bloomington Drosophila Stock Center (BDSC) #23650.

*Jupiter-GFP* expresses GFP tagged version of the microtubule-binding protein Jupiter. Insertion on 3rd chromosome. Always used in combination with *Histone-RFP*. Gift from Jan Felix Evers.

*NLS-GFP* expresses GFP-tagged Nuclear Localization Signal peptide. Insertions on 2nd and 3rd chromosome. Always visualized alongside Histone-RFP. BDSC #1691.

*UtrABD-GFP* expresses GFP tagged version of Actin-Binding Domain of Utrophin. Insertion on 3rd chromosome. Always used in combination with *Histone-RFP* and *Jupiter-GFP*. Gift from Thomas Lecuit.

*67-Gal4* (mat αTub-GAL4-VP16) is a maternal specific Gal4 driver. Insertion on 2nd chromosome. Always used in combination either with *Histone-RFP* or *Histone-RFP* and *Jupiter-GFP* in the overexpression experiments (see below). Gift from Thomas Lecuit.

*SLBP-* and *SLBP.S118A-GFP* expresses GFP tagged versions of SLBP and SLPB.S118A under UAS promoter. Insertions on 3rd chromosome. Gifts from Eric Lecuyer. For these overexpression experiments, we first crossed virgin females from the Gal4 line "+; 67-Gal4; Histone-RFP" to the males of overexpression line "+; +; UAS-X" (X = *SLBP*, or *SLBP.S118A*). F1 virgin females from these crosses were backcrossed to males of the paternal genotype.

The following lines were used to drive the overexpression of various genes of interest in an attempt to rescue the effects of 29 °C. *baz*, BDSC #17613; *dlg*, BDSC #15779; *αCat*, BDSC #65590; *sgg*, BDSC #15886; *mud*, BDSC #22371; *insc*, BDSC #39676. An outcross to *w1118* was used as control. For these overexpression experiments, we first crossed virgin females from the Gal4 line "+; 67-Gal4; Jupiter-GFP, Histone-RFP" to the males of overexpression line "UAS-Y; +; +" (Y = *baz*, *dlg*, *sgg*, or *mud*) or "+; +; UAS-Z" (Z = *αCat*, or *insc*), or to males of *w1118* as control. F1 virgin females from these crosses were backcrossed to males of the paternal genotype.

Genotypes associated with various figure panels are listed in Supplementary Data 3.

**Embryo collection.** For any experiment, males and females of appropriate genotypes (Supplementary Data 3) were placed into transparent cylindrical embryo collection cages (inner diameter ~5 cm). The cage top was closed with fine-wire mesh for air exchange, and bottom with apple-juice agar plates. We prepared the apple-juice agar plates from 1 l of solution: we boiled 24 gm of agar in 750 ml of water, adding 250 ml of fresh store-bought apple juice while stirring on a hot plate, adjusting Sucrose to final amount of 25 gm (accounting for the sugar content of the apple juice). This solution yielded ~80 petri dishes (6 cm diameter). After the agar solidified, the plates were stored at 4 °C. Before the embryo collection, the plates were covered with a thin layer of yeast paste (to stimulate oviposition) and were pre-incubated to the embryo collection temperature for ≥30 min. Since *Drosophila* females can retain embryos long after fertilization, allowing early embryo development inside their abdomen, we performed a 1 h "sham" embryo collection at room temperature (24–25 °C), and discarded these embryos. Following sham collections, we collected the embryos for experiments at specific experimental temperature.

**Embryo lethality experiment.** We chose embryo-to-larva transition as a conservative and technically simple read-out for embryo survival. This excludes any potential later-arising defects in viability, behavior, lifespan, or fertility, meaning that our assay underestimates, rather than overestimates, the deleterious effects of elevated temperature. For estimating the survival, embryos were exposed to three temperature regimes (Fig. 1A). Depending on the regimen, we collected embryos for 1 h either at 24–25 °C (normal temperature) or 29–30 °C (elevated temperature). The 24–25 °C collections were done on the bench top. The 29–30 °C collections were done in a water bath, with embryo collection plate bottom in thermal contact with the heated water. For NN or EN, we collected embryos for 1 h at 24–25 °C or 29–30 °C, respectively, and then incubated for 2 h at the same temperature. The embryos were then bleached for 10–15 s, rinsed thoroughly with tap water, aligned on a 24 × 60 mm coverslip, and covered with halocarbon oil. For NE, we collected embryos for 1 h at 24–25 °C, incubated for 1.5 h at the same temperature, and within 30 min processed as above (bleached, rinsed, aligned, and covered), and subsequently incubated for 3 h at 29–30 °C. After these treatments, embryos from all regimes were incubated at 24–25 °C for ~1 day, following which we counted the number of eclosed larvae and dead embryos. The embryonic lethality was calculated as,

$$embryo\ lethality = \frac{100 * number\ of\ dead\ embryos}{number\ of\ dead\ embryos + number\ of\ larvae}$$

excluding the unfertilized embryos from calculations.

**Imaging live embryos (after temperature treatment).** Here we used 2 of the experimental temperature regimes used in the "Embryo lethality experiment" above, namely NN and EN. In short, we collected embryos at 24–25 °C (NN) or 29–30 °C (EN) for 1 h and then incubated the plate at the same temperature for another 2 h. Then, the embryos were

bleached for 35–40 s, rinsed thoroughly with tap water, aligned with the dorsal side facing a 22 × 40 mm coverslip, covered with halocarbon oil, and immediately taken for microscopy. The embryos were recorded for at least 15 h, at 25 °C. Images were acquired on Leica SP8 DMi8 inverted confocal microscope using 20× glycerol objective (0.75NA), at 2 µm step-size in z, and a final x-y resolution of 0.568 µm/pixel. We recorded the time-lapse at 1 frame every 4 min and recorded up to 5 embryos in parallel.

**Imaging live embryos (at various temperatures).** We used four temperature regimes. We collected embryos for 1 h either at normal temperatures (22–23, 24–25, or 27–28 °C) or elevated temperature (29–30 °C). Collections at 22–23 °C (22 °C imaging) were done in the fly incubator; 24–25 °C (25 °C imaging) on the bench; 27–28 °C (28 °C imaging) or 29–30 °C (29 °C imaging) in a water bath, with embryo collection plate bottom in thermal contact with the heated water. Then, the embryos are bleached for 35–40 s, rinsed with tap water at the collection temperature, aligned on a 25 mm diameter coverslip with the central part of the embryo facing the coverslip on lateral or ventrolateral region, covered with halocarbon oil, and immediately transferred to the temperature controller setup from Warner Instruments (QR-40LP holder on QE-1HC heater/cooler, coupled to LCS-1 liquid cooling and CL-100 bipolar temperature controller, precision of ± 0.1 °C), pre-equilibrated to the experimental temperatures (22, 25, 28 or 29 °C). For the data in Figs. 3–7, 10, S1–5, imaging was performed on Leica SP5 DMI6000CS inverted confocal microscope using 40×/1.1 NA water objective, 1× digital zoom, 2 µm z-steps (20 planes), and 0.379 µm/pixel x-y resolution; acquiring images at 1 frame every 30 s, 1 min, or 2 min depending on whether 1, 2, or 3–4 embryos were recorded in parallel. For the data in Figs. 8 and 9, the imaging was performed on Zeiss LSM 700 inverted confocal microscope using 63×/1.2 NA water objective, 0.5× digital zoom, 2 µm z-steps (12 planes), and 0.198 µm/pixel x-y resolution; acquiring images at 1 frame every 30 s or 1 min depending on the signal strength for the genotype.

**Imaging fixed embryos.** We collected embryos for 3–4 h either at 24–25 °C (normal temperature) or 29–30 °C (elevated temperature). Collections at 24–25 °C were done on the bench top; 29–30 °C done in a water bath, with embryo collection agar plate bottom in thermal contact with the heated water. Then, the embryos were bleached for 40–45 s, rinsed with tap water, and fixed immediately. We only image and analyze the embryos that are before the gastrulation stage. We performed two types of experiments; one, whole-embryo nuclear imaging (Fig. 2); and two, half-embryo imaging of nuclei and F-actin (Fig. S6), using distinct fixation/staining protocols.

Whole-embryo nuclear imaging: we used heat fixation and stained the embryos with DRAQ5. Briefly, we transferred the processed embryos to 50 ml tubes, drained the excess water, added the boiled-to-turbidity heat-fix solution (28% 500 µl NaCl + 5% 200 µl TritonX100 + 19.3 ml water), incubated for ~20 s with thorough mixing, and then immediately added water to cool down the mixture. Then, we transferred the fixed embryos to 2 ml microfuge tubes, sequentially added n-heptane and methanol in equal proportions, vigorously shook for ~30 s to remove the vitelline membrane, and washed 3× with methanol. We started the staining immediately after (though long-term methanol storage is possible). The embryos were rehydrated with PBT (0.01% solution of Tween20 in PBS, 3 washes), stained with 1:1000 DRAQ5 for 1-h, washed 6× with PBT, resuspended in 3:1 glycerol:PBS solution, and mounted between 2 rectangular coverslips, with spacing optimized to immobilize the embryos w/o deforming them. Then we imaged the embryos through both coverslips and the two halves stitched in image analysis (see below). Images acquired on Leica SP8 DMi8 inverted confocal microscope using 20×/0.75 NA glycerol objective, 1.15× digital zoom, 1 µm z-step, and 0.247 µm/pixel x-y resolution.

Half-embryo imaging of nuclei and F-actin: we used formaldehyde fixation and stained the embryos with DAPI and Phallacidin. Briefly, we transferred the processed embryos to a 20 ml vial containing the fixation mix (3500 µl PBS + 37% 1000 µl formaldehyde + 4500 µl n-Heptane). The vial was shaken on a platform shaker for 25 min at 200 rpm. Then, we transferred the fixed embryos to 2 ml microfuge tubes, sequentially added n-heptane and 90% ethanol in equal proportions, vigorously shook for ~30 s to remove the vitelline membrane, and washed 3× with 90% ethanol. We started the staining immediately after (though storage in 90% ethanol possible for ~1 week). The embryos were rehydrated with PBT (3 washes), stained with 1:1000 DAPI for 1 h, 1:500 Phallacidin for 1-h, washed 4× with PBT, resuspended in 3:1 glycerol:PBS solution, and mounted between 2 rectangular coverslips for imaging. Images were acquired on Leica SP8 DMi8 inverted confocal microscope using 20×/0.75 NA glycerol objective, 1.2× digital zoom, 1 µm z-step, and 0.473 µm/pixel x-y resolution.

**Image analysis, statistics, and reproducibility.** All image analysis was performed in FIJI. We conducted 11 types of analyses in live or fixed embryos: (1) Germband extension (live), (2) developmental landmarks analysis (live) (3) whole embryo reconstruction of nuclear distribution (fixed), (4) nuclear expulsions (live), (5) nuclear movements (live), (6) nuclear cycle asynchrony (live), (7) nuclear crowding (live), (8) fraction of mitotic failures (live), (9) SLBP nuclear localization (live), (10) F-actin and microtubule distribution (live), and (11) comparing the distribution of nuclei and F-actin (fixed). All the statistical analysis were performed in Prism (GraphPad), including curve fitting (see below, nuclear crowding, SLBP localization, and F-actin and microtubule distribution), non-parametric statistical significance tests (non-parametric Kruskal–Wallis test with Dunn's uncorrected test for multiple comparisons), and estimating the 95% confidence intervals (CI) on the medians. The straight-line fits for analysis of nuclear movements were done using the curve fitting functions in ImageJ macro language. For every experimental condition and analysis, "n" indicates the number of biological replicates, i.e., the sample size used to derive the statistics.

Germband extension (GBE): we measured the distance of the anterior end of the posterior mid-gut (PMG) from the embryo posterior end. The GBE was calculated as,

$$percent\ GBE = \frac{distance\ of\ PMG\ from\ posterior * 100}{length\ of\ the\ embryo}$$

Plotting percent GBE against time shows the fast and slow phases of GBE.

Developmental landmarks analysis: we noted down the time at which following developmental stages occurred. For 40% *GBE*, we noted down the time when the anterior tip of the PMG was closest to 40% embryo length from the posterior pole. For *Germband retraction begins* and *ends*, we noted down the time when the anterior tip of the elongated ectoderm on the dorsal side first moved towards the posterior and then stopped. For *head involution*, we noted down the time when the ectoderm meets in the anterior.

Whole embryo reconstruction of nuclear distribution: we used a custom-built image analysis pipeline to fuse the two views of the embryo, and visualize subcortical nuclei; either isolated or in clusters. We refer to each view as "half embryo", even though each spans about 70–75% of the embryo width, providing a substantial overlap. The pipeline comprises, as outlined in the following, elements in FIJI for processing the raw images, stitching the two halves, rendering a dorsal-ventral view, segmenting the embryo volume, and then color-coding the nuclei by distance from the segmented embryo surface. We first processed the raw images to improve visualization of the nuclei ("Minimum 3D" filter, radii x = y = 1, z = 3; then "Median 3D" filter, radii x = y = z = 1). Then, we homogenized the brightness/contrast between

the two half embryos. We then manually aligned the halves along their AP axis and used the BigWarp[71] within BigDataViewer[72] to register them using manually selected interest points (subcortical- or yolk-nuclei) visible in both halves. We tightly cropped the output images to reduce the computational load in the steps to follow. Then, we used BigStitcher[73] to further align the embryo, using the phase correlation method, to produce the "fused embryo". In the case of many embryos, additional manual transforms were necessary to obtain an acceptable fused embryo. The fused embryos were rotated and resliced to produce dorsal-ventral views. We then used thresholding on nuclear signal to segment embryo volume, generating a binary embryo mask, and used "3D Distance Map" (from "3DSuite" in ImageJ[74]) to compute the Euclidian distances from embryo surface for each voxel inside the embryo. Voxels with distance <18 μm were classified as cortical, 18–45 μm as subcortical, and >45 μm as yolk region (Fig. 2B); these distances were determined empirically, using measurements from various fused embryos. The regions were then color-coded and presented as maximum projections and cross-sections in Fig. 2. For embryos with a blastoderm hole, we further localized the hole in the embryo coordinates, and mapped these locations from multiple embryos onto a model embryo surface. The maximum intensity projections of the dorsal and ventral halves of the model embryo show that the blastoderm holes predominantly occur in the central region of the embryo.

Nuclear expulsions: we tracked expelled nuclei back in time to identify the mitosis when they were born and to categorize the expulsion as isolated (Fig. 4A, B) or pairwise (Fig. 4E). A nucleus was defined as expelled if its apical tip lay ≥12 μm away from the embryo surface. To simplify the identification of expelled nuclei, we color-coded the Histone-RFP signal as a function of distance from embryo surface. This is similar to above analyses, except we use the "LocalZ-Pojector" plugin[75] in FIJI, with embryo surface inferred from the co-imaged Jupiter-GFP signal. After color-coding the nuclei, we manually identify and count the number of pairwise or isolated nuclear expulsions.

Nuclear movements: we manually tracked the entire lineage for select few nuclei from the mother nucleus in late interphase 10 to all 8 great-granddaughter nuclei in late interphase 13. Nuclei from a single lineage were treated as a single "group", yielding a group center at each time point. We chose nuclei along one continuous row at interphase 10 and only chose those groups whose entire progeny (all 8 great-granddaughters) could be tracked, yielding ≥13 groups per embryo. For each group, we fitted a straight line to the time series of the group-center coordinates (dashed lines in Fig. 5B) to capture the global drift, and then computed, at every time point, the distance of the group-center to this line. An average of these distances during a specific period was used to estimate the group-center deviations over that period. Since the groups move strongly from anaphase through the first few minutes of the subsequent interphase, we analyzed "extensive-movement" and "low-movement" periods separately. Extensive movements consistently began ≥1 timepoint (30 s) before the first telophase appeared in any tracked groups and largely subsided within 3 min after all groups completed telophase. Thus, we defined the extensive-movement window as 30 s before the first telophase nuclei to 3 min after the last telophase nuclei. With these objectively defined windows, average group-center deviations were straightforward to compute. Then, at 29 °C, we further followed the progeny of the 8 great-granddaughter nuclei to identify those nuclear groups that have at least one mitotic failure during mitosis 13, leading to pair-wise nuclear expulsions in interphase 14.

Nuclear cycle asynchrony: we marked key time points in the timelapse recordings to follow nuclear cycle progression in the anterior-, middle-, and posterior-5[th] of the field of view. We focused on the transitions from a telophase 13 to interphase 14. If these timepoints are

"$T_a$" (anterior), "$T_p$" (posterior), and "$T_m$" (middle), the "asynchrony" can be calculated as,

$$asynchrony = T_m - \left( \frac{T_a + T_p}{2} \right)$$

Thus, we assume that in a fully synchronous embryo $T_m$ equals the average of $T_a$ and $T_p$; deviation from this expected value quantifies asynchrony.

Nuclear crowding: we analyzed nuclear distribution at the end of interphases, when the distribution is stable and crowding, if any, is maximal. We first segmented the nuclei using "Find Maxima" to detect local intensity peaks, generating an image of maxima. These maxima were then manually curated: we removed the maxima not corresponding to nuclei, added missing ones, and corrected the maxima that were clearly off-center. Then we used Voronoi tessellation to mark territories (pseudo-cells) around each nucleus. We excluded territories that (1) touched the edge of the image, (2) had aberrant shapes (aspect ratio>2), or (3) contained pixels with <5 units of intensity (typically poorly defined pseudo-cells at the edge of segmentation). Each nucleus was thus associated with a pseudo-cell area ("Area", μm²) and the x-coordinate of the pseudo-cell center ("X", μm) with x-axis aligned the embryonic AP axis. We then fitted a parabola to the Area-X relationship,

$$Area = aX^2 + bX + c$$

Greater nuclear crowding corresponds to stronger curvature (compare Fig. 6I–L); therefore, the fit parameter "a", which dictates the curvature, was used as a quantitative measure of crowding.

To account for the non-uniform nuclear distribution during syncytial blastoderm[76–78], we consistently quantified the nuclear crowding at 30–80% egg length (0%, anterior pole) and on the lateral or ventrolateral side.

Fraction of mitotic failures: we tracked expelled nuclei back in time (see above nuclear expulsion), but additionally recorded their positions, to map where mitotic failures occurred within the field-of-view. In parallel, we generated regions of (1) maximum nuclear crowding, (2) maximum delay in nuclear divisions, and (3) their combination. To construct crowded regions, we drew on previous studies showing that syncytial divisions arrest when a threshold nuclear content is reached at ~70% of the normal cellularization nuclear density[26–29]. The nuclear domain (pseudo-cell) area was previously reported to be 28.8 μm² at cellularization[79]. Accordingly, we assumed 6000 nuclei at interphase 14, each with an area of 29 μm². At 70% density, this corresponds to 4200 nuclei with area 41.43 μm², approximated as 42 μm². Thus, we estimated that divisions should arrest when pseudo-cell area falls below ~42 μm². At 29 °C, many nuclei in the central regions had areas <42 μm² in interphase 13, so we used this criterion to define regions likely to have reached the 70% density one division earlier, locally. At normal temperatures (22, 25, and 28 °C), nuclei rarely had areas <42 μm²; therefore, we defined crowded regions as nuclei in the lower quartile of the pseudo-cell areas. This construction is a practical rationalization and not a mechanistic statement about how the effect of nuclear crowding might manifest. Then, to define delayed region (maximally delayed nuclear divisions), we marked the areas containing the last ~20% of the nuclei to complete nuclear cycle 13. This empirically chosen fraction yielded regions similar in size as the crowded regions. The rationale, based on previous work[25], is that division control is coordinated at the embryo scale, and shared cytoplasm implies that locally delayed cycles will conflict with the embryo-wide biochemical status. Finally, we obtained the combined region using the logical OR operator of crowded and delayed regions, to test the combinatorial effects of crowding and asynchrony.

In the end, for each embryo we computed the fraction of mitotic failures in each region as,

$$fraction\ of\ mitotic\ failures = \frac{number\ of\ mitotic\ failures\ in\ a\ specific\ region}{total\ number\ of\ mitotic\ failures}$$

As we can imagine, the fraction of mitotic failures will also depend on the size of the specific region, as compared to the total field of view. As such, in case of both crowded and delayed regions, we empirically determined criteria for their construction, such that the size of these regions will be comparable: ~20% of the field of view. In case of combined regions, however, the size can vary from ~20% (crowded and delayed regions show complete overlap) to ~40% (crowded and delayed regions show no overlap).

There was a marginal increase in embryo lethality at 25 °C in the controls for the overexpression experiments, which presumably happens due to the presence of additional transgenic insertions (UAS and Gal4). Due to this reason, to err on the side of caution, we chose to perform the imaging at 22 °C, specifically for the Fraction of mitotic failures analysis. As such, we don't expect any biologically significant difference between results at 22 and 25 °C in terms of embryo lethality, blastoderm holes, or other defects.

SLBP nuclear localization: we tracked nuclei from late interphase 13 till early interphase 14 in embryos expressing GFP-tagged SLBP and Histone-RFP, developing at 25 or 29 °C. To measure the nuclear SLBP intensity, we used Maximum Intensity Projected images: we drew ROIs (3 μm diameter) in the center of the nuclei in RFP channel, applying these ROIs to GFP channel. The same procedure was applied to embryos expressing GFP-tagged SLBP.S118A or NLS, along with Histone-RFP, to compare localization kinetics. After identifying the two components of SLBP loss during interphase 14 in nuclei with mitotic failure during mitosis 13, we fitted a straight line to nuclear intensities between 0 and 4 min of interphase 14.

F-actin and microtubule distribution: we tracked F-actin caps and microtubule spindles around the same nuclei during mitosis 13 (late prophase to end of metaphase) in embryos developing at 25 or 29 °C, and followed the divisions to classify them as normal or mitotic failures. For F-actin caps, we manually outlined the cap and fitted an ellipse to this contour to obtain cross-sectional area, aspect ratio, and the orientation of the major axis. For microtubule spindles, we manually drew a line between spindle poles to measure the spindle length and the spindle axis orientation. These measurements were used to plot F-actin cap area and aspect ratio at metaphase, spindle length at metaphase, and the relative angle between cap long axis and spindle axis. Cumulative spindle rotation was computed as the sum of absolute angular changes between consecutive time points. The rate of spindle length change was estimated as a linear-fit to spindle length over time across mitosis.

Comparing the distribution of nuclei and F-actin: we used a pipeline similar to that for "whole embryo reconstruction of nuclear distribution", but without fusing two half-embryos and starting directly from embryo-volume segmentation using the F-actin staining. A 3D distance map was then computed and nuclei classified into cortical, subcortical, or yolk based on their distance from the embryo surface. These nuclei were color-coded and presented as sum projections in Fig. S6 (except for yolk region).

**Analysis of thermoclinal genes.** We sought genes showing variation across natural populations of *Drosophila melanogaster* along large-scale climatic gradients. At continental scale, we reasoned that climatic variation imposes selection gradient with distance from equator, generating geographic genomic variation, i.e., a thermocline, across a temperature range of at least 15 °C. Previous studies reported such variation at various analytical details (whole genome,

Quantitative Trait Loci, or individual genes/proteins, see Supplementary Data 1 for the list of studies). To be able to aggregate over these scales, we chose "genes" as a level of abstraction, and without explicit accounting for altitude as an independent variable. Consequently, we focus our analysis to genes that show consistent clinality across data sets, rather than to specific allelic change (coding vs. non-coding, position in the protein, etc.) that are involved. We thus extracted only genes reported to vary across geographically distant populations, and grouped studies by continent (North America, Europe, or Australasia), assuming these continent-scale gradients arose independently. The resulting "thermoclinal datasets" were then used to assign a score of 0–3 for a gene, counting the number of appearances in individual datasets.

To assemble genes relevant for spindle dynamics, we combined the genes associated with Gene Ontology (GO) terms cell polarity, spindle localization, and spindle organization; none of the individual GO terms alone capture all six candidate genes tested in this study. We queried the thermoclinal datasets with this combined list of genes related to spindle dynamics, assigning each gene a 0 to 3 score as above. For comparison, we applied the same procedure to genes associated with GO term "metabolic process". See also, Supplementary Data 1 and 2.

### Reporting summary
Further information on research design is available in the Nature Portfolio Reporting Summary linked to this article.

## Data availability
Any associated raw data are available upon request from corresponding authors. Source data are provided with this paper.

## Code availability
Codes were developed for batch processing of images, and are available upon request from corresponding authors. The codes developed in the study are used exclusively for automation and have no bearing on the results.

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

## Acknowledgements

We would like to thank Lázaro Centanin, Nicolas Gompel, Marianthi Karageorgi, and Russ Hodge for insightful comments on various drafts of the manuscript, and Lemke lab members for discussions throughout the project. We are grateful to Jochen Wittbrodt for his enthusiasm and his continuous and generous support. We would also like to thank Ingrid Lohmann, Anja Nagel, and Kristen Panfilio for *Drosophila* culture-related support, and Thomas Lecuit, Jan Felix Evers, and Eric Lecuyer for sharing fly lines. We sincerely appreciate the efforts taken by FlyBase to maintain a curated database and Bloomington Drosophila Stock Center for providing transgenic fly lines. This work was funded by the DFG (LE 2787/3-1) to S.L., while G.K. was supported through an HFSP Long-term postdoctoral fellowship (LT000597/2019-L).

## Author contributions

G.K. and S.L. planned the project. G.K. did the experiments with help from P.A. for *baz* and *dlg* overexpressions, and from JJ.D.-L. for co-staining of nuclei and F-actin. G.K. did the image analyses and statistical analyses. G.K. and S.L. wrote the manuscript, and all authors gave comments on the manuscript.

## Funding

## Competing interests

The authors declare no competing interests.
