## [Transparent Peer Review file · Nature Communications]

Elevated temperature fatally disrupts nuclear divisions in the early *Drosophila* embryo

Corresponding Author: Dr Steffen Lemke

Version 0:

Reviewer comments:

Reviewer #1

(Remarks to the Author)

Kale et al. describes in detail the deleterious effect of increased temperature during early insect development. Previous work showed robustness of *Drosophila* embryonic development despite fluctuating temperatures. However, here the authors show that the initial stages of syncytial blastoderm is sensitive to high temperature, which causes modified patterns of mitotic division, altered local nuclear crowding and loss of nuclei from the embryo cortex. These are interesting findings worth of being published. After this descriptive portion of the paper, the authors ask about the mechanisms behind this heat response and test the effect of potential candidate genes to rescue those developmental defects. Although their findings here are exciting, the reasoning and hypothesis for the group of genes tested is a bit lacking (specially the mitotic division group), and some of the conclusions are not well based on the results. For example, the authors propose that high temperatures disrupt protein-protein interactions. Assuming the authors are implying that the protein-protein interactions are affected due to changes in protein structure because of high temperature, how then simply adding more of those proteins would rescue the phenotypes observed? Wouldn't the added proteins still be affected by high temperatures? The findings on *sgg* also require more explanation. What sort of allelic variants are being referred to, do they modify coding regions? Which regions on the protein, and what anticipated modifications they may cause to the protein function? Again, how can the authors compare over-expression of *sgg* vs. natural allelic variants and the ability to buffer embryos from the effects of high temperatures? The definition of lethality used in the paper appears to be based on larvae eclosion only. Do these larvae survive and reach adult stages? Are the flies fertile? Do they display any kind of phenotypes (locomotor defects, short life span, etc)? I believe these are important questions to be answered when arguing about the deleterious effect of temperature on development. Just eclosion after 24 h is a narrow window. In addition, controls would give a good idea of phenotypes caused by the live-imaging preparation itself.

The paper would benefit from the revisions above and additional ones, including the following:

- 1- The authors should consider moving some of the supplemental figures to the main figures. For instance, the holes and nuclei extrusions are much better seen in Figs S1 and S2.
- 2- Defects in chromosomal segregation are not clear in the figures presented (Fig. 3E).
- 3- As cellularization proceeds, the nuclei move towards the center of the embryo (e.g. see Keranen et. al, 2006; Fowlkes et al., 2011; Xue et al, 2023) and increase cell density in that area. Thus, it is important to show what stages were analyzed during cycle 14 embryos between control and experimental in terms of membrane invagination, not just timing, since at high temperatures the embryo may develop faster than at low temperatures.
- 4- Method section is written too colloquially, with unnecessary details and at the same time, missing important information.
 - a) For example, the precision ratings for temperature of the equipment used is not mentioned (incubators and microscope stage). What is the expected +/- variation in temperature? This is pertinent since the authors claim significant phenotypes caused by 29C and not 28C.
 - b) Methods of screening for clinal variations is not explained, only described in the supplement Table legends. Types of alleles should be better described.
 - c) The section describing the analysis of entire embryos ("visualize all nuclei in the entire embryo") is not credible for a number of reasons. First, the confocal used will not provide sufficient resolution along the Z axis to allow for segmentation of nuclei along the curvature of the embryo (e.g. see Keranen et al. 2006 methods description). The proper way to do so would be to use a spinning disk or similar microscope. Second, the stitching method does not rely on labeled landmarks in the embryo (e.g. a gene expression pattern), therefore the junction is a guess. This is not to say that an analysis of the entire

embryo is necessary, but the paper should be revised and correctly mention that what is being visualized are the embryo top and bottom surfaces and not the entire embryo. If the authors indeed managed to segment the entire embryo and can show landmarks used for stitching, then include the data of all segmented objects that show the expected ~6000 nuclei around the embryo cortex in controls in Figure 2 and contrast that to the experimental embryos.

5) Reference 57 (Edgar et al.) should be better presented and discussed in the paper in terms of nucleo-cytoplasmic ratio for mitotic divisions and nuclear density control. I was not able to find this reference being cited in the main text or elsewhere.

6) Discs large (dlg), not disks large

7) Consideration about historic high Earth temperatures in insect evolution could be discussed.

Reviewer #2

(Remarks to the Author)

Kale et al describe the effects of a temperature shifts on *Drosophila* embryogenesis and characterize several types of common failures in development at high temperature. After a careful and detailed description of the changes in cell divisions and localization of nuclei, they test and identify a few candidate genes that can rescue the lethality and mitotic defects. They suggest this may relate to genes that vary in natural populations that are linked to environmental temperature. Overall, the paper explores a very interesting premise that could have major implications, and some of the imaging is very nicely presented and the genetic results are intriguing. However, some of the data is too weak or difficult to interpret as described, and so some conclusions are not yet sufficiently supported.

Major concerns:

The title of the paper refers to the fatal disruption of nuclear division, but it is not clearly demonstrated that nuclear division itself is faulty, nor are changes in the nuclear position shown to be causative to the fatality (although it may correlate – this is also not directly shown). There is a nice, detailed analysis of the nuclear divisions and the movement of nuclei away from the cortex in Figure 3. At some point in the text, it seems that any nuclei that are subcortical are classified as “mitotic failures” but I do not understand why. Is there a marker that indicates mitotic failure or incomplete nuclear division? On line 197, the authors say “We observed chromosome segregation defects preceding the pairwise loss of nuclei (Figure 3E), at all our experimental temperatures.” This figure shows Histones/nuclei and depth of nuclei (magenta), but there is not enough detail to show a chromosomal segregation defect. They conclude later (line 445) “mitosis itself is vulnerable at elevated temperatures.” From the images, it looks like mitosis completes normally but then some nuclei are pushed out afterwards, which seems like a distinct type of defect. In my opinion, a mitotic failure needs to be shown more directly. Also, are the embryos with more displaced nuclei more likely to die? The only link they have between these is the similar frequencies of occurrence, which is a correlation at best.

It is certainly interesting that the lethality and some of the nuclear “mitotic failures” are shown to be rescued by the overexpression of *sgg* or *a-catenin*. However, it is an oversimplification to say this works through cytoskeletal/polarity components since these are known to signal in major signaling pathways (eg, *Wg* and *Hippo* pathways) and can affect many things, not just actin. It is not appropriate to conclude, as the authors do in the abstract that, “Our genetic rescue experiments show that in *Drosophila* embryos, the interaction between cortical F-actin and microtubules is vulnerable to disruption at elevated temperatures.” [Or that (line 397) “These results indicate that the experimental strengthening of interaction between microtubule and F-actin is able to attenuate temperature-dependent destabilizations of the mitotic spindle.” Or that (line 434) “Failing interactions between microtubules and F-actin make early fly development susceptible to elevated temperature”.] To make these conclusions, additional components would need to be tested (eg disrupting or overexpressing actin or tubulin or their direct regulators) in the different temperature conditions. They could at least test their model further and validate their arguments by looking at apical F-actin caps directly when *sgg* or *a-catenin* are up or down regulated at different temperatures.

Although p-values are provided for quantitative data, the specific statistical tests performed are not described, making it difficult to assess if the comparisons are appropriate. (This is especially a concern for the fractions in Fig 6, for which a statistical test should be accounting for lower incident rates in certain conditions.)

Text area beginning on line 137 and related Figure 2 discusses “blastoderm holes”, also sometimes called “gaps” or “subcortical nuclei” in the text (and the switches in nomenclature can be confusing). The terms refer to nuclei that move away from the cortex of the embryo. This is reported to occur in 12% of embryos but it seems like only two cases are shown in Figure 2D/E (out of 77, I think), making it hard to evaluate any potential patterns to the depth of the loss or the stated L/R asymmetry. More data needs to be shown or quantified for this to be convincing- perhaps they can quantify the fraction of yellow/magenta pixels/voxels for different regions of multiple embryos and present this more quantitatively. As it stands, it looks like there is randomness in the “holes” and while there is more on the left side of one example, it is hard to know if that is just true for this case, or why it is even an important point if it is random. Additionally, the embryo in E looks to be slightly older with some head involution starting- are these all staged together?

Claims that some of the observed defects are “a function of temperature” are overstated; for example, in figure 4C, asynchrony is only observed at 29C, not increasing with each temperature difference. In fig 5M, 22C shows differences from 28C or 29C, and Fig S6 only two temperatures are shown. So, the defects should be attributed to the high temperature only, not as a “function of”, which implies it’s gradually shifting.

It is very interesting to consider how certain genes are affected by temperature, and there are major implication for this idea.

However, it is not very clear how they identified the natural variants or where the temperature-dependent clinal scores come from over the large continents analyzed. More detail about this, such as the range of temperatures considered for the natural variants, would be helpful. Why did they combine the “spindle dynamics” category- do they not see overrepresentation in the individual categories of polarity, spindle localization, and spindle organization?

Minor concerns:

For people not familiar with flies, the different temperatures discussed in the introduction are not explained very clearly- it is not obvious that flies are often raised at 25C or 22C or that 29C is hot.

There is too much explanation in the figure legends (for example figure 3 legend that argues the statistical significance is not meaningful- this should be in the results or discussion).

Partway through the paper the authors start looking at different temperature than in the initial characterization, which raises the question of what the differences are between 22 and 25 in terms of lethality, blastoderm holes, etc.?

Figure 5A, why does the region of nuclei considered not include many at the poles? Does this introduce a bias in the size?

The rescue results with the polarity protein Baz and Dlg are hard to interpret given that the same-genotype controls at a low temperature also showed high levels of mitotic failures, but the transgene is likely upregulated at the higher temperature due to effects on Gal4. Thus, the temperature dependent effect is not clear. Similarly, results for Mud overexpression are difficult to interpret.

The manuscript might also improve overall if it were a bit more focused with less supplemental data.

There are numerous typos in the main text. Generally there is room for improvement in how the data is represented.

Reviewer #3

(Remarks to the Author)

This manuscript by Kale et al addresses the mechanisms that impact embryo development at elevated temperature in *Drosophila*. This is an interesting problem, as the impact of temperature on development remains poorly understood and is a question of increasing relevance in the face of global warming. The authors find that the vulnerability of embryonic development to an increased temperature (29 C vs 25 C) is mainly due to defects in the blastoderm divisions. This is an interesting observation as these divisions show specialized control mechanisms to ensure rapid cleavage cycles. The authors then perform a detailed analysis of these divisions and provide insights on the cause of developmental defects, by linking them to mitotic defects. Most notably, by genetic manipulation of factors involved in cell polarity and spindle regulation they can rescue some of the observed defects. They also show that in species that naturally live in increased temperature conditions, the expression of the same factor(s) is similarly up-regulated possibly identifying a mechanism for increase robustness at elevated temperature. This last experimental result represents the main finding of the paper in my opinion and could be of general interest. Overall, the paper is well-written and the conclusions supported. A major area where this paper needs improvement is in relating the authors' findings with relevant literature. It seems that the authors are not aware of or have ignored a large body of literature on the regulation of cleavage divisions in *Drosophila*.

Major comments:

- Previous work has linked the loss of nuclei to the activation of the DNA replication checkpoint, Chk2 in particular (see for example lampietro et al Dev Cell 2014). It seems important to analyze that possibility here. While it is clear that lost nuclei happen in pairs (thus likely due to some mitotic defects), it can't be ruled out that these mitoses were defective due to previous DNA damage and activation of the checkpoint. The results of this analysis would be interesting regardless and would add to the paper.
- There is clearly a connection between nuclear density and mitotic duration in early fly embryos. However, in their analysis the authors treat the two variables as separate, which is confusing and inaccurate. The impact of DNA content (or expressed genes) on cell cycle duration (Blythe and Wieschaus 2015, Deneke et al 2016) and the importance of uniform nuclear positioning for mitotic synchrony are clearly documented (Deneke et al 2019), as well as how inhomogeneities in nuclear density can affect the collective decision of nuclei to arrest the cell cycle at the maternal-to-zygotic transition (Hayden et al 2022). Strangely, these results are not mentioned, and it is not proposed that the increased cell cycle asynchrony could be due to the improper nuclear positioning. Related to this, previous defects in nuclear positioning have been shown to be due to impaired cortical contractions in the pre-blastoderm stages when these contractions generate cytoplasmic flows that move the nuclei along the AP axis (Deneke et al 2019). Again, strangely there is no mention of this nor the possibility that temperature might impact an earlier morphogenetic process is analyzed. While an analysis of this process would be interesting and doable and add significantly to the paper, the authors should at least discuss it. In this version, there is essentially no understanding of the major process driving mitotic and developmental defects, that is nuclear crowding.
- Schubiger's lab performed a screen of mitotic defects and rescues in embryos with increasing cyclin B copies (Ji et al Genetics 2002) finding factors similar to those identified by this paper. Importantly, in these genetic conditions similar nuclear localization defects were also observed so it seems important to discuss this paper, as there are possible links to the authors' findings.

Minor comments:

- In several places, the description of morphogenesis of pre-blastoderm embryos is strangely inaccurate. Nuclei do not migrate from the interior to the cortex in interphase of cycle 10, but they move towards the cortex in a step-like manner from

cycle 7-8 to cycle 10 (Foe and Alberts 1983 and Hur et al BiorXiv 2023 for a rigorous, quantitative analysis). It would be appropriate to describe these processes more accurately.

Version 1:

Reviewer comments:

Reviewer #1

(Remarks to the Author)

The authors made a series of additional experiments in the paper, reorganized main and supplemental figures, and made significant text editing that improved the manuscript. Concern # 3 "as cellularization proceeds, the nuclei move towards the center of the embryo (e.g. see Keranen et. al, 2006; Fowlkes et al., 2011; Xue et al, 2023) and increase cell density in that area. Thus, it is important to show what stages were analyzed during cycle 14 embryos between control and experimental in terms of membrane invagination, not just timing, since at high temperatures the embryo may develop faster than at low temperatures."

The authors did not adequately address this point. The explanation given is that the analysis of nuclear crowding was done in cycle 13, and does not refer to cycle 14. However, the distribution of nuclei in wild type embryos (developing at normal temperature) is not uniform at the onset of cellularization, which means that the nuclei distribution is already asymmetric at cycle 13 regardless of temperature. Second, several of the figures in the paper actually point to the nuclear holes/extrusion phenotypes in stage 14 (eg fig 2, 3, S1, S2,etc), where the size of holes could be amplified in low density areas (the ventral side), whereas nuclear extrusions are more common in high density areas (the dorsal side; this is actually seen in wild type embryos in normal temperatures). Fig 2 indeed shows this effect, and the authors do not have an explanation for the differences seen between dorsal and ventral sides. Finally, F-actin apical caps, and adherens junctions, which interact with beta-catenin and sgg, are also involved in the cell constriction and cell density distribution, which again is likely to amplify the effects seen at high temperatures, decrease cell contacts and push cells inside the embryo. Thus, the effects of asymmetric cell densities in cycle 13 and 14 should not be ignored as a contributing factor.

For the thermocline analysis explanation, the authors need to write a short clarification about latitude vs. altitude in the datasets, as a high altitude location may have lower temperatures even if closer to the equator than low altitude regions.

The images showing defective mitotic divisions are much better now, and to this reviewer, the main defect that can be seen is a rotation on the plane of division. The new data provided on DNA damage with the timeline of SLBP nuclear accumulation helps build the case that the DNA damage might be caused by mitotic failures and not triggered by the high temperature, but I think this still remains quite speculative in the absence of an independent test. For example, if cycle 14 embryos are heated to 29C, would you then never see DNA double strand breaks (e.g. using anti-H2A) at this temperature? I am not suggesting the authors need to do more experiments, but to consider future experiments to clarify the issue. Along these lines, it may indeed be better to modify the title of the paper.

In Fig 8 C, it is hard to see the decrease in nuclear SLBP signal. Can the authors add an explanation on how the signal quantifications were done (in the entire nuclei in 3D?, in different Z positions?), and if it was before the defective nucleus moved inwards.

Overall, the authors made significant effort in improving the methods, results and discussion.

Reviewer #2

(Remarks to the Author)

The manuscript by Kale et al. describe the common failures in Drosophila embryonic development at high temperature. After describing changes in mitotic divisions, localization of nuclei, and DNA damage response, they identify two candidate genes that can rescue the lethality and mitotic defects and suggest these vary in natural populations from different climates. Overall, the paper explores a significant topic with an extensive set of experiments and adds new insight to existing understanding of organismal response to the changing environment.

The revised manuscript describes the authors' logic and results more clearly, and includes a more thorough description of the literature. In addition, the manuscript has been notably strengthened by more detailed characterization of the mitotic spindle and DNA damage, and adjustments to the text and figures have satisfied most of my concerns. At times, I think some statements are a bit strong and imply more direct causation than the correlations in the data, but this has improved and most of the interpretations accurately represent the results. Along these lines, I do support the alternative title suggested in the response to reviews, but I will leave it up to the authors and editor to decide on that issue.

Reviewer #3

(Remarks to the Author)

The authors have addressed my previous comments satisfactorily.

Response to reviewer comments:

We sincerely appreciate the efforts of the reviewers and are happy about the overall positive reception of the manuscript. Please see our detailed point-by-point responses in blue ink, along with quoted text from the revised manuscript in green ink. We have numbered the comments whenever possible, and, in a few instances, split the comment and answered it in sections.

To facilitate the review of our revised manuscript and potential next round of revisions, we would like to first briefly list the major and minor changes in the manuscript during revisions. We believe that these changes have added a few key mechanistic details, bolstered some of the weaker claims in the initial version, and clarified the main message.

Major changes:

1) Following up on comments from reviewer #3 (Major comment 1), we used the localization of Stem-Loop Binding Protein (SLBP) as a reporter for DNA damage (Iampietro et al., 2014). In this experiment, we establish that it's the mitotic failure that leads to DNA damage. The results are now described in the revised Figure 8.

2) Following up on comments from reviewer #2 (major concern 5b), we attempted to visualize F-actin to quantify possible defects in its distribution at elevated temperature – using GFP-tagged Utrabin Actin-Binding Domain (UtrABD). In our pilot experiments, we realized that expressing UtrABD-GFP rescues the mitotic failure defect at elevated temperature, and that blastoderm holes don't form anymore. As a result, currently, we cannot assess the possible role of changing F-actin distributions in the rescue of temperature-related defects by the overexpression of α -catenin or Shaggy. Nonetheless, we decided to co-image UtrABD-GFP and Jupiter-GFP to simultaneously visualize F-actin and microtubule distribution in early embryos, especially during mitosis. The blastoderm holes were still absent in these embryos at elevated temperature; however, the embryos did exhibit an increase in the number of mitotic failures at elevated temperature, which allowed us to quantify changes in the shape of F-actin caps and microtubule spindles. The results of this experiment are presented in the revised Figure 9.

3) We have now extended the Discussion section, following up on the reviewer comments below.

- a) Comments from reviewer #1 (major comment 1): We now discuss our view on how protein-protein interactions, such as those leading to the formation of a protein scaffold, are affected by elevated temperature.
- b) Comments from reviewer #1 (minor concern 5) and reviewer #3 (major comment 2): We now discuss our results on nuclear density (crowding) and nuclear cycle durations (asynchrony) from the perspective of the nuclear-to-cytoplasmic ratio.
- c) Comments from reviewer #3 (major comment 3): We present our results in the context of a previous study (Ji et al. 2002) that identified genes related to microtubule-F-actin interaction in a screen to identify genetic interactors of Cyclin B.
- d) Comments from reviewer #1 (minor concern 7): We comment on how our results could be viewed in the context of historic high Earth temperatures and the effect that might have had on insect evolution.

We believe that these additions have enriched the narrative, also giving us an opportunity to flesh out ideas and present our thoughts more explicitly.

4) Following up on comments from reviewer #1 (major concern 2 and minor concern 4b) and reviewer #2 (major concern 9), we have now included a section in methods on how we analyzed the clinal data.

Minor changes:

1) Following up on comments from reviewer #1 (minor concern 4c), we have now included additional cross-sectional views of the embryos in the revised Figure 2 to demonstrate that the embryos are indeed stitched to render the full volume of the embryo. Also, following up on reviewer #2 (major concern 7b), we have now included a diagram to show the distribution of blastoderm holes in the revised Figure 2.

2) Following up on comments from reviewer #1 (minor concern 2) and reviewer #2 (major concern 3), we have now included an additional timepoint in the revised Figure 4E to depict the chromosome segregation defects better. Further examples of the chromosome segregation defect can now be seen in revised Figure 8C-E.

3) Following up on comments from reviewer #1 (minor concern 1) and reviewer #2 (minor concern 6), we have shifted the following data from supplementary figures to main figures.

- a) Figure S1 is now upgraded to revised Figure 3. The figure is also extended to include embryos at temperatures 22°C and 28°C and an additional time point in early interphase 10.
- b) Our analysis of nuclear cluster deflections, which was presented in Figure S4, has now been shifted to revised Figure 5. We also extend the analysis to ask whether the nuclear clusters (now called nuclear groups) with greater deflections also have greater loss of nuclei in interphase 14.
- c) Our data on 'area fractions' that was previously presented in Figure S7 is now appended in revised Figure 7.

4) Our prior analyses presented in Figures 3, 6, and 7 are now presented, respectively, as revised Figures 4, 7, and 10. This follows from moving data from Figure S1 to revised Figure 3, from Figure S4 to revised Figure 5, merging Figures 5 and 6 into revised Figure 6, and the addition of new figures with the revision experiments involving the DNA damage reporter (revised Figure 8) and cytoskeletal dynamics during mitosis (revised Figure 9).

5) After shifting data from Figures S1, S4, and S7 to the revised Figures 3, 5, and 7, the remaining supplementary figures are renumbered.

6) Following up on comments from reviewer #2 (minor concerns 2), the figure legends are now presented more succinctly, removing unnecessary discussion of the presented data.

7) Following up on comments from reviewer #1 (minor concern 4), and after adding missing details, the methods section is presented more succinctly. We hope that the language now reads less colloquial.

Reviewer #1 (Remarks to the Author):

Kale et al. describes in detail the deleterious effect of increased temperature during early insect development. Previous work showed robustness of *Drosophila* embryonic development despite fluctuating temperatures. However, here the authors show that the initial stages of syncytial blastoderm is sensitive to high temperature, which causes modified patterns of mitotic division, altered local nuclear crowding and loss of nuclei from the embryo cortex. These are interesting findings worth of being published. After this descriptive portion of the paper, the authors ask about the mechanisms behind this heat response and test the effect of potential candidate genes to rescue those developmental defects. Although their findings here are exciting, the reasoning and hypothesis for the group of genes tested is a bit lacking (specially the mitotic division group), and some of the conclusions are not well based on the results.

Major concerns:

1) For example, the authors propose that high temperatures disrupt protein-protein interactions. Assuming the authors are implying that the protein-protein interactions are affected due to changes in protein structure because of high temperature, how then simply adding more of those proteins would rescue the phenotypes observed? Wouldn't the added proteins still be affected by high temperatures?

We thank the reviewer for the invitation to clarify our thinking. The way we think about the disruption of protein-protein interactions does not necessarily imply changes in protein structure through, e.g., partial or full protein unfolding. In fact, we do not think that our elevated temperatures are high enough to induce such structural changes. Rather, as the temperature rises, we assume that protein-protein interactions would get weaker, as the equilibrium between the bound and dissociated states shifts toward the dissociated state. This is especially relevant for protein-protein interactions that produce a scaffold, with increased dissociation effectively depleting the pool of fully formed scaffolds. We would argue that the scaffold composed of α -Catenin, β -Catenin, and Apc2 (mediated by Sgg) that links cortical F-actin and astral microtubules of the spindle (McCartney et al. 2001) would be vulnerable at elevated temperatures due to the weakening of one or more protein-protein interactions that are responsible for constituting the scaffold. Additionally, there is a striking similarity between the defects in our embryos at elevated temperature and the loss-of-function conditions tested by McCartney et al. 2001. Thus, we expect that our overexpressions (α -Catenin and Shaggy) rescue the defects at elevated temperature, as the equilibrium shifts towards bound states, counteracting the "effective depletion" of this protein scaffold.

We have now updated the Discussion section to elaborate on this view. The relevant section now reads

(Page 15, Lines 624-634): "A mild temperature increase such as the one we studied here is unlikely to denature proteins, but may instead weaken protein-protein interactions, such as the ones between components of the tripartite scaffold, involving α Cat, β Cat, and Apc2 (mediated by the protein kinase sgg). Such weakening would effectively deplete functional protein scaffolds, creating a loss-of-function-like scenario, as the equilibrium shifts towards the dissociated state of the scaffold. ... This interpretation is consistent with the observation that overexpression of either α Cat or sgg rescues these defects (Figure 10A-C), as increasing even a single component should be sufficient to raise overall scaffold abundance and ..."

2) The findings on *sgg* also require more explanation. What sort of allelic variants are being referred to, do they modify coding regions? Which regions on the protein, and what anticipated modifications they may cause to the protein function? Again, how can the authors compare over-expression of *sgg* vs. natural allelic variants and the ability to buffer embryos from the effects of high temperatures?

Our list of genes with natural variation integrates data from studies that differ in resolution (QTLs, SNPs, and candidate gene approaches). Because these datasets span broad genomic regions and likely involve different, independently evolved alleles in each of the three continental populations, we chose “gene” as the common level of abstraction. Consequently, we restrict our question to which genes show consistent clinality across data sets, rather than which specific allelic change (coding vs. non-coding, position in the protein, etc.) are involved. At present, we therefore cannot infer whether the natural variants in *sgg* (or other candidates) are coding or non-coding, nor predict the precise effects on protein structure or function.

Given these limitations, we do not attempt to equate our experimental overexpression of *sgg* with any specific natural allelic variant. Instead, after testing their functional relevance experimentally, we use the clinal signal only as support that these genes are plausible candidates for thermal adaptation. For *sgg*, as well as α -Catenin, β -Catenin, and *Apc2*, previous work (McCartney et al. 2001) shows that loss-of-function or knockdown produces paired nuclear loss similar to what we observe at high temperature in wild-type embryos. This suggests that reduced activity of these genes compromises early development, whereas increased activity is more likely to be protective. Our overexpression experiments, therefore, test the hypothesis that elevated activity can buffer embryos against elevated temperature, without claiming that natural allelic variants act solely through overexpression. The methodological details of our gene-level integration of clinal data are now described explicitly in the Methods (addressing also minor concern 4b).

In the revised manuscript, this clarification is reflected in the following paragraph in the methods section:

(Page 22-23, Lines 935-945): “Analysis of thermoclinal genes: We sought genes showing variation across natural populations of *Drosophila melanogaster* along large-scale climatic gradients. At continental scale, we reasoned that climatic variation imposes selection gradient with distance from equator, generating geographic genomic variation, i.e., a thermocline. Previous studies reported such variation at different scales (whole genome, Quantitative Trait Loci, or individual genes/proteins, see Table S1 for the list of studies). To be able to aggregate over these scales we chose “genes” as a level of abstraction. Consequently, we focus our analysis to genes that show consistent clinality across data sets, rather than to specific allelic change (coding vs. non-coding, position in the protein, etc.) are involved. We thus extracted only genes reported to vary across geographically distant populations, and grouped studies by continent (North America, Europe, or Australasia), assuming these continent-scale gradients arose independently.”

Future work, ideally with whole-genome data from the same populations, will be needed to identify the exact variants in *sgg* and other candidates, determine whether they affect coding or regulatory regions, and test both knockdown and overexpression when the function of a gene is not known during the syncytial-stage. We consider these efforts to be, however, beyond the scope of the current manuscript.

3) The definition of lethality used in the paper appears to be based on larvae eclosion only. Do these larvae survive and reach adult stages? Are the flies fertile? Do they display any kind of phenotypes (locomotor defects, short life span, etc)? I believe these are important questions to be answered when arguing about the deleterious effect of temperature on development. Just eclosion after 24 h is a narrow window. In addition, controls would give a good idea of phenotypes caused by the live-imaging preparation itself.

We thank the reviewer for this suggestion, and we agree that larval eclosion is a minimal definition of survival and does not capture later developmental or adult phenotypes. In this study, we deliberately chose embryo-to-larva survival as a conservative and technically simple read-out that can be applied uniformly, and that should also be transferable to non-model Diptera in future comparative work. Any later-arising defects in viability, behavior, lifespan or fertility would therefore mean that our assay underestimates, rather than overestimates, the deleterious effects of high temperature. We thus believe that analyses of later-arising defects are beyond the scope of the current manuscript.

However, we fully share the reviewer's view that more integrative fitness measures are desirable. In ongoing and future work, we plan to implement rapid larval fitness assays, similar to Lockwood et al. 2017, where the distance of pupation from the food source provides a quantitative proxy for larval performance, and to extend this to adult survival and fertility, where feasible. We also agree that parallel controls for the live-imaging preparation will be informative for distinguishing temperature effects from imaging-induced phenotypes, and we are currently developing such control assays.

To better reflect our reasoning, we have clarified in the manuscript that our current measure is restricted to larval eclosion and that this makes our conclusions conservative with respect to total fitness. The corresponding section in the revised manuscript now reads:

(Page 17-18, Lines 724-727): "Embryo lethality experiment: We chose embryo-to-larva transition as a conservative and technically simple read-out for embryo survival. This excludes any potential later-arising defects in viability, behavior, lifespan or fertility, meaning that our assay underestimates, rather than overestimates, the deleterious effects of elevated temperature."

Minor concerns:

The paper would benefit from the revisions above and additional ones, including the following:

1) The authors should consider moving some of the supplemental figures to the main figures. For instance, the holes and nuclei extrusions are much better seen in Figs S1 and S2.

We appreciate this suggestion. Considering the limitations on the number of main figures, we decided to present only the key information in the main figures. After extending the analysis of nuclear displacements, we have shifted supplemental Figure 4 to the revised Figure 5. Then, we have also appended the plots in Figure S7 to the revised Figure 7, and upgraded Figure S1 to revised Figure 3 after adding more data.

2) Defects in chromosomal segregation are not clear in the figures presented (Fig. 3E).

Thank you for raising this point, and apologies for not having included a better resolved dataset before. To visualize the chromosome segregation defects better, we have now included an additional time point, where the missegregated chromosomes can be seen

clearly. Additional examples of chromosome segregation defects can now be seen in revised Figure 8C-E (arrows). In the text, we now also highlight another hallmark of mitotic failures: a delay in telophase.

(Page 6, Lines 216-217): “Consistent with this idea, we observed chromosome segregation defects, and a delay in telophase, preceding the pairwise loss of nuclei (Figure 4E), at all of our experimental temperatures.”

3) As cellularization proceeds, the nuclei move towards the center of the embryo (e.g. see Keranen et. al, 2006; Fowlkes et al., 2011; Xue et al, 2023) and increase cell density in that area. Thus, it is important to show what stages were analyzed during cycle 14 embryos between control and experimental in terms of membrane invagination, not just timing, since at high temperatures the embryo may develop faster than at low temperatures.

We believe there is a misunderstanding. We quantified nuclear crowding at the end of interphase 13, which is one nuclear division before the onset of cellularization. We observed that there is a loss of nuclei during cycle 14, and this is what we presented in Figure 2. However, we analyze the nuclear density defect at the end of interphase 13, and not during cycle 14, as the nuclear crowding defect exists already during cycle 13 (revised Figure 6), and the crowded region during cycle 13 corresponds to the region of increased nuclear loss during cycle 14 (revised Figure 7). We have now clarified this point in the revised manuscript. Relevant sections now read:

(Page 7, Lines 294-299): “To better understand the differences between stochastic mitotic failures and locally clustered failures, we focused on cases with severe nuclear losses in embryos at 29°C. The most severe loss of nuclei in embryos at 29°C was often observed during cellularization (interphase 14) in the central domain of the embryo (Figure 2, 3 and S1). Thus, we wondered if the central regions of the embryo were somehow special, and if there were characteristics in nuclear distribution or nuclear division, already during nuclear cycle 13, that rendered the central regions to be distinct than the anterior/posterior poles.”

(Page 8, Lines 332-337): “To test whether nuclear crowding and/or nuclear cycle asynchrony resulted in a predisposition towards a local increase in mitotic failure, we first defined the regions of the embryo with high nuclear crowding (Figure 7A) at the end of interphase 13, or regions with substantially delayed nuclear division cycles (Figure 7B) at the end of mitosis 13. In parallel, we marked the locations of mitotic failures occurring during cellularization (interphase 14), which could be overlaid on any predefined region to calculate the fraction of mitotic failures in that region.”

4) Method section is written too colloquially, with unnecessary details and at the same time, missing important information.

a) For example, the precision ratings for temperature of the equipment used is not mentioned (incubators and microscope stage). What is the expected +/- variation in temperature? This is pertinent since the authors claim significant phenotypes caused by 29C and not 28C.

Thank you for pointing this out. The temperature conditions for embryo collection (incubators, bench top, and water bath) are mentioned as a range of temperatures to reflect day-to-day variability. The temperature conditions during imaging were much more precise ($\pm 0.1^\circ\text{C}$). We have clarified this in the revised manuscript in the methods section, where the relevant section now reads:

(Page 18, Lines 749-757): “We collected embryos for 1 hour either at normal temperatures (22-23°C, 24-25°C, or 27-28°C) or elevated temperature (29-30°C). Collections at 22-23°C (22°C imaging) were done in the fly incubator; 24-25°C (25°C imaging) on the bench; 27-28°C (28°C imaging) or 29-30°C (29°C imaging) in a water bath, with embryo collection plate bottom in thermal contact with the heated water. Then, the embryos are bleached for 35-40 seconds, rinsed with tap water at the collection temperature, aligned on a 25 mm diameter coverslip, covered with halocarbon oil, and immediately transferred to the temperature controller setup from Warner Instruments (QR-40LP holder on QE-1HC heater/cooler, coupled to LCS-1 liquid cooling and CL-100 bipolar temperature controller, precision of $\pm 0.1^\circ\text{C}$), pre-equilibrated to the experimental temperatures (22, 25, 28 or 29°C).”

b) Methods of screening for clinal variations is not explained, only described in the supplement Table legends. Types of alleles should be better described.

Thank you for pointing out this blind spot. We have now added a methods section (Analysis of thermoclinal genes) to describe how we work with datasets on clinal variation. The relevant sections read as follows: ...

(Page 22-23, Lines 935-945): “Analysis of thermoclinal genes: We sought genes showing variation across natural populations of *Drosophila melanogaster* along large-scale climatic gradients. At continental scale, we reasoned that climatic variation imposes selection gradient with distance from equator, generating geographic genomic variation, i.e., a thermocline, across a temperature range of at least 15°C. Previous studies reported such variation at different scales (whole genome, Quantitative Trait Loci, or individual genes/proteins, see Table S1 for the list of studies). To be able to aggregate over these scales we chose “genes” as a level of abstraction. Consequently, we focus our analysis to genes that show consistent clinality across data sets, rather than to specific allelic change (coding vs. non-coding, position in the protein, etc.) that are involved. We thus extracted only genes reported to vary across geographically distant populations, and grouped studies by continent (North America, Europe, or Australasia), assuming these continent-scale gradients arose independently.”

c) The section describing the analysis of entire embryos (“visualize all nuclei in the entire embryo”) is not credible for a number of reasons. First, the confocal used will not provide sufficient resolution along the Z axis to allow for segmentation of nuclei along the curvature of the embryo (e.g. see Keranen et al. 2006 methods description). The proper way to do so would be to use a spinning disk or similar microscope. Second, the stitching method does not rely on labeled landmarks in the embryo (e.g. a gene expression pattern), therefore the junction is a guess. This is not to say that an analysis of the entire embryo is necessary, but the paper should be revised and correctly mention that what is being visualized are the embryo top and bottom surfaces and not the entire embryo. If the authors indeed managed to segment the entire embryo and can show landmarks used for stitching, then include the data of all segmented objects that show the expected ~6000 nuclei around the embryo cortex in controls in Figure 2 and contrast that to the experimental embryos.

We apologize for using too bombastic terms and thank the reviewer for pointing this out. It was not our intention to claim that we did an *in toto* analysis of all nuclei in the

embryo. Rather, we wanted to test for potential bilateral symmetry, which we would have considered as evidence for the defects being caused by some (bilaterally symmetrical) patterning along the AP axis. We clarified this in the revised version of the manuscript and have adapted the language accordingly.

(Page 4, Lines 155-160): “Within the coordinate system of the embryo, the distribution of neither isolated subcortical nuclei nor blastoderm holes was left-right symmetric. Additionally, even if blastoderm holes were predominantly observed in the central region of the embryo, their location varied in their position across embryos (Figure 2O, P). These results further support the idea that loss of nuclei and blastoderm holes are unlikely to be directly caused by defective patterning: embryonic patterning is left-right symmetrical and accordingly, observed defects in patterning would likewise show left-right symmetry.”

We do concede that some possibly lax language was used while describing our analysis to “visualize all nuclei in the entire embryo” (now called ‘whole embryo reconstruction of nuclear distribution’). We have revised the text to clarify what we image, how we stitch, and what we do – and do not – claim from these data.

1. What volume is actually imaged?

We do not claim single-nucleus-accurate segmentation over the entire embryo cortex. As now stated in the revised text (see below), our approach is designed to visualize nuclear loss (“holes”) at the blastoderm, not to count all cortical nuclei.

Using a point-scanning confocal, we image >50% of the embryo volume from each side and obtain a substantial overlap between the two image stacks. The z-resolution is indeed lower than in x–y, and we agree that this is insufficient for robust automated segmentation along the curved cortex. We have therefore not based any quantitative nuclear counting on these stitched data.

(Page 20, Lines 815-818): “For whole embryo reconstruction of nuclear distribution, we used a custom-built image analysis pipeline to fuse the two views of the embryo, and visualize subcortical nuclei; either isolated or in clusters. We refer to each view as “half embryo”, even though each spans about 70-75% of the embryo width, providing a substantial overlap.”

2. Why point-scanning confocal, not spinning disk?

We didn’t use a spinning-disk for the simple reason that it was not easily accessible for generating a large dataset, with two views for each of the 145+ embryos (145 embryos across 2 temperatures at cellularization stage + many others (data not shown) during earlier syncytial cycles). We could, in principle, also use a SPIM/multi-view light sheet: such systems are excellent for whole-embryo imaging, and our analysis pipeline indeed resembles stitching a multi-view dataset from a lightsheet/SPIM. However, we aimed to establish a method that uses standard confocal systems, making it broadly accessible.

3. How is stitching performed, and what are the landmarks?

We now clarify that we use nuclear landmarks for alignment: non-cortical/yolk nuclei, which are present throughout the embryo interior, provide robust 3D landmarks that are visible in both opposing views. We first perform landmark-based alignment using

BigWarp with manually defined corresponding nuclear landmarks in the overlap region. We then use BigStitcher to perform intensity-based stitching using the full nuclear stain in the overlap.

(Page 20, Lines 822-827): “We then manually aligned the halves along their AP axis and used the BigWarp⁷¹ within BigDataViewer⁷² to register them using manually selected interest points (subcortical- or yolk-nuclei) visible in both halves. We tightly cropped the output images to reduce the computational load in the steps to follow. Then, we used BigStitcher⁷³ to further align the embryo, using the phase correlation method, to produce the “fused embryo”.”

This procedure is conceptually similar to multi-view registration in light-sheet datasets, but implemented on two opposing confocal stacks. The revised Figure 2 now includes cross-sectional views (orthogonal slices) that illustrate (i) the continuity of the nuclear signal across the stitch and (ii) that the reconstructed volume encompasses the entire embryo, not just the top and bottom surfaces.

4. Why we do not provide a ~6000-nuclei count

We agree that accurate counting of all ~6000 cortical nuclei would be the strongest demonstration of complete coverage; however, as the reviewer notes, segmentation along the entire curvature at this z-resolution is challenging and outside the scope of our current aims. Importantly, however, our biological question in the revised Figure 2 is qualitative: to demonstrate the presence and spatial distribution of blastoderm “holes”. The reconstructed volume is used only to visualize these regions of nuclear loss; all quantitative analyses of mitotic defects and nuclear loss are performed on higher-resolution live-imaging datasets.

Accordingly, we have revised the Methods and Results (1) to replace “visualize all nuclei in the entire embryo” with wording that explicitly states we reconstruct the full embryo volume to visualize nuclear distribution to in turn visualize blastoderm holes, (2) to describe the imaging depth from each side (70-75% embryo width), and the landmark-based alignment and stitching (BigWarp/BigStitcher), and (3) to include the new orthogonal cross-sections in the revised Figure 2 to document stitching accuracy, particularly using yolk nuclei as internal landmarks.

Together, this should make the scope and technical limitations of the whole-embryo visualization clear, while demonstrating that the stitched volumes are reliable for the qualitative purpose they serve in this study.

We hope that the methods section ‘whole embryo reconstruction of nuclear distribution’ now clarifies our view.

5) Reference 57 (Edgar et al.) should be better presented and discussed in the paper in terms of nucleocytoplasmic ratio for mitotic divisions and nuclear density control. I was not able to find this reference being cited in the main text or elsewhere.

We thank the reviewer for pointing out this oversight: this reference was actually only mentioned in the methods section. In general, we had not discussed the literature focusing on the N/C ratio, as we thought that that level of detail might not prove beneficial to simplify the main message of our manuscript. However, encouraged by the reviewers' comments, we

have now added the following text in the discussion section, citing Edgar et al. as reference #26.

(Page 9, Lines 352-361): "... crowding may locally elevate the Nuclear-to-Cytoplasmic (N/C) ratio, a parameter that reflects the amount of cytoplasm available to each nucleus. The N/C ratio is known to regulate the pace of nuclear division cycles, such that higher N/C ratios are associated with slower nuclear division cycle²⁵⁻³⁰, likely due to limited cytoplasmic resources required for the cell cycle. Nuclear crowding would therefore slow nuclear cycle indirectly through its effect on N/C ratio. Consistent with this interpretation, the reduced area of the 'combined regions' at 29°C (Figure 7K) indicates substantial spatial overlap between crowded regions and delayed regions, particularly in embryos exhibiting blastoderm holes. Thus, at elevated temperature, these combined regions likely reflect both the direct mechanical consequences of crowding and its indirect effects mediated through altered nuclear division timing."

(Page 14-15, Lines 600-607): "Our analysis shows that clustering of mitotic failures occurs in embryonic regions that lag behind in the nuclear division cycle and exhibit nuclear crowding (Figure 6). While we treat these factors as independent for most of their analysis, a combination of these factors – captured in 'combined regions' – predispose particular regions of the embryo to increased mitotic failures at 29°C (Figure 7). Thus, our findings are consistent with previous studies, which suggest that they are mechanistically coupled through the N/C ratio²⁵⁻³⁰. These findings suggest a mechanism in which the dependence of nuclear cycle duration on nuclear density could amplify the likelihood for mitotic failure at elevated temperature; whether a similar interplay affects embryonic development under normal temperature conditions remains unclear."

6) Discs large (dlg), not disks large

Noted. We have now corrected this error.

7) Consideration about historic high Earth temperatures in insect evolution could be discussed.

We have now added a few lines in the discussion section on this. In our reading of the literature, there seems to be a relationship between periods of cooler climate and bursts of diversification, and these cooler periods follow warmer periods when biodiversity was lost. This could hint towards a correlation between warming and species going extinct. However, there seem to be additional factors, such as volcanic eruptions and meteorite impacts, that show stronger causality with (mass) extinction events, blurring the correlation with warming events. Given these caveats, we aim to keep the statements speculative.

(Page 16, Lines 676-681): "Reports of contemporary insect biodiversity loss^{64,65} resonate with patterns of extinction and diversification during past climatic fluctuations^{66,67}. Although multiple abiotic factors contribute to these historical trends⁶⁸⁻⁷⁰, the current rapid rise in global temperature highlights heat stress as a unique relevant driver. Limited data make it difficult to test whether past warming events directly drove innovation in blastoderm architecture, but it is intriguing to speculate that early developmental processes may serve as both vulnerabilities and substrates for evolutionary change."

Reviewer #2 (Remarks to the Author):

Kale et al describe the effects of a temperature shifts on *Drosophila* embryogenesis and characterize several types of common failures in development at high temperature. After a careful and detailed description of the changes in cell divisions and localization of nuclei, they test and identify a few candidate genes that can rescue the lethality and mitotic defects. They suggest this may relate to genes that vary in natural populations that are linked to environmental temperature. Overall, the paper explores a very interesting premise that could have major implications, and some of the imaging is very nicely presented and the genetic results are intriguing. However, some of the data is too weak or difficult to interpret as described, and so some conclusions are not yet sufficiently supported.

Major concerns:

1) The title of the paper refers to the fatal disruption of nuclear division, but it is not clearly demonstrated that nuclear division itself is faulty, nor are changes in the nuclear position shown to be causative to the fatality (although it may correlate – this is also not directly shown).

This is a fair point, and probably is also partly reflected in the comment from reviewer #3 (major comment 1). However, with our experiments on DNA damage, we are adding a crucial mechanistic detail. Our analysis of cytoskeletal dynamics (F-actin and microtubules) during mitosis strongly suggests a defect in their interaction: an interaction that we think is strengthened when we overexpress α -catenin or shaggy, as these overexpressions also rescue embryo lethality. While we think that our original title is now better supported, we would also be happy to go ahead with the title below if the reviewer so wishes.

Elevated temperature compromises nuclear divisions in early development, leading to increased lethality in the *Drosophila* embryo.

2) There is a nice, detailed analysis of the nuclear divisions and the movement of nuclei away from the cortex in Figure 3. At some point in the text, it seems that any nuclei that are subcortical are classified as “mitotic failures” but I do not understand why. Is there a marker that indicates mitotic failure or incomplete nuclear division?

We understand the confusion, and you are correct: in the initial manuscript, we have not used any objective marker to identify mitotic failures (we were using chromosome segregation defects, followed by a pair-wise loss of nuclei from the embryo cortex, as the indicator of mitotic failure). During revisions, however, we further analysed DNA damage response, which now strengthens the connection between subcortical pairs of nuclei and mitotic failures by adding an important mechanistic detail. (see also comment by reviewer 3 (major comment 1), as well as the added revised figure 8, and result section ‘Mitotic failures trigger DNA damage response’).

3) On line 197, the authors say “We observed chromosome segregation defects preceding the pairwise loss of nuclei (Figure 3E), at all our experimental temperatures.” This figure shows Histones/nuclei and depth of nuclei (magenta), but there is not enough detail to show a chromosomal segregation defect. They conclude later (line 445) “mitosis itself is vulnerable at elevated temperatures.” From the images, it looks like mitosis completes normally but then some nuclei are pushed out afterwards, which seems like a distinct type of defect. In my opinion, a mitotic failure needs to be shown more directly.

We thank the reviewer for this suggestion. To better demonstrate the chromosome segregation defects, we have now included an additional time point, where the missegregated chromosomes can be seen more clearly. Apologies for not having included this time point before. Additionally, more examples of chromosome segregation defects can now be seen in revised Figure 8C-E (arrows). In the text, we now also highlight another hallmark of mitotic failures, a delay in telophase, which implies a defective mitosis.

(Page 6, Lines 216-217): “Consistent with this idea, we observed chromosome segregation defects, and a delay in telophase, preceding the pairwise loss of nuclei (Figure 4E), at all of our experimental temperatures.”

4) Also, are the embryos with more displaced nuclei more likely to die? The only link they have between these is the similar frequencies of occurrence, which is a correlation at best.

We completely agree with the reviewer that what we are demonstrating is only a correlation between the presence of blastoderm holes and embryo lethality, given that both are caused by elevated temperature at the same rate of incidence. While we believe a causal link exists and that our results from the genetic rescue experiments indirectly support it, we are yet to demonstrate it conclusively. Now we clarify this point in the discussion. The relevant text reads:

(Page 14, Lines 585-590): “Our work demonstrates that processes of syncytial blastoderm formation and cellularization in *Drosophila* embryonic development are vulnerable to elevated temperature. Embryos exposed to 29°C during these stages exhibit defects in maintaining the bilateral symmetry of developmental processes, and in extreme cases, enter a lethal trajectory during gastrulation (Figure 1). We argue that these extreme cases of developmental failure likely coincide with structural defects in the blastoderm epithelium, which manifest as regions devoid of nuclei/cells and discontinuities in this tissue (Figure 2, 3 and S1).”

5a) It is certainly interesting that the lethality and some of the nuclear “mitotic failures” are shown to be rescued by the overexpression of *sgg* or *a-catenin*. However, it is an oversimplification to say this works through cytoskeletal/polarity components since these are known to signal in major signaling pathways (eg, Wg and Hippo pathways) and can affect many things, not just actin. It is not appropriate to conclude, as the authors do in the abstract that, “Our genetic rescue experiments show that in *Drosophila* embryos, the interaction between cortical F-actin and microtubules is vulnerable to disruption at elevated temperatures.” [Or that (line 397) “These results indicate that the experimental strengthening of interaction between microtubule and F-actin is able to attenuate temperature-dependent destabilizations of the mitotic spindle.” Or that (line 434) “Failing interactions between microtubules and F-actin make early fly development susceptible to elevated temperature”.] To make these conclusions, additional components would need to be tested (eg disrupting or overexpressing actin or tubulin or their direct regulators) in the different temperature conditions.

We appreciate the reviewer's suggestion and helping us identify claims that are not fully supported yet (and which are in fact difficult to demonstrate, as others have previously failed to look into this). We acknowledge that there might be a contribution from the wingless and hippo pathways, especially during gastrulation stages, where our overexpression is likely active as well, and, in principle, they can affect many things beyond F-actin. But in our

context, the experiments could be explained by the disruption of the interaction between microtubules and F-actin.

Additionally, we have now included new analyses of mitotic spindles and F-actin caps, which indicate that the spindles are weakly anchored to the actin caps at elevated temperature. Although post-hoc, this analysis indicates the necessity for strengthening the interaction between cortical F-actin and astral microtubules of the spindle during mitosis: a function associated with the tripartite protein complex of α -Catenin, β -Catenin, and Apc2 (mediated by Sgg).

That being said, it is clear that our assertions about the vulnerability of interaction between F-actin and microtubules were premature, and we thank the reviewer for demanding a higher standard for evidence to support our statements. The following instance reflects our change of tone in the revised manuscript:

(Page 13, Lines 547-551): “These results suggest that overexpression either α Cat or sgg strengthen the interaction between microtubule and F-actin, which in turn attenuates temperature-dependent destabilizations of the mitotic spindle. We therefore propose that the scaffolding activity of α Cat, β Cat, and Apc2, mediated by sgg, constitutes a thermosensitive controller of nuclear division fidelity in the syncytial *Drosophila* embryo.”

5b) They could at least test their model further and validate their arguments by looking at apical F-actin caps directly when sgg or a-catenin are up or down regulated at different temperatures.

We thank the reviewer for this suggestion. To assess the interaction between F-actin and microtubules during mitosis, we aimed to visualize the F-actin distribution alongside nuclei and/or microtubules at different temperatures.

Unfortunately, expression of the GFP-tagged Utrabin Actin-Binding Domain (UtrABD-GFP) to visualize F-actin seems to rescue the defects normally observed at elevated temperature: the frequency of mitotic failures is reduced, and blastoderm holes are gone. We believe that this is due to the F-actin-stabilizing side-effect of expressing this reporter, and we aim to test this more formally in future work. However, for the current manuscript, this precludes the possibility of testing whether α -Catenin or sgg overexpression affects F-actin dynamics.

6) Although p-values are provided for quantitative data, the specific statistical tests performed are not described, making it difficult to assess if the comparisons are appropriate. (This is especially a concern for the fractions in Fig 6, for which a statistical test should be accounting for lower incident rates in certain conditions.)

Thanks for pointing out this missing detail. For all p-values, we are using non-parametric statistical tests (non-parametric Kruskal-Wallis test with Dunn’s uncorrected test for multiple comparisons). This detail is now added in the ‘Image Analysis and Statistics’ section in methods, and in figure legends wherever relevant.

7a) Text area beginning on line 137 and related Figure 2 discusses “blastoderm holes”, also sometimes called “gaps” or “subcortical nuclei” in the text (and the switches in nomenclature can be confusing). The terms refer to nuclei that move away from the cortex of the embryo.

We thank the reviewer for pointing out the variation in nomenclature. We have now rephrased the text, clearly defined ‘blastoderm holes’, and are consistently using ‘blastoderm holes’ for a clustered loss of nuclei.

(Page 4, Lines 153-154): "...in a small fraction of embryos developing at 29°C, we observed a notable void in the cortical layer of nuclei (henceforth, blastoderm hole), coinciding with a subcortical cluster of at least 10 nuclei."

7b) This is reported to occur in 12% of embryos but it seems like only two cases are shown in Figure 2D/E (out of 77, I think), making it hard to evaluate any potential patterns to the depth of the loss or the stated L/R asymmetry. More data needs to be shown or quantified for this to be convincing- perhaps they can quantify the fraction of yellow/magenta pixels/voxels for different regions of multiple embryos and present this more quantitatively. As it stands, it looks like there is randomness in the "holes" and while there is more on the left side of one example, it is hard to know if that is just true for this case, or why it is even an important point if it is random.

We apologize for the poor phrasing that seems to have led to a misunderstanding: it was not our intention to make a statement about L/R asymmetry. What we wanted to say is exactly what the reviewer points out: that it looks like there is randomness in the holes. Our argument is that if it is random, it is unlikely to be associated with patterning. We have now revised the text to clarify this point (see below).

Further, in the revised Figure 2O-P, we also present the distribution of blastoderm holes in the 11 embryos that have them. As we can see in these pictures, the blastoderm holes are distributed rather randomly across the embryonic midline (no left-right pattern), while occurring more frequently in the central region of the embryo. Thank you for prompting this analysis: it certainly adds a level of clarity to the manuscript. The revised section now reads:

(Page 4, Lines 155-160): "Within the coordinate system of the embryo, the distribution of neither isolated subcortical nuclei nor blastoderm holes was left-right symmetric. Additionally, even if blastoderm holes were predominantly observed in the central region of the embryo, their location varied in their position across embryos (Figure 2O, P). These results further support the idea that loss of nuclei and blastoderm holes are unlikely to be directly caused by defective patterning: embryonic patterning is left-right symmetrical and accordingly, observed defects in patterning would likewise show left-right symmetry."

7c) Additionally, the embryo in E looks to be slightly older with some head involution starting- are these all staged together?

The apparent structures in E are likely caused by inhomogeneities in nuclear density; we do not observe actual folds on the embryo surface in any region. If these were true tissue folds in the head region due to the embryo being older, we would also expect to see deep folds in the ventral region associated with ventral furrow invagination. To avoid such ambiguity, we did not image embryos that showed tissue folds related to the onset of gastrulation movements. Then, among the imaged embryos, we categorised embryos by nuclear density to identify those in the cellularization stage. We further refined this classification using trans-illumination images (data not shown), in which the extent of cellularization can be readily assessed. The embryos shown in C-E were selected because all were determined to be in mid-cellularization based on the trans-illumination images. We have now updated the figure legend to clearly mention that these are indeed mid-cellularization embryos

(Page 24, Lines 982-984): "(C-H) Distribution of nuclei in embryos during mid-cellularization (interphase 14), in fixed embryos that were developing at normal temperature (C, F) or elevated temperature that shows either a mild defect (D, G) or a severe defect (E, H)."

8) Claims that some of the observed defects are “a function of temperature” are overstated; for example, in figure 4C, asynchrony is only observed at 29C, not increasing with each temperature difference. In fig 5M, 22C shows differences from 28C or 29C, and Fig S6 only two temperatures are shown. So, the defects should be attributed to the high temperature only, not as a “function of”, which implies it’s gradually shifting.

We thank the reviewer for pointing this out. We have now rephrased all occurrences of function-of-temperature, and simply attribute the effect/defect to an exposure to elevated temperature.

9) It is very interesting to consider how certain genes are affected by temperature, and there are major implication for this idea. However, it is not very clear how they identified the natural variants or where the temperature-dependent clinal scores come from over the large continents analyzed. More detail about this, such as the range of temperatures considered for the natural variants, would be helpful. Why did they combine the “spindle dynamics” category- do they not see overrepresentation in the individual categories of polarity, spindle localization, and spindle organization?

Our list of genes with natural variation is compiled from multiple population-genetic studies that used different approaches (QTL mapping, SNP scans, and candidate-gene analyses). Rather than reproducing all dataset-specific details, we abstracted across studies and extracted only those genes consistently identified as clinal with respect to temperature across continents. We now describe the underlying datasets, including the temperature ranges and references, in the Methods.

(Page 22-23, Lines 935-945): “Analysis of thermoclinal genes: We sought genes showing variation across natural populations of *Drosophila melanogaster* along large-scale climatic gradients. At continental scale, we reasoned that climatic variation imposes selection gradient with distance from equator, generating geographic genomic variation, i.e., a thermocline, across a temperature range of at least 15°C. Previous studies reported such variation at different scales (whole genome, Quantitative Trait Loci, or individual genes/proteins, see Table S1 for the list of studies). To be able to aggregate over these scales we chose “genes” as a level of abstraction. Consequently, we focus our analysis to genes that show consistent clinality across data sets, rather than to specific allelic change (coding vs. non-coding, position in the protein, etc.) that are involved. We thus extracted only genes reported to vary across geographically distant populations, and grouped studies by continent (North America, Europe, or Australasia), assuming these continent-scale gradients arose independently.”

Regarding the “spindle dynamics” category, our goal was to evaluate enrichment for the specific genes we tested experimentally (dlg, baz, aCat, sgg, mud, insc) under a single functional umbrella. No sufficiently specific single GO term captured all of these genes, whereas very broad terms (e.g., “cellular process”) were not informative. We therefore combined three related GO terms - cell polarity, spindle localization, and spindle organization - into one functional group, which we refer to as “spindle dynamics.” Of the three genes we identify as thermoclinal, all are annotated for cell polarity, one for spindle organization, and none for spindle localization; given this pleiotropy and partial overlap, we consider the combined category more meaningful than treating each GO term separately.

Minor concerns:

1) For people not familiar with flies, the different temperatures discussed in the introduction are not explained very clearly- it is not obvious that flies are often raised at 25C or 22C or that 29C is hot.

Thank you for highlighting this blind spot. We have added this detail in the results section.

(Page 2, Lines 64-67): With a transition from 28°C to 30°C, *Drosophila* no longer exhibit an acceleration of embryonic development¹⁰⁻¹², suggesting that 29°C marks a threshold at which elevated temperatures challenge development and exceeds the physiological range over which the Arrhenius equation accurately describes developmental rates, while lower temperatures remain effectively normal.”

(Page 4, Lines 166-169): “To determine whether the nuclear loss at 29°C reflects a threshold response associated with a transition to non-Arrhenius behavior, we examined temperature dependencies additionally at 22°C, 25°C, and 28°C, temperatures within the normal physiological range.”

2) There is too much explanation in the figure legends (for example figure 3 legend that argues the statistical significance is not meaningful- this should be in the results or discussion).

Thank you for pointing this out. We have now rephrased the figure legends to make them more succinct and moved the arguments to results/discussion as appropriate.

3) Partway through the paper the authors start looking at different temperature than in the initial characterization, which raises the question of what the differences are between 22 and 25 in terms of lethality, blastoderm holes, etc.?

Thank you for pointing this out. We changed the “normal” control temperature from 25 °C to 22°C only for the candidate gene overexpression experiments. Based on our current data, we do not expect qualitative differences between 22 °C and 25 °C in lethality, blastoderm holes, or other defects, and in principle, the overexpression experiments could also have been performed at 25 °C.

Our motivation was pragmatic: overexpression lines carry additional transgenic insertions (UAS and Gal4), and it is a shared experience in the fly community that these additional transgenic insertions render the fly lines “sicker”. Consistent with this, we occasionally observed slightly elevated lethality in the outcross controls at 25 °C as compared to OregonR, with the upper 95% confidence interval on the median being higher (see adjoining plot). To minimize stress due to genetic background that could be a potential confounding factor, we opted to perform live-imaging overexpression experiments at 22 °C – one of the temperatures we used in previous analyses (see revised Figure 3-7). Of note, our overexpressions rescue the developmental defects and embryo lethality, despite the “sicker” genetic background and the elevated temperature.

Our decision to reduce experimental temperature is also consistent with a commonly applied thumb-rule for fly culture keeping: “sick” lines in a fly lab are maintained at a lower temperature (18°C). The lower temperature presumably helps, as the overall life cycle progression is slower than that at 25°C (about twice as slow at 18°C). It has also been noted that certain null mutants, while severely compromising development and survival at 25°C, are homozygously viable at lower temperatures.

We now clarify this change in temperature and our rationale in the text, which now reads:

(Page 22, Lines 908-912): “There was a marginal increase in embryo lethality at 25°C in the controls for the over-expression experiments, which presumably happens due to the presence of additional transgenic insertions (UAS and Gal4). Due to this reason, to err on the side of caution, we chose to perform the imaging at 22°C, specifically for the ‘fraction of mitotic failures’ analysis. As such we don’t expect any biologically significant difference between results at 22°C and 25°C terms of in embryo lethality, blastoderm holes, or other defects.”

Also, as mentioned in response to minor concern 1, we also highlight that both 22°C and 25°C are normal developmental temperatures, further implying that for all instances and purposes, the development at these temperatures is comparable, and differences, if any, are biologically insignificant.

4) Figure 5A, why does the region of nuclei considered not include many at the poles? Does this introduce a bias in the size?

The imaging for this analysis doesn’t include more nuclei towards the poles, as the curvature of the embryo can introduce a bias in the estimation of the pseudo-cell areas, which are calculated in the X-Y plane. Thus, we only focus on the region of the embryo that is closest to the coverslip for the sake of simplicity.

5) The rescue results with the polarity protein Baz and Dlg are hard to interpret given that the same-genotype controls at a low temperature also showed high levels of mitotic failures, but the transgene is likely upregulated at the higher temperature due to effects on Gal4. Thus, the temperature dependent effect is not clear. Similarly, results for Mud overexpression are difficult to interpret.

We agree that these overexpression results are difficult to interpret. We think the mild rescue of embryo lethality by Dlg and Baz overexpression most likely reflects their roles later in development. Because we drive Gal4 with the maternal- α -tubulin promoter, overexpression is certainly active during cellularization and gastrulation stages, i.e., the stages at which embryos fail to develop at elevated temperature and where improved robustness of cell polarity could provide partial rescue. In the case of Mud, the data suggest that its level may need to be more tightly controlled: strong overexpression via maternal- α -tubulin-Gal4 may itself be deleterious, and a weaker driver might be required to test for a true rescue effect. However, we consider this to be outside the scope of this manuscript.

Regarding the concern about temperature effects on Gal4: In our design, Gal4 expression and the resulting transgene mRNA are entirely maternal, given that we are using the maternal- α -tubulin promoter. By virtue of our experimental design, we don’t expose adults to elevated temperature, but only the embryo: only the agar plate, on which the embryos are laid, is at elevated temperature. It is certainly possible that the Gal4 protein that is loaded into the embryo will trigger transcription once the zygotic genome is activated, and might be subject

to differences in experimental temperatures. However, the maternal load of the overexpressed protein is going to be so high that the zygotically produced protein will not affect the analysis. In future experiments, we aim to test two approaches to remedy this problem: 1) introduce paternal Gal80 (Gal4 inhibitory protein) to counteract the zygotic transcription induced by maternally loaded Gal4, and 2) identify and use a weaker maternal Gal4 driver.

6) The manuscript might also improve overall if it were a bit more focused with less supplemental data.

Thank you for this suggestion. After extending the analysis of nuclear displacements, we have shifted supplemental Figure 4 to the revised Figure 4. Then, we have also appended the supplemental Figure 7 to the revised Figure 7, and upgraded Figure S1 to revised Figure 3 after adding more data.

7) There are numerous typos in the main text. Generally there is room for improvement in how the data is represented.

We hope the revised version now reads better.

Reviewer #3 (Remarks to the Author):

This manuscript by Kale et al addresses the mechanisms that impact embryo development at elevated temperature in *Drosophila*. This is an interesting problem, as the impact of temperature on development remains poorly understood and is a question of increasing relevance in the face of global warming. The authors find that the vulnerability of embryonic development to an increased temperature (29 C vs 25 C) is mainly due to defects in the blastoderm divisions. This is an interesting observation as these divisions show specialized control mechanisms to ensure rapid cleavage cycles. The authors then perform a detailed analysis of these divisions and provide insights on the cause of developmental defects, by linking them to mitotic defects. Most notably, by genetic manipulation of factors involved in cell polarity and spindle regulation they can rescue some of the observed defects. They also show that in species that naturally live in increased temperature conditions, the expression of the same factor(s) is similarly up-regulated possibly identifying a mechanism for increase robustness at elevated temperature. This last experimental result represents the main finding of the paper in my opinion and could be of general interest. Overall, the paper is well-written and the conclusions supported. A major area where this paper needs improvement is in relating the authors' findings with relevant literature. It seems that the authors are not aware of or have ignored a large body of literature on the regulation of cleavage divisions in *Drosophila*.

Major comments:

1) Previous work has linked the loss of nuclei to the activation of the DNA replication checkpoint, Chk2 in particular (see for example Lampietro et al Dev Cell 2014). It seems important to analyze that possibility here. While it is clear that lost nuclei happen in pairs (thus likely due to some mitotic defects), it can't be ruled out that these mitoses were defective due to previous DNA damage and activation of the checkpoint. The results of this analysis would be interesting regardless and would add to the paper.

We thank the reviewer for this excellent suggestion! We have now included additional experiments, where we use nuclear SLBP-GFP depletion as a marker for DNA damage, based on the results in Lampietro et al. 2014. With these experiments, we conclude that, indeed, mitotic failures lead to DNA damage, and that nuclei with DNA damage are mitotically arrested and are expelled from blastoderm epithelium. The data is presented in revised Figure 8.

2a) There is clearly a connection between nuclear density and mitotic duration in early fly embryos. However, in their analysis the authors treat the two variables as separate, which is confusing and inaccurate. The impact of DNA content (or expressed genes) on cell cycle duration (Blythe and Wieschaus 2015, Deneke et al 2016) and the importance of uniform nuclear positioning for mitotic synchrony are clearly documented (Deneke et al 2019), as well as how inhomogeneities in nuclear density can affect the collective decision of nuclei to arrest the cell cycle at the maternal-to-zygotic transition (Hayden et al 2022). Strangely, these results are not mentioned, and it is not proposed that the increased cell cycle asynchrony could be due to the improper nuclear positioning.

We thank the reviewer for this summary of the literature. We are aware of the link between nuclear density and the duration of nuclear division cycles, and indeed, our experiments don't reject that possibility. In the initial manuscript, we decided not to include this line of thinking for two reasons.

- 1) Both of these defects, i.e., nuclear crowding and nuclear cycle asynchrony, can be independently linked to developmental defects: nuclear crowding could lead to misinterpretation of the positional information (which is seemingly refractory to temperature fluctuations), leading to defects in cell fate specification; while nuclear cycle asynchrony can further lead to asynchronous cellularization, and disrupt the coordination of developmental processes in distant parts of the embryo. Thus, we decided to keep their analyses independent.
- 2) The link between nuclear density and the rate of nuclear division cycles via the N/C ratio has been established mostly at the level of an entire embryo. In other words, the literature – notably Edgar, Kiehle, Schaubiger (1986), Blythe and Wieschaus (2015), and Deneke et al. 2016 – so far mainly links the changes in the duration of a nuclear cycle with the nuclear density in the whole embryo. As the reviewer points out, Hayden et al. (2022) demonstrated a link between the gradient of nuclear density and nuclear division cycles under mutant conditions. As far as we understand the literature, it is yet to be demonstrated, in wild-type embryos, that inhomogeneities in nuclear density lead to inhomogeneous phases of nuclear division cycles. We believe this to be the case, given our results. However, in our analysis, a strong link between nuclear density and the duration of nuclear cycles seemingly exists only at 29°C: the link being strongest in embryos with blastoderm holes. This temperature dependence made us question the importance of this link towards understanding the temperature vulnerability, especially when contrasted with, for example, mitotic failures that exist at all temperatures, and not just in embryos with a blastoderm hole at 29°C.

Given the interest from reviewers, we have now brought up the connection between nuclear density and nuclear cycle durations in the results and have extended our discussion section to put our results in the context of the literature related to the N/C ratio. Specifically, the relevant section now reads:

(Page 9, Lines 352-361): "... crowding may locally elevate the Nuclear-to-Cytoplasmic (N/C) ratio, a parameter that reflects the amount of cytoplasm available to each nucleus. The N/C ratio is known to regulate the pace of nuclear division cycles, such that higher N/C ratios are associated with slower nuclear division cycle²⁵⁻³⁰, likely due to limited cytoplasmic resources required for the cell cycle. Nuclear crowding would therefore slow nuclear cycle indirectly through its effect on N/C ratio. Consistent with this interpretation, the reduced area of the 'combined regions' at 29°C (Figure 7K) indicates substantial spatial overlap between crowded regions and delayed regions, particularly in embryos exhibiting blastoderm holes. Thus, at elevated temperature, these combined regions likely reflect both the direct mechanical consequences of crowding and its indirect effects mediated through altered nuclear division timing."

(Page 14-15, Lines 600-607): "Our analysis shows that clustering of mitotic failures occurs in embryonic regions that lag behind in the nuclear division cycle and exhibit nuclear crowding (Figure 6). While we treat these factors as independent for most of their analysis, a combination of these factors – captured in 'combined regions' – predispose particular regions of the embryo to increased mitotic failures at 29°C (Figure 7). Thus, our findings are consistent with previous studies, which suggest that they are mechanistically coupled through the N/C ratio²⁵⁻³⁰. These findings suggest a mechanism in which the dependence of nuclear cycle duration on nuclear density could amplify the likelihood for mitotic failure at elevated temperature; whether a similar interplay affects embryonic development under normal temperature conditions remains unclear."

2b) Related to this, previous defects in nuclear positioning have been shown to be due to impaired cortical contractions in the pre-blastoderm stages when these contractions generate cytoplasmic flows that move the nuclei along the AP axis (Deneke et al 2019). Again, strangely there is no mention of this nor the possibility that temperature might impact an earlier morphogenetic process is analyzed. While an analysis of this process would be interesting and doable and add significantly to the paper, the authors should at least discuss it. In this version, there is essentially no understanding of the major process driving mitotic and developmental defects, that is nuclear crowding.

Thank you for pointing out this gap in our description of the defects at elevated temperature. The movement of nuclei along the AP axis, such as that described in Deneke et al. 2019, is seemingly normal at 29°C. When we quantify nuclear crowding at the end of interphase 10 and 11, we don't see any difference across 22, 25, 28, or 29°C. However, we don't present this data, as it is not central to the conclusions of the manuscript. We have now added a sentence to clarify the lack of difference in the cortical arrival of nuclei.

(Page 5, Lines 170-172): "The cortical arrival of nuclei during interphase 10, from the yolk to the embryo cortex, was comparable between embryos developing at 22°C, 25°C, 28°C and 29°C (Figure 3A-E, top panels), indicating that the preblastoderm stage spreading of nuclei along the AP axis²² was comparable between these temperatures."

3) Schubiger's lab performed a screen of mitotic defects and rescues in embryos with increasing cyclin B copies (Ji et al Genetics 2002) finding factors similar to those identified by this paper. Importantly, in these genetic conditions similar nuclear localization defects were also observed so it seems important to discuss this paper, as there are possible links to the authors' findings.

We thank the reviewer for highlighting this article, which has an interesting genetic analysis of nuclear localization defects in embryos with increased maternal cyclin B. While the temperature dependency of cyclin B activity remains to be tested, the results of the study, highlighting the role of *Arp87C*, are certainly relevant to our analysis, given its role as an F-actin - microtubule interacting gene. As indicated in the study, *Arp87C* activity is supposed to restrict the length of astral microtubules, and it seems to be the case that this is a bottleneck at elevated temperature, especially in light of the analysis in the revised Figure 8. Following the suggestion, this study is now included in the discussion section.

(Page 15, Lines 635-639): “Independent support for this mechanistic framework comes from previous work in a different context. The nuclear crowding defect we report here is reminiscent of a previous study, focusing on embryos with increased maternal cyclin B, though at normal temperatures⁴¹. While the impact of elevated temperature on cyclin B activity remains to be characterized, this study found that genetic perturbations enhancing microtubule-F-actin interactions suppress the phenotype.”

Minor comments:

1) In several places, the description of morphogenesis of pre-blastoderm embryos is strangely inaccurate. Nuclei do not migrate from the interior to the cortex in interphase of cycle 10, but they move towards the cortex in a step-like manner from cycle 7-8 to cycle 10 (Foe and Alberts 1983 and Hur et al BiorXiv 2023 for a rigorous, quantitative analysis). It would be appropriate to describe these processes more accurately.

We thank the reviewer for identifying this blind spot in our description of the pre-blastoderm development. Our aim has been to focus on the anchorage of nuclei to the embryonic cortex, which first happens during interphase 10. We have now rephrased the following sentences in the introduction and results. We hope the new version reflects our focus better: the anchorage of nuclei to the embryo cortex and the nuclear cycles that follow.

(Page 2, Lines 73-75): “Following fertilization, the embryo undergoes nine rounds of syncytial nuclear divisions within the yolk. Following these divisions, most nuclei are anchored to the periphery of the embryo, where four more rounds of nuclear division take place during a stage called syncytial blastoderm¹⁴.”

(Page 4, Lines 163-166): “Starting with the first appearance of nuclei at the embryo cortex (occurring during the interphase of the 10th nuclear cycle), we observed nuclei through their cortical divisions during syncytial blastoderm (through mitoses 10 to 13), then through cellularization (interphase 14), and until the onset of gastrulation.”

Point-by-point response to reviewer comments:

We sincerely appreciate the efforts of the reviewers and are happy about the overall positive reception of the manuscript. Below we address a few last points raised by reviewers: our point-by-point responses are in orange ink along with quoted text in the revised manuscript in purple ink. We have split and numbered the reviewer comments, as necessary, and answered them as such.

Reviewer #1 (Remarks to the Author):

The authors made a series of additional experiments in the paper, reorganized main and supplemental figures, and made significant text editing that improved the manuscript.

We thank the reviewer for this summary of the revision, and we are glad that the reviewer appreciates the improvements in the manuscript.

1) Concern # 3 "as cellularization proceeds, the nuclei move towards the center of the embryo (e.g. see Keranen et. al, 2006; Fowlkes et al., 2011; Xue et al, 2023) and increase cell density in that area. Thus, it is important to show what stages were analyzed during cycle 14 embryos between control and experimental in terms of membrane invagination, not just timing, since at high temperatures the embryo may develop faster than at low temperatures."

The authors did not adequately address this point. The explanation given is that the analysis of nuclear crowding was done in cycle 13, and does not refer to cycle 14. However, the distribution of nuclei in wild type embryos (developing at normal temperature) is not uniform at the onset of cellularization, which means that the nuclei distribution is already asymmetric at cycle 13 regardless of temperature.

We thank the reviewer for further clarification of their previous concern and apologize for not having been able to fully address it in our previous revision. We fully agree with the reviewer that the distribution of nuclei in the *Drosophila* embryo is non-uniform, in particular along the anterior-to-posterior axis due to bicoid activity (Blankenship and Wieschaus 2001), and, as the reviewer points out, along the dorsal-to-ventral axis as shown by Keranen et. al, 2006 and Fowlkes et al., 2011. To account for the non-uniform distribution of nuclei, embryos were mounted laterally and measurements were taken consistently within the future trunk ectoderm domain. We have updated our methods section accordingly, which now reads: "... aligned on a 25 mm diameter coverslip with the central part of the embryo facing the coverslip on lateral or ventrolateral region, ...".

2) Second, several of the figures in the paper actually point to the nuclear holes/extrusion phenotypes in stage 14 (eg fig 2, 3, S1, S2,etc), where the size of holes could be amplified in low density areas (the ventral side), whereas nuclear extrusions are more common in high density areas (the dorsal side; this is actually seen in wild type embryos in normal temperatures). Fig 2 indeed shows this effect, and the authors do not have an explanation for the differences seen between dorsal and ventral sides.

We thank the reviewer for raising this important point regarding potential dorsal–ventral differences in nuclear density and their impact on the observed phenotypes. We agree that, in principle, the extrusion of a similar number of nuclei in regions of differing local density could result in differences in the apparent size of blastoderm holes.

In our dataset, however, nuclear extrusion events in wild-type embryos at normal temperature are rare, and we do not observe clear evidence that these isolated events give rise to bona fide blastoderm holes. Consistent with the reviewer's observation, we do note a slight imbalance in the spatial distribution of holes (e.g., dorsal vs. ventral: 7 vs. 4 events; central regions: 5 vs. 4 events). However, given the low overall frequency of these events, we do not consider these differences to be statistically robust, and they are consistent with stochastic variation.

Importantly, our main conclusions are based on comparisons between control and experimental conditions within the same spatial domain of the embryo (see Methods), which minimizes potential biases arising from global dorsal–ventral differences in nuclear density. While we cannot formally exclude a contribution of local nuclear density to the spatial distribution or morphology of blastoderm holes, a rigorous analysis of dorsal–ventral asymmetries would require substantially larger datasets and dedicated spatial quantification, which, we would like to argue, is beyond the scope of the present study.

3) Finally, F-actin apical caps, and adherens junctions, which interact with beta-catenin and *sgg*, are also involved in the cell constriction and cell density distribution, which again is likely to amplify the effects seen at high temperatures, decrease cell contacts and push cells inside the embryo. Thus, the effects of asymmetric cell densities in cycle 13 and 14 should not be ignored as a contributing factor.

Again, we apologize for not having seen the argument clearer before: as we address in 1, we controlled for putative differences in spatial distribution by consistently sampling at the same area of the embryo. Regarding contributions of “F-actin apical caps, and adherens junctions [... to] cell constriction and cell density distribution”, we do not see that our experiments have sufficient resolution to be able to conclusively comment on how adherens junction components might affect or modulate nuclear density distribution at elevated temperature. This limitation applies in particular to interphase 13, where in the absence of cells it appears unclear for how adherens junctions could regulate cell constriction. Against this background, we feel unable to address this point in the current manuscript.

4) For the thermocline analysis explanation, the authors need to write a short clarification about latitude vs. altitude in the datasets, as a high altitude location may have lower temperatures even if closer to the equator than low altitude regions.

We thank the reviewer for this helpful suggestion and agree that altitude can significantly influence local temperature, such that high-altitude populations near the equator may experience environmental conditions comparable to those at higher latitudes.

Our analysis focuses on identifying genetic variation associated with broad latitudinal temperature clines, and therefore does not explicitly account for altitude as an independent variable. As a result, some populations may deviate from the expected temperature–latitude relationship, potentially reducing the sensitivity of our approach to detect all relevant candidates.

However, our goal was not to provide a comprehensive catalog of temperature-associated genes, but rather to test whether consistent signals could be identified across multiple independent clinal datasets. In this context, studies with weaker or inconsistent clinal signals (potentially due to altitude effects) contribute less to the set of shared candidates, but do not bias the overall conclusions.

We have updated our methods section accordingly, which now reads: “To be able to aggregate over these scales we chose “genes” as a level of abstraction, and without explicit accounting for altitude as an independent variable”.

5) The images showing defective mitotic divisions are much better now, and to this reviewer, the main defect that can be seen is a rotation on the plane of division.

We thank the reviewer for this positive assessment of the revised images and for highlighting this phenotype. We agree that the misorientation of the division axis is a very notable phenotype. To clarify terminology, we note that it is the axis of division that rotates, while the plane of division remains within the epithelial plane. Thus, divisions still occur in-plane at elevated temperature, but with altered orientation.

6) The new data provided on DNA damage with the timeline of SLBP nuclear accumulation helps build the case that the DNA damage might be caused by mitotic failures and not triggered by the high temperature, but I think this still remains quite speculative in the absence of an independent test. For example, if cycle 14 embryos are heated to 29°C, would you then never see DNA double strand breaks (e.g. using anti-H2A) at this temperature? I am not suggesting the authors need to do more experiments, but to consider future experiments to clarify the issue. Along these lines, it may indeed be better to modify the title of the paper.

We thank the reviewer for suggesting this interesting experiment. In our experience, the embryo needs to be exposed to elevated temperature from very early on to see an experimentally reproducible effect. For this reason, the suggested experiment is not without challenges. If we were to test whether an increase in temperature during cellularization can cause a loss of nuclei, the results may not be straight forward to interpret: specifically, if we do not see a loss of nuclei, it will not be clear whether there are no more double strand breaks, or whether the effect of temperature shift is not penetrant enough. Besides, the analysis in Iampietro et al., 2014, is detailed enough to convince us that the reporter we used indeed ‘responds’ to DNA damage, with the arrow of causality pointing from DNA damage to nuclear exclusion of the reporter.

In any case, it is worth noting that all of the nuclear division cycles happening till cellularization are also at elevated temperature, while the division defects are happening predominantly during mitosis 13. This suggests that, genome integrity per se is not challenged at 29°C, and that nuclear divisions are defective due to additional factors (e.g., nuclear crowding).

7) In Fig 8 C, it is hard to see the decrease in nuclear SLBP signal. Can the authors add an explanation on how the signal quantifications were done (in the entire nuclei in 3D?, in different Z positions?), and if it was before the defective nucleus moved inwards.

Thank you for highlighting this point. We did the quantifications on maximum intensity projections. We used the Histone-RFP images for drawing the regions, and then used the GFP channel (SLBP, SLBP.S118A, or NLS) for intensity measurements. Within our time frame of tracking, the defective nuclei were still close to the surface, as we could confirm that in the Histone channel. We have updated the related methods sections accordingly. The relevant text in the methods section now includes the phrase: “... we used Maximum Intensity Projected images ...”

Overall, the authors made significant effort in improving the methods, results and discussion.

We thank the reviewer for a critical evaluation of the manuscript and for the time they spent on extensive comments. We appreciate the details in the comments, which certainly improved the manuscript.

Reviewer #2 (Remarks to the Author):

The manuscript by Kale et al. describe the common failures in *Drosophila* embryonic development at high temperature. After describing changes in mitotic divisions, localization of nuclei, and DNA damage response, they identify two candidate genes that can rescue the lethality and mitotic defects and suggest these vary in natural populations from different climates. Overall, the paper explores a significant topic with an extensive set of experiments and adds new insight to existing understanding of organismal response to the changing environment.

The revised manuscript describes the authors' logic and results more clearly, and includes a more thorough description of the literature. In addition, the manuscript has been notably strengthened by more detailed characterization of the mitotic spindle and DNA damage, and adjustments to the text and figures have satisfied most of my concerns. At times, I think some statements are a bit strong and imply more direct causation than the correlations in the data, but this has improved and most of the interpretations accurately represent the results. Along these lines, I do support the alternative title suggested in the response to reviews, but I will leave it up to the authors and editor to decide on that issue.

We sincerely thank the reviewer for a critical evaluation of the manuscript and for their comments. The reviewer highlighted several blind spots in our writing, especially from the perspective of a reader who might not be familiar with the literature on embryo development in *Drosophila*. The tone of comments was highly constructive, which certainly made it easier to implement them, resulting in the improved manuscript. We are glad to see that the revised version of the manuscript is well received and appreciated.

Reviewer #3 (Remarks to the Author):

The authors have addressed my previous comments satisfactorily.

We very much thank the reviewer for their critical evaluation of the manuscript and comments. In addition to identifying loose wording, the reviewer suggested key experiments which lead to a significant improvement in the mechanistic details presented in the manuscript, hopefully leading to an increase in scientific impact. The reviewer also constructively criticized our conservative account of literature, encouraging us to link our work with existing literature more explicitly, which certainly broaden the impact of the work and strengthened the manuscript. We are glad that the additional experiments are well received and the revised manuscript is appreciated.